# Discovery of key whole-brain transitions and dynamics during human wakefulness and non-REM sleep

A.B.A. Stevner [1,2,3], D. Vidaurre [4], J. Cabral [1,5], K. Rapuano[6], S.F.V. Nielsen[7], E. Tagliazucchi[8,9,10], H. Laufs [9,10], P. Vuust[3], G. Deco[11,12,13,14], M.W. Woolrich[4], E. Van Someren[8,15] & M.L. Kringelbach[1,2,3,5]

The modern understanding of sleep is based on the classification of sleep into stages defined by their electroencephalography (EEG) signatures, but the underlying brain dynamics remain unclear. Here we aimed to move significantly beyond the current state-of-the-art description of sleep, and in particular to characterise the spatiotemporal complexity of whole-brain networks and state transitions during sleep. In order to obtain the most unbiased estimate of how whole-brain network states evolve through the human sleep cycle, we used a Markovian data-driven analysis of continuous neuroimaging data from 57 healthy participants falling asleep during simultaneous functional magnetic resonance imaging (fMRI) and EEG. This Hidden Markov Model (HMM) facilitated discovery of the dynamic choreography between different whole-brain networks across the wake-non-REM sleep cycle. Notably, our results reveal key trajectories to switch within and between EEG-based sleep stages, while highlighting the heterogeneities of stage N1 sleep and wakefulness before and after sleep.

[1] Department of Psychiatry, University of Oxford, Warneford Hospital, OX3 7JX Oxford, UK. [2] Center of Functionally Integrative Neuroscience (CFIN), Aarhus University, 8000 Aarhus, Denmark. [3] Center for Music in the Brain (MIB), Department of Clinical Medicine, Aarhus University, 8000 Aarhus, Denmark. [4] Wellcome Centre for Integrative Neuroimaging, Oxford Centre for Human Brain Activity (OHBA), University of Oxford, Warneford Hospital, OX3 7JX Oxford, UK. [5] Life and Health Sciences Research Institute (ICVS), School of Health Sciences, University of Minho, 4710-057 Braga, Portugal. [6] Department of Psychological and Brain Sciences, Dartmouth College, 03755 Hanover, NH, USA. [7] Department of Applied Mathematics and Computer Science, Technical University of Denmark, 2800 Kgs. Lyngby, Denmark. [8] Netherlands Institute for Neuroscience, 1105 BA Amsterdam, The Netherlands. [9] Department of Neurology, University Hospital Schleswig Holstein, Christian-Alrbrechts-Universität, 24105 Kiel, Germany. [10] Department of Neurology and Brain Imaging Center, Goethe University, 60528 Frankfurt am Main, Germany. [11] Center for Brain and Cognition, Computational Neuroscience Group, Department of Information and Communication Technologies, Universitat Pompeu Fabra, Roc Boronat 138, Barcelona 08018, Spain. [12] Institució Catalana de la Recerca i Estudis Avançats (ICREA), Passeig Lluís Companys 23, Barcelona 08010, Spain. [13] Department of Neuropsychology, Max Planck Institute for Human Cognitive and Brain Sciences, 04103 Leipzig, Germany. [14] School of Psychological Sciences, Monash University, Melbourne, Clayton, VIC 3800, Australia. [15] Departments of Integrative Neurophysiology and Psychiatry GGZ-InGeest, Amsterdam Neuroscience, VU University and Medical Center, 1081 HV Amsterdam, The Netherlands. These authors jointly supervised this work: D. Vidaurre, M. L. Kringelbach. Correspondence and requests for materials should be addressed to A.B.A.S. (email: angus.stevner@psych.ox.ac.uk)

The primary behavioural observation of sleep is a lack of interaction with, and responsiveness to, the external world, i.e., a decreased level of arousal[1]. The lack of communication with sleeping subjects implies that we rely on physiological recordings to scientifically describe and categorise sleep. The advent of modern neuroimaging techniques and network analyses has been explored to map and characterise spontaneous large-scale brain activity during wakefulness with high-spatiotemporal precision. Yet, our understanding of brain activity during sleep remains dictated by observations in a few channels of electro-encephalographic (EEG) recordings.

Today, the dominant description of normal human sleep is represented by polysomnography (PSG), which relies mainly on EEG but also electromyography (EMG), electrooculography (EOG) and electrocardiography (ECG), as well as measures of respiration[2]. On-going brain activity is recorded from a low number of EEG electrodes and typically categorised into wakefulness, rapid-eye movement (REM) sleep and—according to the most recent set of guidelines—three stages of non-REM (NREM) sleep (N1–N3)[2]. Staging is based on the visual detection of spectral EEG qualities (e.g., alpha- and delta-frequency power) and sleep graphoelements (sleep spindles and K-complexes), many of which have been known since the 1930s[3].

PSG has been essential in the development of modern sleep research, and remains undoubtedly the quickest and easiest way to establish arousal levels in individuals. Indeed, PSG-defined sleep stages were originally devised from EEG as surrogate markers of arousal thresholds, yet over time many have come to see them as a more or less exhaustive set of intrinsic canonical states that cover the full repertoire of brain activity during sleep. However, the use of (1) fixed scoring windows of 30 s and (2) only a few EEG electrodes means that PSG involves considerable averaging of brain activity in both time and space[4]—arguably leading to an incomplete representation of brain activity.

Furthermore, PSG corresponds relatively poorly to the subjective perception of sleep. Participants may experience being awake during periods with EEG signals otherwise fulfilling PSG criteria of NREM sleep[5,6]. The relative lack of correspondence between PSG and subjective experience becomes important in populations with sleep complaints, where PSG is not indicated in the clinical evaluation of insomnia, the most common of all sleep disorders[7,8].

Recent developments in whole-brain neuroimaging and analyses support the examination of more sophisticated features of brain networks through functional connectivity (FC) and structural connectivity analyses, the detection of task-related and resting-state functional networks[9,10], and the development of mechanistic computational models[11,12]. Yet, studies that have applied these promising tools to investigate large-scale brain activity of sleep have commonly relied upon PSG in a strict sense, thus regressing PSG stages onto functional brain data. This approach has yielded whole-brain correlates of PSG stages and sleep graphoelements, in terms of activation maps[13,14], FC patterns[15–19], graph-theoretical measures[20,21] and EEG-microstates[22]. However, this top-down constraint by the low-resolution PSG scoring comes at the cost of exploring only a small fraction of the information available in the high-resolution neuroimaging data.

Rather than constraining analyses by traditional definitions of sleep stages, we propose to use novel data-driven analysis methods to elucidate whole-brain networks that can complement and potentially expand the classical understanding of sleep. This requires a sufficiently sensitive decomposition of whole-brain network activity in time. Building on a recent study showing that individual PSG stages can be extracted from functional magnetic resonance imaging (fMRI) recordings in a data-driven way[23], we

here leveraged the full spatiotemporal resolution of blood-oxygen-level dependent (BOLD) signals to find large-scale networks in sleep, applying a Hidden Markov Model (HMM)[24] on fMRI recordings of 57 healthy participants, who—according to simultaneously acquired EEG—cycled through PSG-defined stages of wakefulness and NREM sleep. Crucially, the HMM decomposition was not constrained by PSG stages, but rather allowed us to discover directly from the data, at a time-scale of seconds, the relevant brain network transitions explored by the human brain during the wake-NREM sleep cycle. Compared to other methods for extracting dynamic FC[25], the HMM framework explicitly models the transition probabilities between its inferred states. We show that this information can be used to discover new whole-brain aspects of sleep, complementing the traditional segmentation of brain activity offered by PSG.

## Results

**Whole-brain network states identified by HMM.** In order to extract the large-scale networks inherent to whole-brain recordings of the wakefulness-NREM sleep cycle, we estimated an HMM on fMRI data from 57 healthy participants (age 23.5 ± 3.3 years, 39 females). Participants were instructed to lie still in the scanner with their eyes closed. Each recording had a duration of 52 min, and was accompanied by acquisitions of EEG, EMG, ECG and EOG, based on which PSG staging was performed by an expert, according to the AASM criteria[2] (see Supplementary Table 1). Following preprocessing, the voxel-wise BOLD time-courses were temporally averaged over 90 region-of-interest (ROI) timecourses, using the cortical and subcortical regions of the automated anatomical labelling (AAL) atlas[26]. ROI time-courses were demeaned and variance-normalised for each participant, and subsequently concatenated across participants along the temporal dimension.

The estimated HMM contained a set of whole-brain network states, each defined as a multivariate Gaussian distribution, including: (i) a mean activation distribution, representing the mean level of activity in each ROI when a state is active; and (ii) an FC matrix, summarising the pairwise temporal co-variations occurring between the ROIs during that state. The HMM also contained a transition probability matrix with the probabilities of transitioning between each pair of states. Each state also had an associated state timecourse describing the points in time (defined by the fMRI sampling, TR = 2.08 s) where the state was active[24,27]. The HMM was endowed with 19 states, and, crucially, was given no information about the PSG staging for its estimation. An illustration of the analysis workflow is given in Fig. 1 (see Methods for details).

To allow for unbiased within-participant testing, when comparing the HMM output to the PSG scoring we considered the subset of the HMM output that corresponded to the data from the 18 participants that reached all four PSG stages (wakefulness, N1, N2 and N3, see Supplementary Table 1).

**Whole-brain network states underlie PSG sleep stages**. The 19 whole-brain network states, inferred solely from the fMRI by the HMM, contained most of the temporal information given by the PSG stages that were scored from the EEG independently of the fMRI. The HMM state timecourses and the PSG scoring are plotted together in Fig. 2a for the 18 participants that included all four PSG stages, illustrating how the activity of the different HMM states co-varied with specific PSG stages. We have highlighted the HMM state timecourses of two participants to ease visual inspection, however, the temporal relationships between HMM states and PSG stages were consistent across the group. It can be observed, for example, that HMM state 8 occurred most

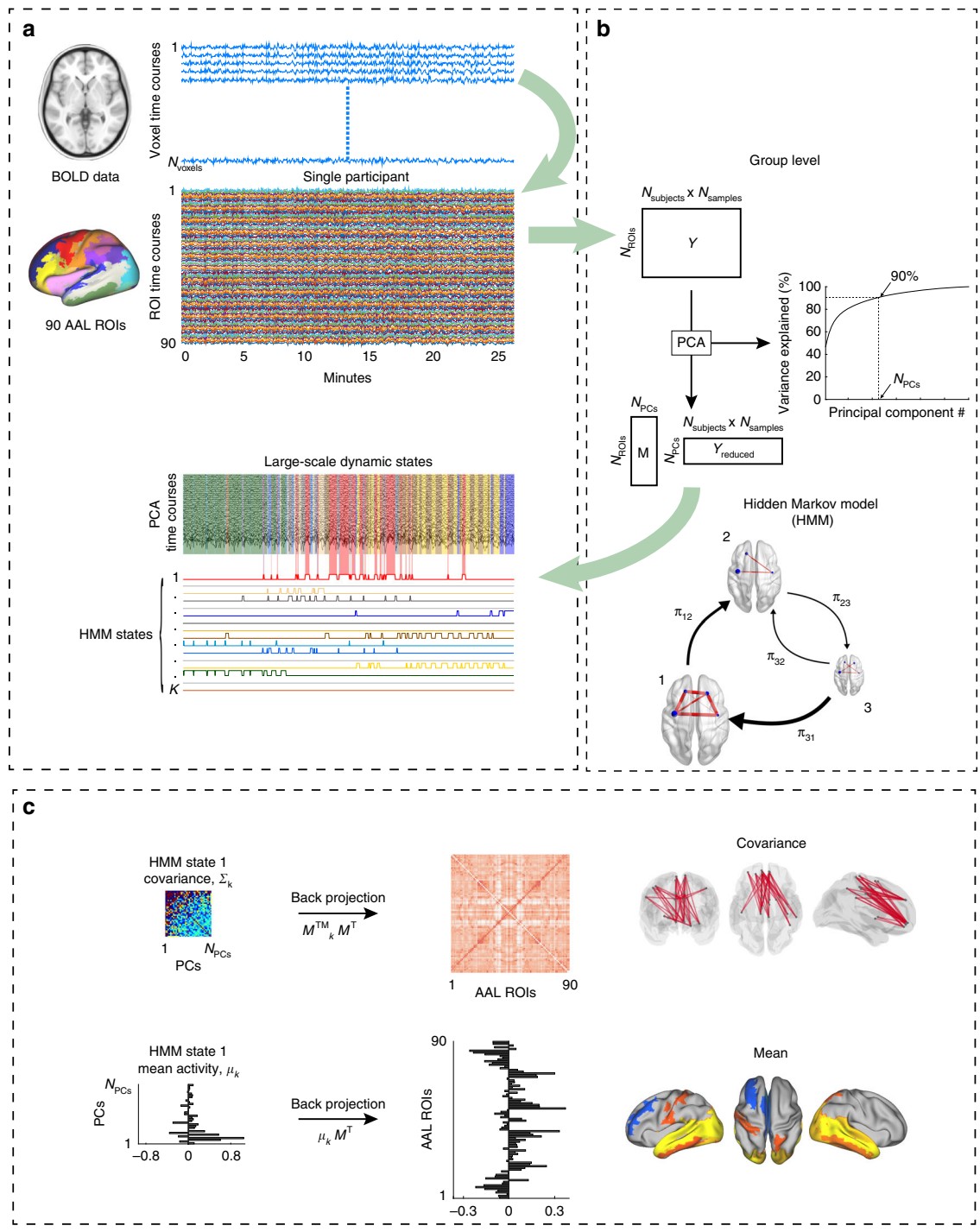

**Fig. 1** Dynamic whole-brain networks from fMRI sleep recordings using a Hidden Markov Model. **a** ROI timecourses were extracted by averaging BOLD signals across voxels within each of the 90 cortical and subcortical AAL areas for each participant. Each ROI timecourse was demeaned and normalised by its standard deviation. **b** The data were concatenated across participants, and the dimensionality was reduced using PCA (principal component analysis), such that ~90% of the variance of the ROI timecourses was retained. The HMM was run on the PCA timecourses, resulting in $K$ number of states with associated timecourses, each describing the points in time each state is active and inactive. **c** Each HMM state was characterised by a multivariate Gaussian distribution comprising a covariance matrix, $\Sigma_K$, and a mean distribution, $\mu_K$. The state-specific mean distributions and covariance matrices were back-projected to the MNI space of the AAL by using the mixing matrix, $M^T$ from the PCA decomposition, yielding a mean activation map and an FC matrix for each HMM state

often during wakefulness, HMM state 3 occurred during N2 sleep and HMM state 16 occurred during N3 sleep.

We quantified the temporal association between the PSG stages and the HMM state timecourses using multivariate analysis of variance (MANOVA). This allowed us to ask if the 19 HMM states were significantly grouped in time by the four PSG stages (for the 18 participants that included all four PSG stages). Through non-parametric testing (see Methods) we confirmed this temporal relationship ($p < 0.05$, permutation testing, see Supplementary Figure 1b). The MANOVA placed the PSG stages in the

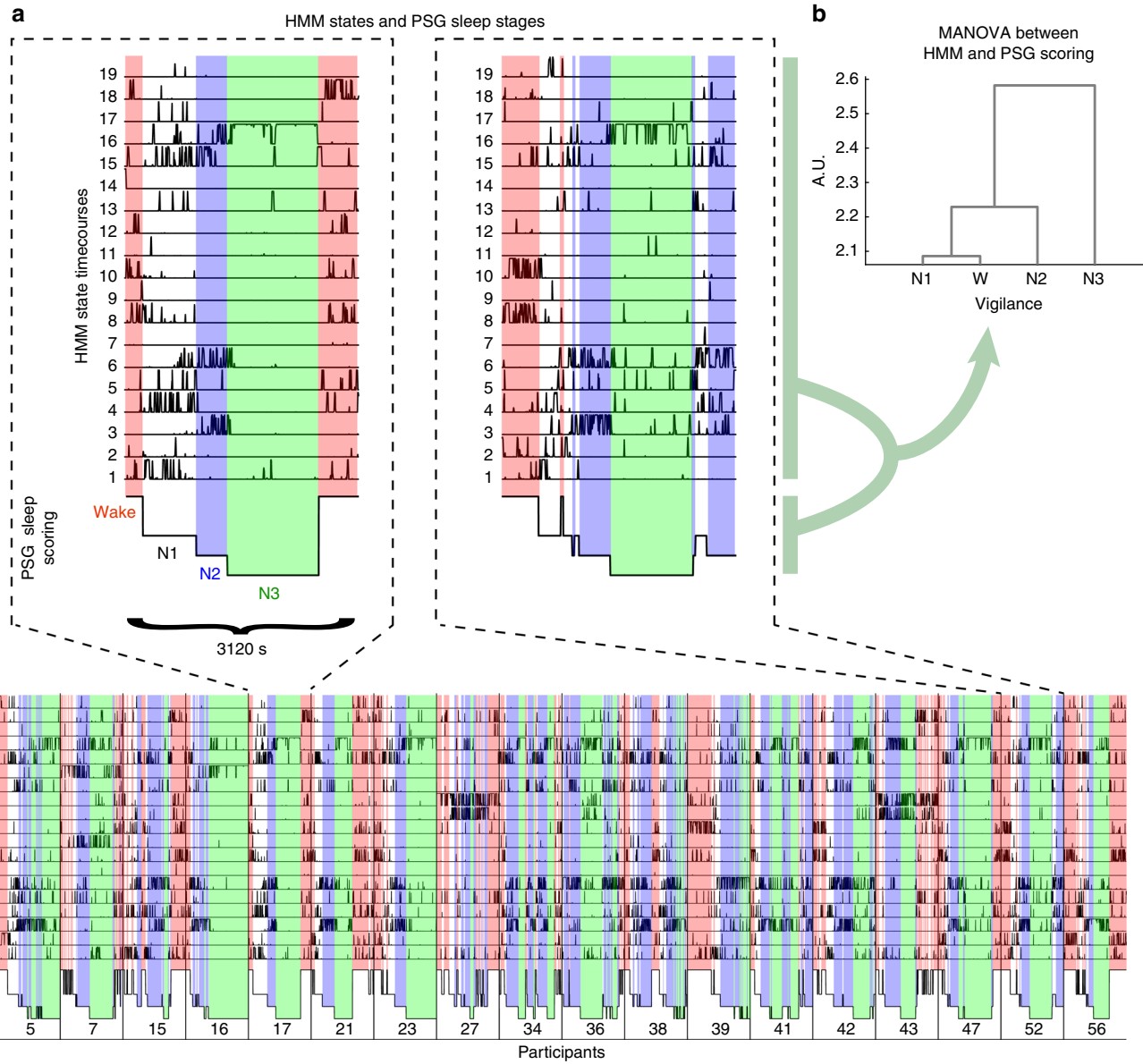

**Fig. 2** State timecourses of whole-brain network states and their association to polysomnography. **a** The figure shows the 19 HMM state timecourses describing each state's probability of being active at each sample point of the fMRI sessions in the 18 participants that reached all four PSG stages. Below the HMM state timecourses are shown the independently obtained PSG sleep scoring (based on the simultaneously acquired EEG). The coloured overlay shows periods scored as wakefulness (red), N1 (white), N2 (blue) and N3 (green). The two dashed boxes highlight the HMM state timecourses and PSG scoring of two representative participants. Note, how the majority of HMM timecourses varied with the PSG stages, in highly consistent ways across participants. A few 'sporadic' HMM states, occurring mainly in a few participants, are also visible (e.g., states 11 and 12). **b** Quantifying the multivariate relationship between the HMM states and the PSG scoring, through the use of MANOVA, revealed a hierarchical grouping of the HMM states, in which wakefulness and N1 sleep were separated from N2 sleep, which in turn was separated from N3

space of the HMM state timecourses, resulting in the clustering dendrogram of Fig. 2b, with wakefulness and N1 sleep significantly separated from N2 sleep, which in turn was further separated from N3 sleep.

**Whole-brain network states track different PSG stages**. Next, we examined the contribution of the individual whole-brain network states to the multivariate relationship, established above, between the HMM and the PSG scoring. We quantified the temporal sensitivity and specificity of the HMM states for each of the PSG stages. For each of the 18 participants that included all four PSG stages, we defined the sensitivity of each HMM state as the proportion of total time spent in a PSG stage, in which this

HMM state was active. Specificity was defined as the likelihood of finding each HMM state active during a given PSG stage. We compared the sensitivity and specificity for each PSG stage within each of the HMM states, using paired *t* tests and a randomisation scheme of the PSG scoring (see Methods). The results are presented in Fig. 3a, b. HMM state 8 occupied a large proportion of PSG-scored wakefulness, i.e., it exhibited high sensitivity for wakefulness (see Fig. 3a). Since this whole-brain network state was significantly more sensitive to wakefulness than to any of the other PSG stages, i.e., it rarely occurred outside of wakefulness, its specificity for wakefulness was also high (see Fig. 3b). This combined sensitivity and specificity for wakefulness was also found for HMM states 10 and 18.

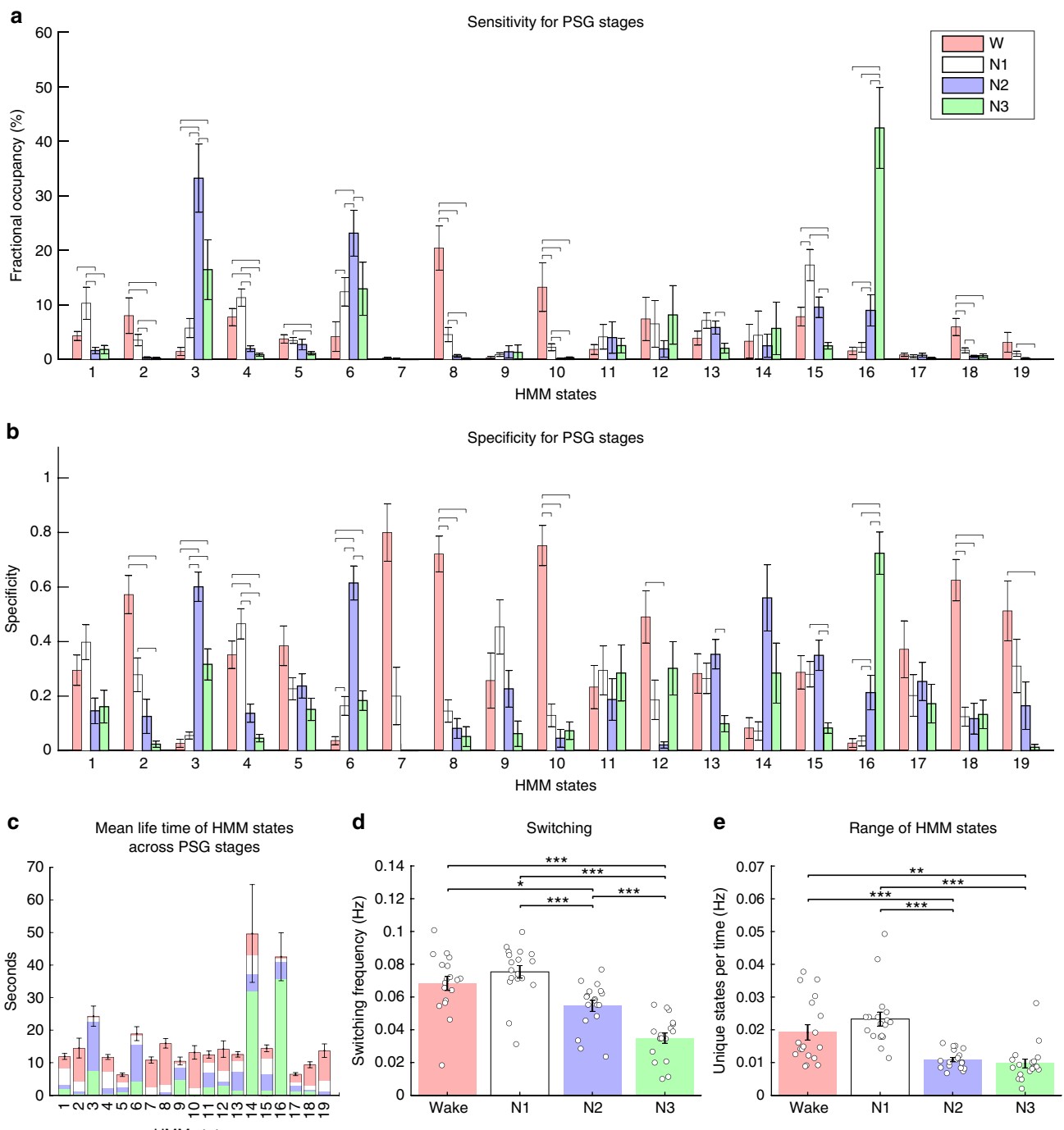

**Fig. 3** Sensitivity and specificity of HMM states and dynamics within polysomnography stages. **a** Fractional occupancies of each of the 19 HMM states computed within the four PSG stages corresponded to the PSG-sensitivity of the whole-brain network states. The coloured bars and error bars show the average and standard error, respectively, across the 18 participants that included all four PSG stages. **b** PSG-specificity of the HMM states for each of the four PSG stages. Specificity corresponds to the probability of an HMM state occurring within a PSG stage. The bars represent the group average and the error bars the standard error ($n = 18$). In **a** and **b** horizontal lines show significant differences within HMM states, with $p$ values < 0.01 as evaluated through paired $t$ tests and permutation testing. **c** The mean life times of the 19 HMM states are shown by the bars, representing values averaged across the 18 participants. Error-bars represent the standard error across participants. Each HMM state is coloured according to the probability of finding it within each of the four PSG stages, i.e., their PSG specificity. Note how HMM states with high specificity for N3—and to a lesser extent N2—exhibit longer mean life times. **d** The dynamics of the HMM transitions were calculated within each of the four PSG stages, in terms of switching frequency ('Switching'), and **e** the number of different HMM states visited per time ('Range of HMM states'). These measures significantly separate the four PSG stages suggestive of a higher dynamical repertoire during wakefulness and N1. In **d** and **e** error bars represent standard error across participants and significant differences between PSG stages are denoted by stars: one star: $p < 0.05$, two stars: $p < 0.01$ and three stars $p < 0.001$; all evaluated using paired $t$ tests and permutations. W: wakefulness, N1: N1 sleep, N2: N2 sleep, N3: N3 sleep

Select whole-brain network states displayed similarly exclusive sensitivity and specificity profiles for N2 (HMM states 3 and 6) and N3 sleep (HMM states 16). Notably, this was not the case for N1 sleep. The whole-brain network states occupying most of N1 sleep, such as HMM states 1, 4 and 15, were not found specific for this PSG stage. Instead these states would also occur with considerable likelihood outside of N1 sleep, although rarely during N3 sleep.

In summary, wakefulness was found to correspond to a collection of whole-brain networks states, while N2 and N3 were characterised by less state-diversity, and dominated by two and one whole-brain states, respectively. In contrast, no single whole-brain states were found specific for N1 sleep, which instead was modelled by a collection of HMM states with mixed PSG profiles.

**Changes in whole-brain network dynamics between PSG stages.** Having the whole-brain network states temporally defined allowed us to investigate the large-scale brain dynamics of the traditionally defined PSG stages in the 18 participants that reached all PSG stages during their recordings.

In Fig. 3c, the HMM states are represented by a bar plot showing their mean lifetimes, i.e., the average duration of the state visits. The bars have been overlaid with colours depicting the PSG specificity averaged across the corresponding HMM states. HMM states with high specificity for N2 and N3 (HMM states 3, 6 and 16) generally expressed longer mean lifetimes than those related to wakefulness and N1. The mean lifetimes of the HMM states ranged from seconds to tens of seconds.

Figure 3d, e shows two summary measures for the dynamics of the whole-brain network states during the individual PSG stages: (i) the amount of switching defined as the average number of transitions between HMM states during a given PSG stage divided by the total time a participant spent in this PSG stage and (ii) the range of HMM states defined as the number of unique states visited during the given PSG stage divided by the total time a participant spent in this PSG stage. Both measures were estimated for each PSG stage, within each of the 18 participants that included all four PSG stages, and normalised by time. Wakefulness and N1 sleep expressed significantly higher values than N2 and N3. Interestingly, the amount of switching was particularly low for N3 sleep.

In summary, unique state visits per time were few and of long durations during N2 and N3 relative to wakefulness and N1 sleep. Consequently, the switching between and range of HMM states were significantly higher in wakefulness and N1.

**Sleep stages as modules of whole-brain network transitions.** So far, we have used the traditional PSG stages to organise and evaluate the temporally resolved whole-brain network states. Yet, the data-driven nature of the HMM also allowed us to perform reverse inference, and consider the temporal progression of HMM states, taking this—rather than the PSG staging—as a starting point. This way, we were able to ask if the high-resolution, fMRI-based, HMM suggests new aspects of the wake-NREM sleep cycle, hidden from the EEG-based PSG. For this purpose, we examined the transition probabilities of the HMM states, extracting modules of HMM states that transitioned more often between each other than to other states—as recently identified for the waking resting state in ref. [24].

The whole-brain network states organised into a transition map as presented in Fig. 4, where the $19 \times 19$ transition probability matrix (Fig. 4a) was submitted to a modularity analysis (see Methods). By considering the most frequent transitions between the HMM states that were consistent across participants (see Fig. 4b), the thresholded transition matrix organised into four

partitions or transition modules (see Fig. 4c, and Methods), suggestive of a lower time scale (see ref. [24] and Supplementary Discussion 1). When these most consistent transitions are presented as a transition map, and each whole-brain network state is represented by a circle plot indicating its specificity for each of the four PSG stages, it can be seen that the HMM states exhibit a strong temporal structure (Fig. 4d). In line with the MANOVA results above, this transition map describes an overall progression from states with high specificity for PSG-defined wakefulness (red module) through states with more activity during, albeit not significant specificity for, N1. From here transitions lead towards states specific to N2 sleep and finally to a single whole-brain network state modelling N3 sleep. The N2- and N3-related HMM states thus grouped together in the blue module.

Interestingly a collection of HMM states with mixed PSG-specificity formed a transition module of their own. This white module was intercalated between the red module of wakefulness in the top and the blue module of N2/N3 sleep. Even if the included HMM states were not specific for PSG-defined N1 sleep, the white module appears in the location of the transition map, where one would expect to find N1 or rather sleep onset.

The transition map suggested two sub-divisions of HMM states with high specificity for wakefulness. The red module in close proximity to the white module of N1-related states, and the black module sending transitions to the blue module of consolidated NREM sleep. This apparent separation of wakefulness and the asymmetric relationship to the sleep-related HMM states led us to the hypothesis that one of these could represent wakefulness after sleep onset (WASO). Given the poor correspondence between the HMM states and the general uncertainty associated with the staging of PSG-defined N1 sleep (see Discussion), we chose to define WASO as PSG-staged wakefulness, which followed after visits to N2 sleep[28]. By computing the sensitivity and specificity of the whole-brain network states in the subset of the data corresponding to the 31 participants who woke up after having reached N2 sleep (see Supplementary Table 2), we were able to confirm this hypothesis. As shown in Supplementary Figure 2, HMM states 5, 17 and 18 were all more sensitive and specific to WASO compared to wakefulness prior to N2 sleep. Whereas periods of wakefulness prior to and after sleep are scored equally in PSG, the whole-brain network states separated these into two different transition modules.

Although this transition map suggests multiple pathways from wakefulness (red module) to the white module of NREM sleep, it is interesting to note that HMM state 8 has direct access to HMM state 15, which in turn guards the transition to the blue module of N2/N3 sleep. Similarly, waking-up relates to a transition from HMM state 4 to HMM state 10, which in turn connects with HMM state 18 of the black WASO module. Further it is worth noticing the strong triangular transition structure within the blue module between the N2-specific whole-brain network states (HMM states 3 and 6) and the N3-modelling HMM state 16.

In summary, while agreeing with the overall sequence of PSG stages, the organisation of the transition modules also points to aspects of sleep-related brain activity that the PSG scoring cannot access, including the data-driven suggestions of N1 sleep, WASO-related whole-brain network states, and multiple transition pathways between wakefulness and sleep.

**Spatial activation and FC maps of whole-brain network states.** We present the spatial maps of the whole-brain network states in the order suggested by the transition modules of Fig. 4d. Figure 5 and Fig. 6 show the mean activation maps, while the corresponding FC information is presented in Supplementary Figures 3–5 and 17–18 (see also Supplementary Note 5).

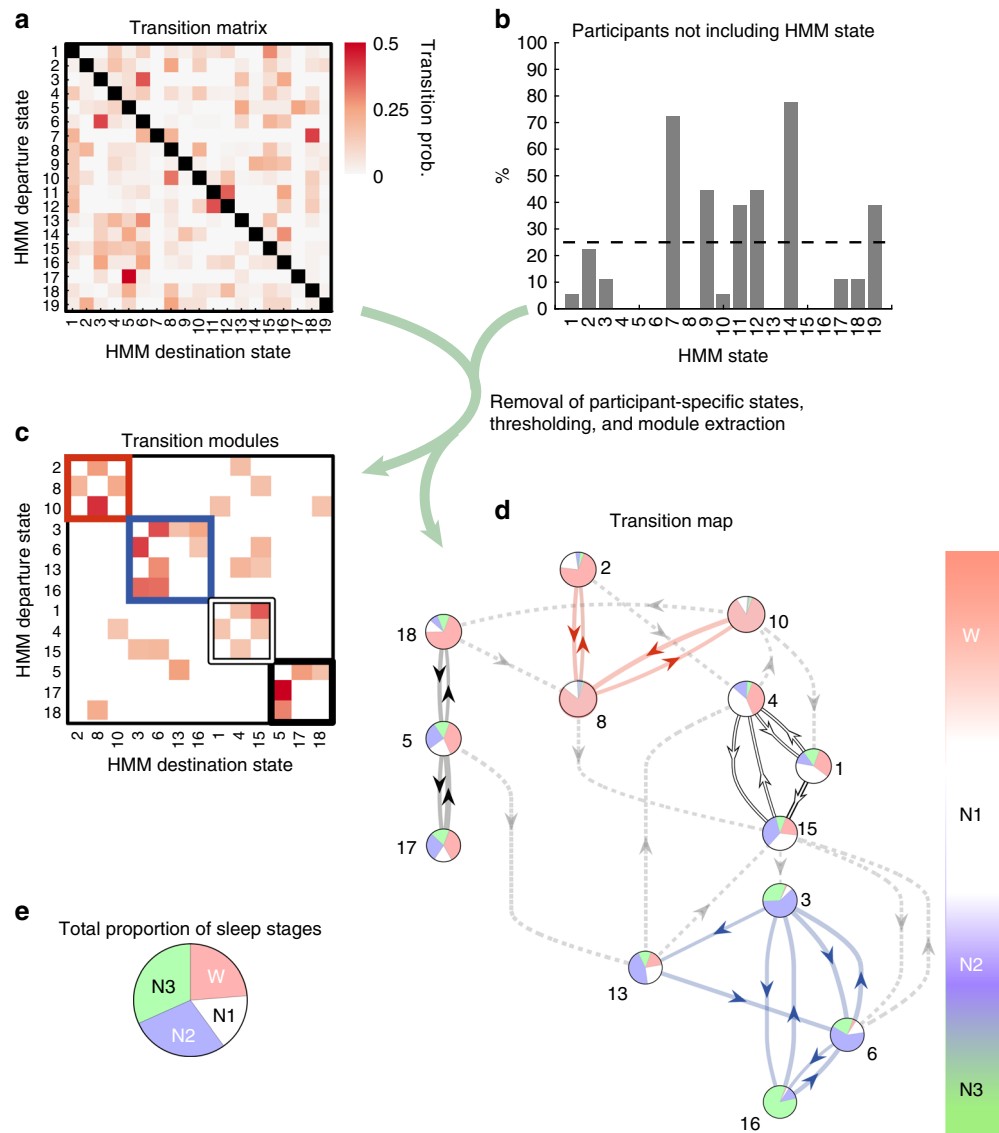

**Fig. 4** Investigating modules of transitions between whole-brain network states. **a** The figure shows the 19 × 19 transition probability matrix of the HMM states calculated for the 18 participants that included all four PSG stages in their respective scanning session. This quantifies the likelihood of transitioning from any given state to any other state, yielding each matrix entry: transition probability from departure state to destination state. **b** A few HMM states were 'sporadic' and did not occur consistently across participants. HMM states not occurring in more than 25% of the participants were excluded. **c** The strongest transitions of the consistent HMM states were partitioned through a modularity analysis, and reorganised in a matrix according to the four resulting modules. **d** The transitions shown in **c** are presented as a transition map with each state depicted as a pie plot expressing its specificity for each of the four PSG stages. Arrows show the direction of the transitions with thickness proportional to the transition probability. The transitions describe a passage from HMM states with high activity during wakefulness in the top, further down through HMM states including more N1, and down to HMM states specific to N2 and finally N3. Interestingly, wakefulness appears to be represented by two modules (red and black). Even though no individual HMM state showed clear specificity for N1 sleep, a module (white) is evident between HMM states with specificity for wakefulness and HMM states with specificity for N2 and N3 sleep. **e** Pie chart showing the total proportion PSG stages within the 18 participants. Transition prob.: transition probability

In Fig. 5a, which shows the red module of wakefulness, the mean activation maps of HMM states 2 and 8 resemble resting-state network (RSN) configurations[9,10]. The main increases of HMM state 8 were thus seen in key areas of the default-mode network (DMN)[29], including the bilateral posterior cingulate cortex, bilateral angular cortex, bilateral middle temporal cortex, and bilateral medial prefrontal cortex. These DMN-like increases in HMM state 8 were accompanied by decreases in the so-called anti-correlated network (ACN), involving the supramarginal gyrus and the dorsolateral part of the frontal cortex[30]. In contrast, HMM state 2 was characterised by increases in many of these ACN-areas, including the bilateral supramarginal gyrus, middle

cingulate cortex and dorsolateral part of the frontal cortex. These results suggest an inverse relationship between the activity of the DMN and the ACN, which is an established trait of these RSNs[30]. Since the discovery of these RSN patterns they have been hypothesised to reflect complex cognitive processes. The DMN has been linked to inwardly directed mentation, such as autobiographical memory and mind wandering[31,32]. The ACN overlaps with areas also referred to as the dorsal attention network[33] or the central executive network (CEN)[34], and has been proposed to be involved in more externally directed processes, including attention[35]. In agreement with this, we found these high-order RSNs to be relatively exclusive for

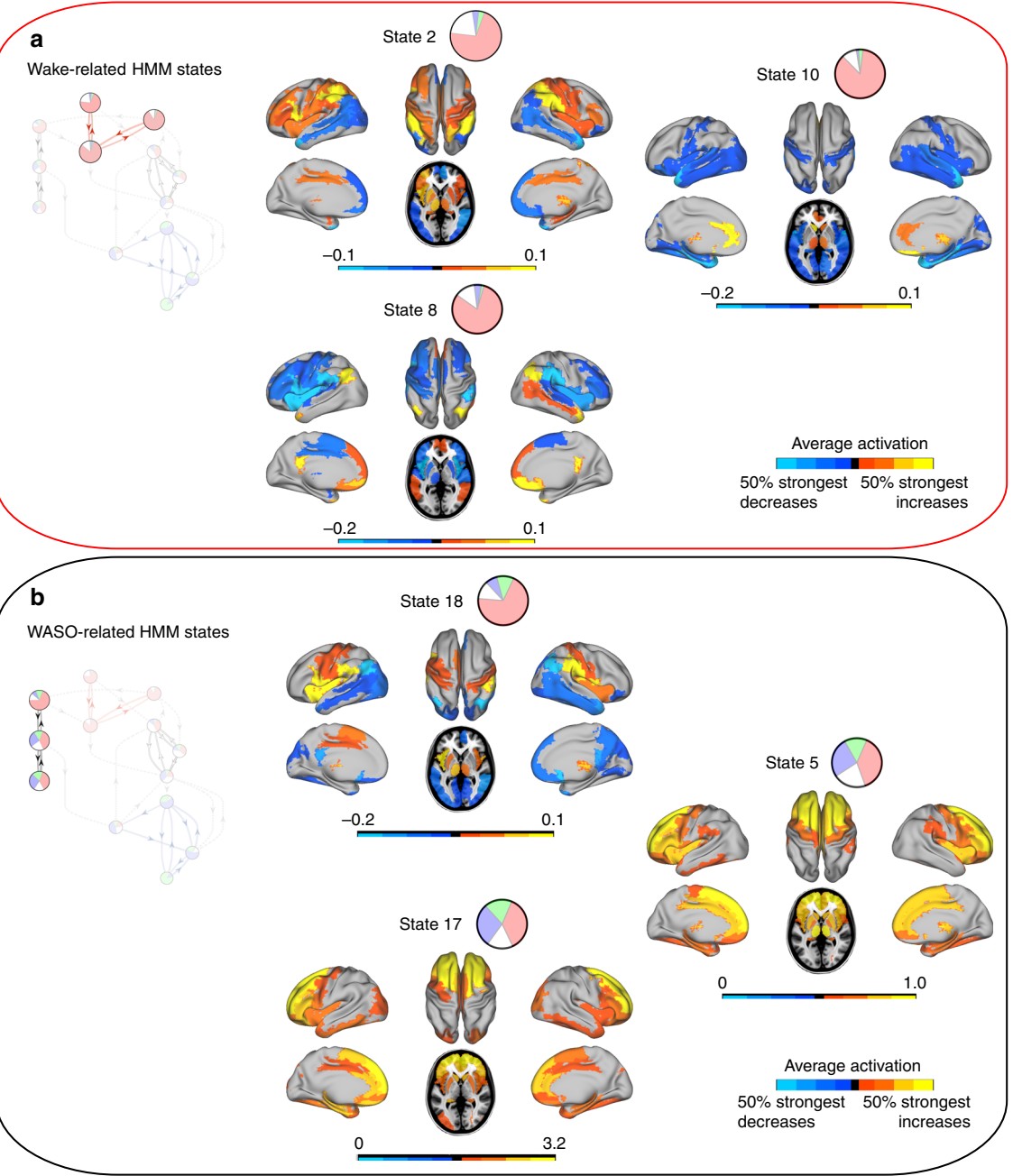

**Fig. 5** Mean activation distributions of wakefulness-related HMM states. **a** Three HMM states identified as sensitive and specific to wakefulness prior to sleep. Note HMM state 8 with DMN-like configuration increases, and concomitant decreases in DAN/CEN-like areas. **b** Three HMM states associated to wakefulness after sleep. There are marked frontal increases in mean activation in HMM states 5 and 17. All maps were thresholded above the 50% strongest positive and negative changes, respectively

wakefulness. However, previous investigations have suggested a rather ubiquitous presence of both the DMN and the ACN, not just in wakefulness but, in all stages of NREM sleep[15,17,36] (see Supplementary Discussion 1).

Figure 5b shows the mean activation maps of whole-brain network states with higher sensitivity and specificity for WASO (black module). The mean activation map of HMM state 18 expressed a distribution similar to that of HMM state 8 (see Fig. 5a), but with opposite signs. Hence, HMM state 18 showed decreases in DMN-related areas, and increases in regions overlapping the ACN. HMM states 5 and 17 were both characterised by mean activation increases in the frontal cortices. Interestingly, findings from high-density EEG studies of participants waking up

from sleep show that the posterior parts of the cortex are particularly 'slow' at returning to levels of activity seen prior to sleep[37]. Consistent with this, converging evidence from PET and fMRI have indicated frontal cortical activity to be increased relative to that of posterior areas upon awakening[28,38,39].

The whole-brain network states of the white N1-related module are represented in Fig. 6a. A general observation for these spatial maps is the inverse relationship between mean activation in subcortical areas (thalamus and parts of the basal ganglia) and primary sensory cortical areas. Increases in subcortical activity were accompanied by decreases in primary sensory areas of the cortex and vice versa. This was true for HMM states 4 and 15 (and HMM state 1 although its decreases were not

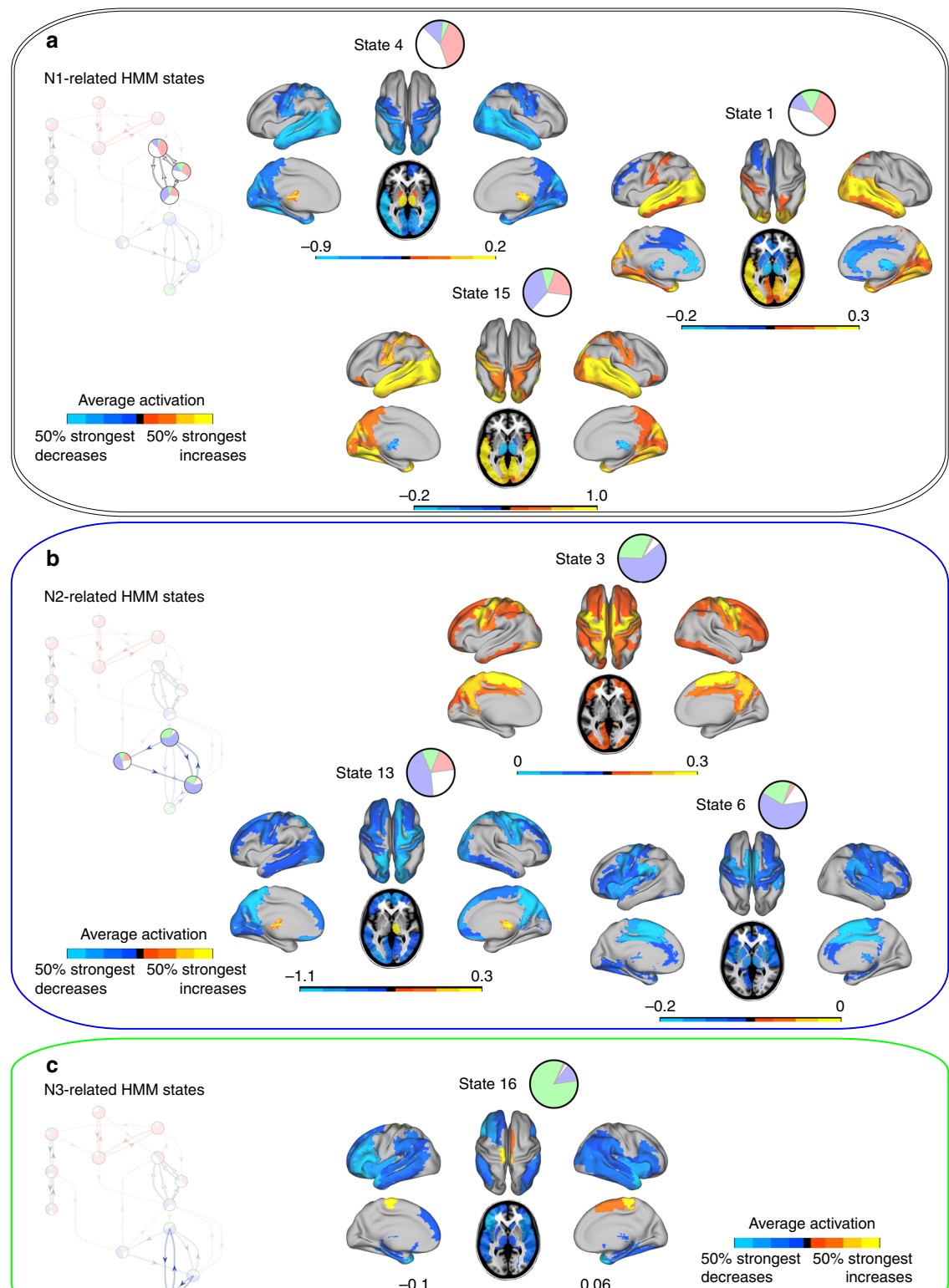

**Fig. 6** Mean activation distributions of sleep-related HMM states. **a** HMM states associated with N1 sleep showed opposite signs in mean activation in subcortical areas and primary sensory areas of the cortex. **b** Three HMM states related to N2 sleep. HMM states 3 and 6 in particular showed peak increases and decreases, respectively, in areas previously identified as fMRI-correlates of sleep spindles. **c** HMM state 16 is dominating slow wave sleep (N3). Interestingly, there are marked decreases in mean activation in frontal areas and insula, and very localised increases in the supplementary motor area and paracentral lobule. All maps were thresholded above the 50% strongest positive and negative changes, respectively

confined to subcortical areas, but supplemented by decreases in the anterior and middle cingulate cortex). This is consistent with intracortical studies of the sleep onset process in rats[40] and in humans[41] showing that thalamic changes in dynamics precede those of cortical areas near the onset of NREM sleep. Previous fMRI studies of NREM sleep have suggested decreased connectivity between the thalamus and cortical regions as perhaps the most consistent trait of FC during N1 sleep[16,18,19,23].

N2 sleep was dominated by HMM states 3 and 6, and the mean activation maps of these whole-brain network states are shown in Fig. 6b. The supplementary motor area was involved in both of these states; in HMM state 3 as increases in concert with the bilateral precuneus and primary motor cortices; and in HMM state 6 as decreases together with the bilateral thalamus, middle cingulate, supramarginal cortex, and the rolandic operculum. Interestingly, these configurations overlap considerably with those previously reported in studies mapping fMRI-correlates of sleep spindles[42,43], which represent a defining EEG-feature of N2 sleep. However, no HMM state appeared to be driven solely by either sleep spindles or K-complexes. By identifying sleep spindles and K-complexes from the EEG data, we assessed the temporal relationships between these graphoelements and the HMM states. In summary the HMM states that were dominant during N2 sleep showed comparable sensitivity and specificity to both types of graphoelements, and hence the HMM did not appear to have assigned individual states for either spindles or K-complexes (for further information please see the Supplementary Discussion 1, Supplementary Note 4, Supplementary Table 3, and Supplementary Figures 19–21).

HMM state 16 accounted for the majority of time spent in N3 sleep. The corresponding mean activation map is shown in Fig. 6c. Apart from some very localised increases in the paracentral lobule and adjacent supplementary motor area the mean activation was characterised mainly by decreases, particularly in the bilateral middle and superior temporal pole, the orbital part and the operculum of the inferior frontal cortex, bilateral insula as well as medial temporal areas. These frontal decreases are consistent with previous PET findings of decreased metabolism in these areas during N3 sleep, which in turn are believed to reflect the high, localised concentration of slow-wave activity[44].

## Discussion

Using a data-driven exploration of large-scale brain networks and associated dynamics from continuous fMRI recordings, we have explored the rich dynamical complexity in spatiotemporal patterns of brain activity during the healthy wake-NREM sleep cycle. Moving beyond the traditional PSG stages of sleep, we used a HMM to extract 19 recurring whole-brain network states, defined in space by patterns of mean BOLD activation and FC, and defined in time as the probability of being active at each time point of the fMRI sampling. Comparing the temporal evolution of the HMM-derived whole-brain network states with the independently obtained EEG-based PSG scoring, we have discovered a rich repertoire of brain dynamics underpinning the traditional PSG stages. The temporal resolution of the HMM identified state lifetimes on the order of seconds, providing a temporally fine-grained description of the traditional PSG stages. Crucially, a close examination of the HMM transition map furthermore revealed a heterogeneity of large-scale network activity that PSG cannot fully capture.

The description of brain activity offered by PSG has for long been acknowledged as incomplete, and attempts have been made to harvest more information from scalp EEG in a search for features relevant for sleep, overlooked by PSG[45–47]. Our work adheres to this aim, while, through fMRI, incorporating evidence

of whole-brain spatial detail. Previous studies have indicated that fMRI can be used to identify dynamic re-configurations of large-scale brain activity during the conventional EEG-based sleep stages, either in form of voxel-wise changes in activity[48], changes in connection strengths in resting-state networks[49] or through long-range temporal dependencies in the BOLD signal[36]. Rather than direct reflections of the conventional sleep stages, what has emerged from our HMM analysis is a probabilistic representation of the PSG scoring in the space of whole-brain network states and transitions. Agreements as well as disagreements between the PSG scoring and the independent HMM decomposition became clear in the transition map (Fig. 7). Wakefulness, N2 sleep and N3 sleep were each represented by one or more whole-brain network states, forming a good correspondence with PSG. In contrast, no states were found specific for N1 sleep. Furthermore, while treated equivalently in PSG staging, wakefulness prior to sleep and WASO were represented in the transition map as two different modules with different repertoires of large-scale brain networks. Consequently, the transition map also identified specific whole-brain network transitions underlying the descent to, and ascent from, NREM sleep.

Consistent with previous neuroimaging studies that have used regression analyses to identify consistent differences between traditionally defined sleep stages in terms of large-scale brain activity[13,14,50], PSG-defined wakefulness, N2 and N3 sleep each corresponded well to specific collections of whole-brain network states (see Supplementary Discussion 1). However, the HMM additionally provided access to the large-scale brain dynamics of the PSG stages, showing that the state repertoire, when estimated as amount of switching and range of states visited, is higher in wakefulness than in both N2 and N3 sleep. That a higher and more complex state repertoire is important for the brain to support wakeful consciousness follows from theoretical frameworks[51–53] and has received empirical support from a series of combined TMS and EEG studies[54]. From a large-scale network perspective fMRI has been used to show how an enhanced state repertoire is associated with an 'expanded' consciousness during the psychedelic experience[55,56]. In the context of sleep, however, the large-scale network evidence is mainly represented by static FC studies suggesting decreased information integration during N2 and N3 sleep using graph theory[20,21], as well as a higher exploration of the structural connectome during wakefulness[57]. Here, we have provided more direct evidence of a higher state repertoire in whole-brain dynamics during wakefulness.

The transition map identified a key trajectory from wakefulness in the red transition module to NREM sleep in the white transition module (see Fig. 7d), represented by the transition departing from the whole-brain network state with increased mean activation in the DMN. The proposed association between the DMN and inwardly directed mentation[31,32] makes this finding intriguing, in the sense that it may suggest a role for the DMN as a 'gate' in the process of initiating sleep. Whole-brain network evidence of sleep initiation may improve our understanding of sleep disorders like insomnia where PSG criteria are difficult to apply[7,58], and hypersomnia disorders[59]. Related hereto, a recent study identified switching instability to and from N2 sleep, together with difficulties reaching N3 sleep as important traits of insomnia[60]. In the transition map we saw N2- and N3-related whole-brain network states forming a strong triangular loop of transitions (see Fig. 7e). This stable configuration of transitions may not be present in people suffering from insomnia.

The two main incongruities between the temporal segmentation suggested by the HMM and the PSG scoring concerned N1 sleep and WASO. N1 sleep did not correspond to any single state or any group of states identified by the HMM. This is likely related to the current consensus that PSG-defined N1 does not represent a

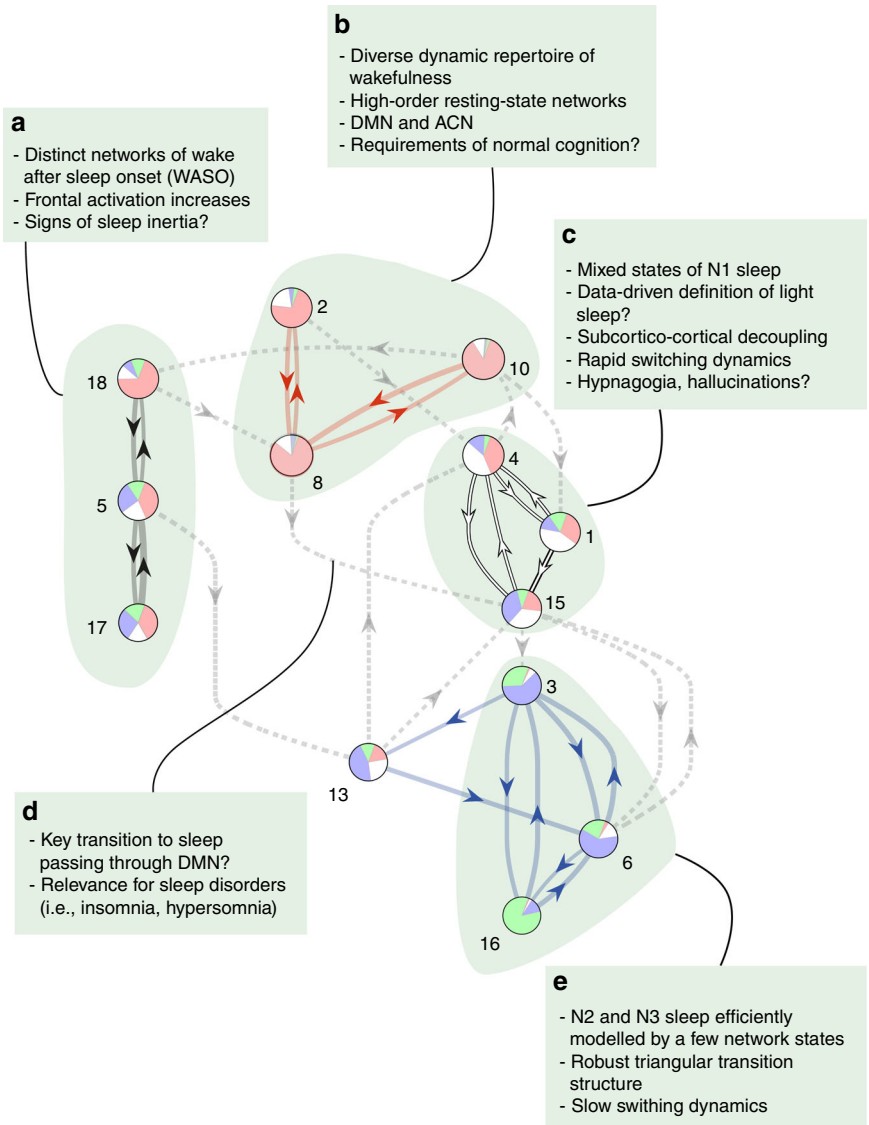

**a**
- Distinct networks of wake after sleep onset (WASO)
- Frontal activation increases
- Signs of sleep inertia?

**b**
- Diverse dynamic repertoire of wakefulness
- High-order resting-state networks
- DMN and ACN
- Requirements of normal cognition?

**c**
- Mixed states of N1 sleep
- Data-driven definition of light sleep?
- Subcortico-cortical decoupling
- Rapid switching dynamics
- Hypnagogia, hallucinations?

**d**
- Key transition to sleep passing through DMN?
- Relevance for sleep disorders (i.e., insomnia, hypersomnia)

**e**
- N2 and N3 sleep efficiently modelled by a few network states
- Robust triangular transition structure
- Slow swithing dynamics

**Fig. 7** A whole-brain network perspective on the human wake-NREM sleep cycle. The main discussion points of our results have been highlighted in the transition map of the whole-brain network states. Boxes a to e summarise the new perspectives provided by our HMM analysis

clear-cut sleep stage[61], but rather an ill-understood mix of wakefulness and sleep. Compared to N2 and N3 sleep with their well-defined EEG spectral properties, such as K-complexes, spindles, and slow waves, N1 remains the most vaguely defined sleep stage within PSG. A recent report by the American Academy of Sleep Medicine (AASM) shows that staging of N1 is associated with the highest inter-rater scoring uncertainty of all PSG stages[62]. Furthermore, N1 sleep has proven the most difficult PSG stage to classify from fMRI FC information in machine-learning studies[18,19]. Addressing the microstructure of N1, a line of evoked response potential-studies have demonstrated a high degree of variability in the cortical processing of external stimuli during early NREM sleep (for reviews, see ref. [63,64]). Phenomenologically, the sleep onset period is known to be rather complex, with varying mental content and responsiveness to sensory stimuli[64,65], and authors have long argued against the assumed homogeneity found in PSG definitions of N1 sleep, an opposition exemplified by Hori's proposal of nine stages of early sleep[45]. If PSG-defined N1 does in fact represent a mix of wakefulness and sleep, this would explain why we found the highest range of whole-brain states during this PSG stage. While this primarily serves to underline the common

notion that N1 is unlikely to be a reliable demarcation between wakefulness and sleep, the fact that the data-driven HMM was able to identify a transition module occurring between wakefulness and consolidated sleep (N2 and N3)—represented by whole-brain states characterised by subcortico-cortical decoupling consistent with intracortical evidence of brain activity during sleep onset[41]—suggests that an improved and principled categorisation of early sleep could be within reach.

PSG does not differentiate between brain activity prior to and after sleep onset. However, in line with the common subjective experience of grogginess when waking from sleep, behavioural experiments have shown cognitive deficits in the period following awakening. The term sleep inertia is often used to describe this phenomenon[66]. Our results confirm that falling asleep and waking up are two asymmetric processes, leading to two separated transition modules of whole-brain network states during wakefulness, with one more likely to occur after consolidated sleep. Like the N1-related findings discussed above, this too serves as a prime example of how information-rich neuroimaging data, when treated in a data-driven way, can be carefully evaluated in light of established knowledge (PSG in this case) to make new

discoveries from, and categorisations of, brain activity. The presented findings point ahead to a research agenda making hypothesis-driven assessments of how the alternative, data-driven, temporal segmentations and dynamics of whole-brain networks across the NREM sleep cycle relate to sleep-behaviour and cognition, when the latter is measured independently of PSG (see Supplementary Discussion 3). Furthermore, there is scope for HMM explorations with higher temporal detail using electrophysiological modalities, such as magnetoencephalography (MEG) and high-density EEG (see Supplementary Discussion 3).

For further discussion about the reproducibility of our results across different numbers of states, different initialisations of the HMM, and different parcellation schemes, as well as discussion about our choice of data inclusion, pre-processing steps, such as spatial smoothing, and the use of the RETROICOR method to remove physiological signals from the fMRI data, we refer to the Supplementary Discussion 2, Supplementary Notes 1–3, and Supplementary Figures 1, 6–16, and 22–25.

In summary, the work presented here demonstrates how data-driven, temporally sensitive analyses of large-scale fMRI brain activity can be used to explore fundamental changes in behaviour and cognition, in the form of the wake-NREM sleep cycle. The results reveal a higher complexity of brain activity than what traditional sleep scoring—and neuroimaging relying strictly on PSG—can reveal. We projected the traditional stages of wakefulness and NREM sleep onto a probabilistic map of transitions between whole-brain network states. By studying these transitions we have shown a significant decrease in whole-brain dynamics during consolidated stages of NREM sleep; that brain activity prior to sleep is significantly different from just after sleep; that whole-brain network activity do not support traditional criteria to define N1 sleep; and that increased activity in the DMN might serve a gate-function for the entry into NREM sleep. By using fMRI data we have increased the spatiotemporal resolution of traditional NREM sleep stages, using a framework that should be sought expanded to include other fundamental changes in brain activity, such as REM sleep, sleep disorders, anaesthesia and psychedelic experiences. Finally, future work should aim to leverage even finer temporal details through modalities such as MEG and high-density EEG.

## Methods

**Acquisition and processing of fMRI and PSG data.** fMRI was acquired on a 3 T system (Siemens Trio, Erlangen, Germany) with the following settings: 1505 volumes of T2*-weighted echo planar images with a repetition time (TR) of 2.08 s, and an echo time of 30 ms; matrix $64 \times 64$, voxel size $3 \times 3 \times 2$ mm$^3$, distance factor 50%, FOV 192 mm$^2$.

The EPI data were realigned, normalised to MNI space, and spatially smoothed using a Gaussian kernel of 8 mm$^3$ FWHM in SPM8 (http://www.fil.ion.ucl.ac.uk/spm/). Spatial downsampling was then performed to a $4 \times 4 \times 4$ mm resolution. From the simultaneously recorded ECG and respiration, cardiac- and respiratory-induced noise components were estimated using the RETROICOR method[67], and together with motion parameters these were regressed out of the signals. The data were temporally band-pass filtered in the range 0.01–0.1 Hz using a sixth-order Butterworth filter. Please note that the fMRI data were temporally filtered with no consideration of the later established PSG structure of the data. Hence, our findings of relative differences between the various PSG stages should not be affected by these pre-processing steps. We show that this is the case for the temporal filter in Supplementary Figure 16, where the plots of Fig. 3 from the main text have been re-computed for an HMM with 19 states on BOLD data, which had not been temporally filtered.

Simultaneous PSG was performed through the recording of EEG, EMG, ECG, EOG, pulse oximetry and respiration. EEG was recorded using a cap (modified BrainCapMR, Easycap, Herrsching, Germany) with 30 channels, of which the FCz electrode was used as reference. The sampling rate of the EEG was 5 kHz, and a low-pass filter was applied at 250 Hz. MRI and pulse artefact correction were applied based on the average artefact subtraction method[68] in Vision Analyzer2 (Brain Products, Germany). EMG was collected with chin and tibial derivations, and as the ECG and EOG recorded bipolarly at a sampling rate of 5 kHz with a low-pass filter at 1 kHz. Pulse oximetry was collected using the Trio scanner, and respiration with MR-compatible devices (BrainAmp MR+, BrainAmp ExG; Brain Products, Gilching, Germany).

Participants were instructed to lie still in the scanner with their eyes closed and relax. Sleep classification was performed by a sleep expert based on the EEG recordings in accordance with the AASM criteria (2007).

Results using the same data and the same preprocessing have previously been reported in ref. [18].

**Participants.** We used fMRI and PSG data from 57 participants taken from a larger data-base[18]. Exclusion criteria focussed on the quality of the concomitant acquisition of EEG, EMG, fMRI and physiological recordings. Written informed consent was obtained, and the study was approved by the ethics committee of the Faculty of Medicine at the Goethe University of Frankfurt, Germany.

Following the HMM decomposition, two different subsets of the solution were used for post hoc evaluation of the HMM. The first corresponded to the 18 participants that reached all four stages of PSG, and the second corresponded to the 31 participants that woke up after having reached consolidated sleep (the WASO group).

**HMM general overview.** In order to resolve dynamic whole-brain networks in the fMRI signals in a data-driven way, we applied a HMM[24,27] to timecourses extracted from 90 ROIs defined by the cortical and subcortical areas of the AAL[26], however, please see Supplementary Note 1 for a demonstration of the robustness of our results using an alternative parcellation.

We used the FSL (https://fsl.fmrib.ox.ac.uk/fsl/fslwiki/) function fslmeants to average over voxels within each ROI to get the representative timecourses. The participant-specific sets of 90 ROI-timecourses were demeaned, divided by their standard deviation, and concatenated across participants, yielding a data matrix of dimensions $90 \times (57 \times 1500)$, with 1500 samples corresponding to 52 min given a TR of 2.08 s. The HMM inference estimated a number of recurring discrete states, each of which was characterised by a unique configuration of data statistics. We employed a Gaussian HMM, implemented using the Matlab toolbox HMM-MAR (https://github.com/OHBA-analysis/HMM-MAR), such that each state was modelled as a multivariate normal distribution with first (mean activity) and second order statistics (covariance matrix). The parameters of the states were defined at the group level, whereas the state timecourses are defined for each subject separately. Therefore, the HMM inference identified periods of time of quasi-stationary activity, where the 90 ROI timecourses could be described by certain configurations of mean activity and FC. The HMM represents a tool for decomposing multivariate data into fewer dimensions. Given the high spatial dimensionality of fMRI, it is common to use principal component analysis (PCA) to reduce the number of parameters to be estimated in the decomposition, increasing the signal-to-noise ratio of the data and improving the robustness of the results[24,27]. Accordingly, we submitted the demeaned, standardised and concatenated BOLD timecourses to PCA prior to the HMM inference. Keeping approximately 90% of the signal variance, we used the top 25 principal components (see Fig. 1), yielding a data matrix of dimensions $25 \times (57 \times 1500)$, which were then fed to the HMM. An overview of the analysis workflow is given in Fig. 1 of the main text. For certain analyses, such as the MANOVA and the test for WASO-specific HMM states, we used subsets of the full set of HMM states.

**Choice of number of HMM states.** The HMM was implemented with variational Bayes inference, which was used to probabilistically estimate the state statistics and transition probabilities[24,27]. The number of states of the HMM was a free parameter, which had to be chosen before further evaluation. Determining the number of states present in recordings of spontaneous brain activity is a non-trivial task, which may be approached in a number of ways. We ran the HMM for model orders spanning 4–45, and evaluated each solution by a number of summary statistics, the most important of which are plotted in Supplementary Figure 1.

Supplementary Figure 1a shows the minimum free energy as a function of the HMM model order. The free energy is the statistical measure that is minimised during the (variational inference) Bayesian optimisation process. Technically speaking, it is an approximation of the model evidence, and includes two terms: how well the model fits the data, and the complexity of the model (measured as how much it departs from the prior distribution). Whereas the free energy is a reasonable criterion for choosing the ideal number of states for the HMM, its biological validity remains unclear in so far as the HMM does not represent a biophysical model. As apparent from the plot in Supplementary Figure 1a, the minimum free energy was monotonically decreasing over the large range of tested numbers of states, showing no negative peaks. Hence, like in previous applications of the HMM[24,27], the free energy was not informative for choosing the number of states in our case.

We defined fractional occupancy as the temporal proportion of a recording, in which an HMM state was active[27]. In Supplementary Figure 1c is plotted the development of the median fractional occupancy across HMM states as a function of model order. While the curve decreases rapidly for low values of $K$, meaning that, as expected, each HMM state on average accounted for less of the total recording time as the number of states was increased, this trend ceased from around $K = 19$. This stagnation for higher model orders was caused by the occurrence of 'sporadic' states, which modelled very (participant-) specific subparts of the data (see Supplementary Figure 6). This phenomenon was also reflected in the development of the average HMM state lifetime, which too stabilised around the same value of $K$, as shown in Supplementary Figure 1d.

To test whether the fMRI-based HMM states showed a significant relationship with the EEG-based sleep scoring, we used a multivariate analysis of variance (MANOVA). The built-in MATLAB function manova1 provided the summary statistic Wilk's $\Lambda$, which described how well the $K$ number of HMM state timecourses could be grouped according to the sleep scoring. In order to test if the relationship was significant, we performed MANOVA's on 1000 permuted cases of the sleep scoring, collecting the Wilk's $\Lambda$ for each run. Each permutation was constructed, such that the original stage transition points, stage counts, and periods were preserved, while the sleep-stage labelling of each of these periods was shuffled randomly within participants. For each permutation, each participant thus retained the same sleep stages, but the temporal orderings of these were random. Supplementary Figure 1b plots Wilk's $\Lambda$ as a function of the HMM model order for both the original sleep scoring and the permuted cases. For number of states above $K = 7$, the HMM state timecourses were grouped significantly better by the original compared to the permuted sleep scoring. This result suggests that only when using more than seven states, the HMM identified states with a significant dependency on the PSG scoring.

The results in Supplementary Figure 1b–d are computed from a subset of the HMM solutions corresponding to the participants including all four available PSG stages (18 participants with: wakefulness, N1 sleep, N2 sleep, and N3, see right part of Supplementary Table 1). This was done to minimise the unevenness in the representation of PSG stages. Based on the evaluations above, we concluded that the HMM was able to, in data-driven fashion, estimate the temporal structure given by the EEG-based sleep scoring for model orders above 7. We chose $K = 19$ states, because increasing the number of states above this point mainly resulted in the addition of HMM states of low-fractional occupancy and participant-specific occurrences.

**Significance testing**. In order to evaluate the PSG-sensitivity and -specificity, we used paired $t$ tests to test for significant differences within the HMM states. As such, with the original PSG stages, we compared the sensitivity and the specificity, respectively, for W, N1, N2 and N3 within each HMM state for the 18 participants that included all of these four stages. This yielded six comparisons within each HMM state (W-vs.-N1, W-vs.-N2, W-vs.-N3, N1-vs.-N2, N1-vs.-N3, N2-vs.-N3), for both the sensitivity and specificity measures, and each of these comparisons was associated with a $t$-statistic. To test if these $t$-statistics were larger than random, we used permutation testing as explained above (see section Choice of number of HMM states), where the EEG-based sleep scoring vector was permuted 1000 times with number of PSG stages and periods kept constant, but with their temporal order randomly shuffled within each participant. We computed the PSG-sensitivity and -specificity for each permutation and performed paired $t$ tests for every case. The resulting $t$-statistics were used to build null-distributions, and the original $t$-statistics were compared against these to get $p$ values for the original tests. The bar plots of PSG-sensitivity and -specificity in Fig. 3a, b of the main text include line crossbars indicating the cases with $p$ values < 0.01 (paired $t$ tests with permutations). Please note that the use of paired $t$ tests ensured that the identified differences were consistent within participants, and not merely as a group effect.

We evaluated the hypothesis that certain HMM states had higher activity in periods of wakefulness after sleep onset (WASO) by considering the subgroup of participants that, according to PSG scoring, woke up after having reached N2 sleep. This corresponded to 31 participants, and we defined WASO as polysomnographically estimated periods of wakefulness that followed N2 sleep. From here we followed the same steps outlines above for the original PSG stages, and the results are shown in Supplementary Figure 2.

To test for differences in the measures of dynamics, amount of switching and range of HMM states, we also employed paired $t$ tests and the permutation scheme explained above to establish chance levels.

**Analysis and visualisation of HMM transitions**. The matrix of transition probabilities, which were explicitly modelled by the HMM, contained a clear organisation, in which subnetworks of HMM states expressed more frequent transitions within each other than to states outside. In other words, the transition matrix represented a directed graph with modular organisation. We demonstrated this by submitting the transition matrix (shown in Fig. 4a) to a modularity analysis, using Matlab functions from the Brain Connectivity Toolbox (https://sites.google.com/site/bctnet/Home)[69], based on Newman's spectral community detection[70]. Prior to running the modularity algorithm, we excluded the transitions of the HMM states that did not occur consistently across participants, i.e., sporadic states (see Methods, Choice of number of states, and Fig. 4b), and thresholded the remaining transition matrix to include the strongest elements. The choice of this latter threshold was done for visualisation purposes (for the results shown in the main text using 19 states we included the 21% strongest transitions), however, different thresholds resulted in highly similar module partitions. The modular organisation is presented in a reordered matrix (Fig. 4c) and as a map (Fig. 4d).

**Visualising mean activation maps of HMM states**. The mean activation maps have been overlaid on brain surfaces in Figs. 5 and 6 of the main text, using the Human Connectome Project software Connectome Workbench (https://www.humanconnectome.org/software/connectome-workbench). The HMM was inferred in volumetric MNI152 space and mapped to the surface of the Conte-69 template using the Workbench function wb_command –volume-to-surface-mapping. The presented surface maps are shown with the 50% strongest increases and the 50% strongest decreases in activation for each HMM state relative to baseline averaged over all HMM states.

## Data availability

The datasets generated during and/or analysed during as well as code used during the current study are available from the corresponding author on reasonable request.

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

## Acknowledgements

G.D. was supported by the Spanish Research Project PSI2016-75688-P (AEI/FEDER) and by the European Union's Horizon 2020 research and innovation programme under Grant agreement no. 720270 (HBP SGA1). M.L.K. was supported by the ERC Consolidator Grant CAREGIVING (615539) and the Center for Music in the Brain, funded by the Danish National Research Foundation Grant DNRF117. J.C. was supported under the project NORTE-01-0145-FEDER-000023. The authors would like to thank Martin Dietz for his inputs regarding the hemodynamic response function convolution.

## Author contributions

H.L. oversaw data acquisition. E.T., A.B.A.S. and D.V. preprocessed the data. A.B.A.S., M.L. K., D.V., S.F.V.N. and M.W. conceptualised the data analyses. A.B.A.S., D.V., M.L.K. and K.R. analysed the data. A.B.A.S., D.V. and M.L.K. wrote the manuscript. A.B.A.S., D.V., M. L.K., M.W., G.D., E.V.S., P.V., J.C., H.L., E.T., S.F.V.N. and K.R. edited the manuscript.

## Additional information

**Competing interests:** The authors declare no competing interests.

