## [Peer Review File · Nature Communications]

Reviewers' comments:

Reviewer #1 (Remarks to the Author):

In this interesting work, Stevner and colleagues reanalyzed previously acquired EEG/fMRI data of wakefulness and NREM sleep. They applied a hidden Markov model (HMM) to the fMRI data, which incorporates both activity and connectivity information in a temporally sensitive manner. The fMRI data were extracted with a predefined anatomical atlas and the amount of HMM states was set to 19. A hierarchical grouping of the 19 HMM states showed a good overlap with wakefulness and the NREM stages. The authors reported relevant characteristics for these states (e.g., sensitivity, specificity, switching) and provided an intriguing matrix representing the likelihood of transitioning from one state to another. The authors conclude they discovered a significant decrease in whole brain dynamics during consolidated stages of NREM sleep, that fMRI activity/connectivity doesn't map well onto N1 and that the default mode network (DMN) could serve a "gate-function" for the entry into NREM sleep, among others. The paper makes for an interesting read on a timely topic, but some issues need addressing, particularly on the stability of the analyses and to what extent the findings go beyond our current understanding of sleep.

Introduction

- The rationale for applying HMM to these fMRI data needs more explanation. Why this model and not multivariate autoregressive or other kinds of models? Could the authors discuss the conditions for this model, and if and how these differ from other Markov based processes regarding long range temporal dependence? This should preferably be related to the authors' own work on non-Markovianity in resting-state EEG and a reduction of long range temporal dependence in sleep in various fMRI networks in a previous analysis of these data (Tagliazucchi et al., 2013; von Wegner et al., 2017).

Results

- The number of states was set to 19, which was not due to some formal criterion like minimal free energy. Yet the authors' line of reasoning can be followed, although it doesn't fully tackle the relative arbitrariness of the chosen solution. If we go with this setting, some of the states turn out to have very similar anatomical maps. For instance, the states 4 and 15 that were more N1-related seem to be the inverted map of each other, with some minor differences. If the analysis would have been set to a lower number, would these states still be recognized as different states or would they morph into the same state? Or in other words, are these states different enough to be labelled different or is that a consequence of the setting? This could also be relevant for some of the other findings, e.g. the exclusivity of transitioning from state 8 to 15 at sleep onset – is this still visible with a higher numbers of states for the analyses?

- The triangulation pattern observed among two N2 states and the one N3 state is due to states 3 and 6 in N2, with state 6 showing some overlap with sleep spindle related activity and connectivity as in previous work, which is also noted by the authors in the discussion. This implies that the N3 state can be preceded from N2 sleep with and without spindles, which can be seen in visually scored polysomnographic recordings. This now leads to the possibility that the HMM findings still represent the neural correlates of classical EEG states and events throughout sleep. Since the authors claim that their analysis goes beyond PSG, they have to show that the states do not merely reflect specific EEG events and states. The authors already state that "future work should address more specifically the relationship between data-driven, HMM-identified networks and EEG-defined spindles as well as other graphoelements such as K-complexes", but they can and should do this themselves since it is central to their claim that their analysis expands the classical understanding of sleep. This applies not just to spindles or K-complexes, but also to N1 phenomena (reduction in posterior alpha) or EEG-based vigilance fluctuations within wakefulness (e.g., Olbrich et al., 2009). Also, the authors need to discuss whether the move beyond PSG

staging cannot be done by EEG itself, since it allows a much more fine-grained temporal dissection even though the PSG stage labelling itself is rather analogue.

Discussion/Conclusions

- In the title the authors and throughout the paper the authors claim the "discovery of key whole-brain transitions and dynamics underlying the human sleep cycle". But without REM sleep, they are studying the NREM part of the sleep cycle.

- One main conclusion is that N1 sleep did not correspond to any single state or a group of states identified by the HMM, which is in line with the notion that N1 is not a consolidated sleep stage but a transition stage between wakefulness and sleep. Note that this does not go against the current consensus in sleep research. For instance in the textbook for sleep, the Principles and Practice of Sleep Medicine (5th ed), Carskadon & Dement write that sleep stage 1 is a "wake-to-sleep transition" stage, and that it "occurs as a transitional stage throughout the night" (Carskadon & Dement, 2011). Furthermore, there is a whole line of research on ERPs in stage 1 sleep, which is in line with the notion that sleep stage 1 is a transition stage and not a straightforward sleep stage. The authors should incorporate such previous work and common notions on N1.

- Do the authors have any data on mentations the subjects reported at awakening? If not, the authors may want to attenuate their speculations on mental content and the dynamic state regime in N1.

- Another main conclusion focuses on the observed differences between pre- to post-sleep wakefulness, about which the authors write that PSG does not differentiate between both and the "neural underpinnings remain scarcely explored". Here it would be helpful to cite the early EEG/PET work on pre- to post-sleep wakefulness differences (e.g., Braun et al., 1997); a careful comparison with this work would be informative.

Methods

- The images were smoothed early in the process. It is not exactly clear from which images the regional data were extracted, this needs to be clarified. I presume for now they used the error images from the analyses in which noise components and motion parameters were "regressed out of the signals", and that these error images were then temporally filtered. This would mean that the regional data were extracted from images that were smoothed with an 8 mm FWHM kernel, which creates artificial dependencies between the regional time-courses. If that is the case, please re-run the analysis on the images that were not smoothed.

- The stability of the analyses could be strengthened by using different atlases, for instance based on functional connectivity, and applying the same pipeline. It would be useful to know if this again leads to 19 states, if the same transition matrix can be observed and if they are anatomically similar to the current states.

- In the fMRI paragraph, the 2nd sentence "MRI and pulse artefact correction were applied on the average artefact subtraction method ..." describes a method used to clean the EEG.

References

- Braun, A. R., Balkin, T. J., Wesenten, N. J., et al. (1997). Regional cerebral blood flow throughout the sleep-wake cycle. An h2(15)o pet study. *Brain*, 120 1173-1197.
- Carskadon, M. A., & Dement, W. C. (2011). Chapter 2 - normal human sleep: An overview. In *Principles and practice of sleep medicine (fifth edition)* (pp. 16-26). Philadelphia: W.B. Saunders.
- Olbrich, S., Mulert, C., Karch, S., et al. (2009). Eeg-vigilance and bold effect during simultaneous eeg/fmri measurement. *Neuroimage*, 45(2), 319-332.
- Tagliazucchi, E., von Wegner, F., Morzelewski, A., et al. (2013). Breakdown of long-range

temporal dependence in default mode and attention networks during deep sleep. Proc Natl Acad Sci U S A, 110(38), 15419-15424.

von Wegner, F., Tagliazucchi, E., & Laufs, H. (2017). Information-theoretical analysis of resting state eeg microstate sequences - non-markovianity, non-stationarity and periodicities. Neuroimage, 158, 99-111.

Reviewer #2 (Remarks to the Author):

The authors describe the use of a data driven method to study the spatiotemporal structure of previously acquired fMRI data with the goal of characterizing changes in across the sleep-wake cycle. This is an interesting topic as it is a relatively unexplored area, in part due to the sleep adverse conditions of the MRI scanner. The approach is based on a Markov model of the fMRI signal and allows classification of brain activity in a set of prototypical patterns whose occurrence and intensity may vary with arousal state. This potentially allows a better characterization of sleep than previous fMRI studies based on polysomnography (PSG) sleep staging or correlations with EEG events such as K-complexes and spindles. The authors, limiting their analysis to non-REM sleep, identify a number of fMRI activity patterns ("states") that preferentially occur during wake and each of three EEG-defined sleep stages, and quantify transition probabilities between these states. Based on this, they conclude that the results provide a novel and more complete characterization of sleep than conventional PSG-based sleep staging, that the default-mode network serves as a "gate" for the transition into non-REM sleep, and that there is a decrease in "whole brain dynamics during consolidated stages of non-REM sleep".

Overall I believe this is a valuable study presenting a more rigorous way for the analysis of fMRI sleep data than previously presented. The identified spatiotemporal patterns are interesting and may provide more insight into the nature of sleep. I do have a number of reservations though about both the methodology, and the presentation of the results:

- Throughout the manuscript, PSG based sleep staging is being used as straw man to contrast with the authors' method. This is not appropriate, as the purpose of PSG is not to comprehensively characterize sleep and its activity patterns, rather than discriminating between various depths of sleep (arousal thresholds). At the same time, numerous other methods have been used to study sleep, including EEG, MEG, ECoG, and even PET and fMRI (the former 3 distinguishing microstates, the latter 2 exploring e.g. the PET or fMRI correlate of K-complexes and spindles). This literature should be discussed. In addition, it should be mentioned that PSG can be used on single subjects, rather than requiring the large dataset analyzed here.

- The hidden Markov model (HMM) approach to perform data-driven analysis is interesting but by itself does not guarantee the identification of neuro-scientifically relevant patterns of brain activity. Like a number of previously presented approaches (co-activation pattern based analysis, temporal functional modes, point-process analysis etc. etc), it allows extraction of statistically different patterns of fMRI activity but provides little understanding of what these patterns relate to or mean. Specific problems I see with the implementation here is the quite arbitrary choice of the number of states (see minor comments as well), and the use of an unequal distribution of data over the different PSG stages. This could affect the fMRI patterns of the individual states, and the conclusion about what these patterns may mean. Regarding the conclusion that fewer states were found during the deepest PSG sleep stage, this could be simply due to the fact that there was only 10% of deep sleep in the 57 subjects studies. To this point, HMM state 16 that was the only state found for N3 sleep was only specific to N3 in 4 (out of 57!) subjects. Similarly on page 21, the authors say they "provide direct evidence of a higher state repertoire" during wakefulness. Again, much more data was collected during wake, potentially biasing the number of wake vs sleep states. In addition, more states found across subject does not mean more states WITHIN subjects,

while the latter would be the more interesting finding. I would not be surprised if different people were thinking about different things during wake! (see also below).

- The conclusion that the default-mode network serves as a "gate" for the transition into non-REM sleep appears based on the higher transition probability between state 8 (with high DMN activity) and state 15. This seems unwarranted, as there are a significant number of transitions between state 2 (low DMN activity) and state 15 as well.

- The authors comment on the variety of states occurring during each EEG sleep stage but this is a bit misleading as this was a group-based determination. It is possible that some each subject exhibits a smaller repertoire of states. Similarly, talking about a "higher state repertoire" during wake may be valid across subjects, but authors have not demonstrated this to be the case in individual subjects.

- As physiological changes are known to be prominent across sleep (in fact, they are increasingly being used to perform sleep staging), the authors should comments on how much they believe RETROICOR removes systemic physiological effects from the fMRI data, as well as the potential remaining confounds.

- The abstract is vague and somewhat speculative. Terms like "top-down defined sleep stages" and "comprehensive image of brain states" are uninformative. "going beyond PSG, HMM allowed ..." is misleading. PSG is not meant to characterize the spatiotemporal patterns of brain activity, merely provide a surrogate marker of arousal threshold. Also, likely other (non HMM) analysis methods will allow this characterization, so I feel the important new thing here is the data driven aspect, not the specific analysis method used. Lastly, the authors should list here the specific neuro-scientific findings of their work.

Minor comments:

1) Several times the authors mention the study subjects "Across the sleep wake cycle". This is not correct as REM sleep was not analyzed.

2) Page 1, Line 1: Is "underlying" the best word for the title? Perhaps not, but there may be a better way to signal to potential readers that the analysis was performed across the conventional sleep stages. Maybe "Discovery of whole-brain transitions and dynamics across the conventional human sleep stages."

3) Page 1, Line -14: The first phrase of the abstract is true but unnecessary, distantly related to the specific sleep neuroscience under study

4) Page 2, Line -2: Was polysomnography ever indicated in the clinical evaluation of insomnia?

5) It is stated at the bottom of page 2 that PSG is poor for studying insomnia. Are the authors proposing that fMRI is better?

6) Top page 3; are predictions by the listed theories captured by states found by authors?

7) Page 3, Line 12: Are the text and references on brain activity during wakefulness necessary? It may be prudent to delete the text between "wakefulness" and "however"?

8) Page 3, Line 19: The authors exhibit intellectual honesty by including their publications in the list of publications that they, in the next sentence, carefully suggest has a key limitation.

9) Page 3, Line -7: I do not understand why Reference 54 is unique. Does it not have the same limitation of a top-down constraint of the conventional sleep stages as the above references? It seems out of place here. Should it be moved and grouped with those publications at the beginning of the preceding paragraph? The same questions apply to the text associated with it at the initial two points in the Discussion section where it is specifically mentioned.

10) Page 4, Line 1: The different sample sizes used throughout the manuscript is confusing. It would be worthwhile to add some unambiguous text here and in the Participants subsection in the Methods section.

11) Page 4, Line 7: The subsample of the 33 wakefulness-after-sleep-onset subjects should be added to Table S1.

12) Why are some states not represented in Fig.4 (e.g. state14 which seems to have substantial fractional occupancy)

13) Neuro-electrical activity such as K-complexes and spindles are known to be prevalent during sleep stages N1 and N2. Do the authors expect these activities to be represented in specific HMM

states? Please Comment.

14) The color bar in Fig. 5 needs to be labeled

15) Page 18. "Looking through glass, darkly" is awkward language.

16) Page 18: "other studies have taken PSG as ground truth". To some extent, this is what the authors do as well! (e.g. in grouping of states in sleep stages in figs 4, 5,7 and the conclusions that flow from that.

17) Page 6, Line -12: Incorporating Reference 54 in this justification is awkward and unnecessary. This justification stands on its own quite well without the introductory phrase.

18) Page 11, Line 5: Perhaps I missed it, but how was the threshold determined?

19) Page 11, Line -12: The attempt to extend states across the conventional sleep stages is a very novel and important part of the manuscript. The above validation steps are nice, but this is where we will actually learn something new about sleep neuroscience. Indeed, that is exactly what happened when the authors separated wakefulness and wakefulness after sleep onset.

20) Page 11, Line -12: Perhaps Figure S3 should be moved to the main text. The statistically significant sensitivity differences in opposite directions for states 2, 8, and 10 (wakefulness is higher) versus states 5, 17, and 18 (wakefulness after sleep onset is higher) is very compelling.

21) Page 19, Line 1: The authors may want to begin this paragraph by briefly discussing how others have attempted to subdivide the conventional sleep stages with more-sophisticated analyses of EEG data (e.g., Reference 7 and Borbely, Baumann, Brandeis, Strauch, & Lehmann, 1981).

22) Page 19, Line 4: Does Figure 7 add information that is not available in the text?

23) Page 20, Line 9: The authors may want to review a PET study on post-sleep wakefulness (Balkin et al., 2002) and determine whether it is worth including here.

24) Page 20, Line 17: Reference 86 did not examine N1.

25) Page 21, Line 14: How dependent is this interpretation on the number of states initially chosen for the analysis?

26) Page 22, Line -12: The differences between wakefulness and wakefulness after sleep onset should receive greater attention. They are the most novel, interesting, and exciting part of the manuscript. They should be cited as an example of how the approach and other similar approaches can be used to make truly new discoveries in sleep neuroscience.

27) Page 22, Line -12: Related to the previous comment, the authors may want to discuss other attempts to subdivide the conventional sleep stages with fMRI (Picchioni et al., 2008; Watanabe et al., 2014).

28) Page 22, Line -9: I appreciate how the authors make a call for the inclusion of other important brain processes. One that they might consider mentioning are behavioral measures of sleep such as eyelid closure, which has been applied to fMRI data during fluctuations of arousal (Chang et al., 2016), and arousal threshold, which forms the most important component of the original behavioral definition of sleep.

29) Page 22, Line -9: Related to the previous comment, when going beyond the conventional sleep stages, it may be worthwhile to mention that pre- versus postsleep adaptive brain processes related to the function of sleep (e.g., memory consolidation) should also be included.

30) Page 22, Line -3: Is discussing the application to other altered states of consciousness too loosely linked to the current study? Perhaps this sentence should be deleted.

31) Page 26, Line -5: The terms "more robust" and "potential noisy" are vague. It is not clear to the reader why principal component analysis was performed. Why was it performed here and not in the authors' prior publication (Reference 34)?

32) Page 26, Line -2: Figure S1 is not necessary in general and is included as a panel in Figure 1.

33) Page 27, Line 11: More details on what exactly minimum free energy is measuring may be warranted.

34) Page 27, Line 16: Why did minimum free energy fail as the method for choosing the number of states? What is unique about these data that triggered this deviation from standard practice?

35) Page 27, Line -8: The authors used how well the states related to the conventional nonrapid eye movement sleep stages here to help them determine how many states to use. Does this not defeat the purpose of analyzing the fMRI data independently of the conventional sleep stages?

36) Page 29, Line 1: The AASM manual defines wakefulness after sleep onset as wakefulness after

having reached any stage. Why did the authors choose to deviate from this standard?

37) Supplementary Material Page 5, Line 1: The authors should consider simplifying this figure further by also excluding the bars representing N1 and N2. The key question is whether W and WASO differ.

38) Supplementary Material Page 5, Line 1: "Including" may not be the best word. Is "separating" better?

39) The justification for choosing the number of HMM states is not sound. E.g. in caption of Fig. S2 it says: that the "median fractional occupancy stagnates around $K=19$ " to make the point that that higher K did not split states. These issues are not directly related though.

40) I found Figure 7 quite speculative, with several statements that are questionable in the light of my major comments (unequal amount of data in various sleep stages, states present in some subject but not in others, multiple pathways for the transition of wake to sleep etc).

REFERENCES:

- Balkin, T. J., Braun, A. R., Wesensten, N. J., Jeffries, K., Varga, M., Baldwin, P., . . . Herscovitch, P. (2002). The process of awakening: a PET study of regional brain activity patterns mediating the re-establishment of alertness and consciousness. *Brain*, 125, 2308-2319.
- Borbely, A. A., Baumann, F., Brandeis, D., Strauch, I., & Lehmann, D. (1981). Sleep deprivation: effect on sleep stages and EEG power density in man. *Electroencephalography and Clinical Neurophysiology*, 51, 483-495.
- Chang, C., Leopold, D. A., Scholvinck, M. L., Mandelkow, H., Ir, D., Liu, X., . . . Duyn, J. H. (2016). Tracking brain arousal fluctuations with fMRI. *Proceedings of the National Academy of Sciences of the United States of America*, 113, 4518-4523. doi: 10.1073/pnas.1520613113
- Noirhomme, Q., Soddu, A., Lehembre, R., Vanhauzenhuyse, A., Boveroux, P., Boly, M., & Laureys, S. (2010). Brain connectivity in pathological and pharmacological coma. *Frontiers in Systems Neuroscience*, 4, 160. doi: 10.3389/fnsys.2010.00160
- Picchioni, D., Fukunaga, M., Carr, W. S., Braun, A. R., Balkin, T. J., Duyn, J. H., & Horowitz, S. G. (2008). fMRI differences between early and late stage-1 sleep. *Neuroscience Letters*, 441, 81-85. doi: 10.1016/j.neulet.2008.06.010
- Watanabe, T., Kan, S., Koike, T., Misaki, M., Konishi, S., Miyauchi, S., . . . Masuda, N. (2014). Network-dependent modulation of brain activity during sleep. *Neuroimage*, 98, 1-10. doi: 10.1016/j.neuroimage.2014.04.079

Reviewer #3 (Remarks to the Author):

My commits will mainly focus on the HMM part.

When applying Gaussian HMM on fMRI analysis, some papers assume the mean vector is zero and only study the covariance matrix as a functional connectivity matrix. Some papers assume the covariance matrix to be identity matrix and only focus on the mean vector as a mean activation pattern. In this paper, there seem to be no constraints on mean vector or covariance matrix; therefore, the state is represented by a mean activation pattern and a covariance matrix. In this case, substrating/averaging two/multiple covariance matrices from different brain states may be problematic (Figure S5 and S6), because this ignores the fact that these brain states has a different baseline activation pattern.

If we look at the 19 covariance matrices in Figure S4, most part of them is positive. Even for the negative part, the values are relatively low compared with the positive part (-0.2 vs. +1.2). I am

wondering whether this is due to the lack of global signal regression or it is because each of the covariance matrices is associated with a mean vector and that makes it different from the conventional FC matrix.

I suspect the temporal features (e.g. mean life time (around 10-20s), switching frequency) will be highly influenced by the temporal filter (0.01-0.1Hz). It would be nice to prove that at least their relative relationship will not change by using a different temporal filter.

My biggest concern about these results is their reproducibility. As far as I know, methods like HMM are sensitive to its initialization. Training the model twice with different initialization may give you different results. Some states may appear slightly differently and some states may disappear. The temporal features of the model, including life time and transition matrix, may also change. Therefore, which part of the results is actually reproducible (insensitive to initialization)?

In page 5, please use multiplication symbol instead of letter "x" in the dimension of the matrix.

Reviewers' comments:

Reviewer #1 (Remarks to the Author):

In this interesting work, Stevner and colleagues reanalyzed previously acquired EEG/fMRI data of wakefulness and NREM sleep. They applied a hidden Markov model (HMM) to the fMRI data, which incorporates both activity and connectivity information in a temporally sensitive manner. The fMRI data were extracted with a predefined anatomical atlas and the amount of HMM states was set to 19. A hierarchical grouping of the 19 HMM states showed a good overlap with wakefulness and the NREM stages. The authors reported relevant characteristics for these states (e.g., sensitivity, specificity, switching) and provided an intriguing matrix representing the likelihood of transitioning from one state to another. The authors conclude they discovered a significant decrease in whole brain dynamics during consolidated stages of NREM sleep, that fMRI activity/connectivity doesn't map well onto NI and that the default mode network (DMN) could serve a "gate-function" for the entry into NREM sleep, among others. The paper makes for an interesting read on a timely topic, but some issues need addressing, particularly on the stability of the analyses and to what extent the findings go beyond our current understanding of sleep.

We thank the reviewer for the kind words.

Introduction

1 - The rationale for applying HMM to these fMRI data needs more explanation. Why this model and not multivariate autoregressive or other kinds of models? Could the authors discuss the conditions for this model, and if and how these differ from other Markov based processes regarding long range temporal dependence? This should preferably be related to the authors' own work on non-Markovianity in resting-state EEG and a reduction of long range temporal dependence in sleep in various fMRI networks in a previous analysis of these data (Tagliazucchi et al., 2013; von Wegner et al., 2017).

We thank the reviewer for giving us the opportunity to expand on the rationale for using HMM to study sleep. HMM is of course only one of several methods that can be used to decompose multivariate data into temporally defined states. Yet, it is exactly the *data-driven*, temporal segmentation which is the strength of the HMM, that is key to our study, since we wanted to get an unbiased, data-driven estimate of the state changes involved in normal human sleep. This also allowed us to assess how well traditional sleep staging is reflected in the large-scale patterns found in fMRI BOLD activity.

Compared to other methods such as cluster-based approaches, the HMM explicitly models the transition probabilities between states, which in turn gives a clear and concise description of the related state dynamics, which must be fundamental for understanding the sleep cycle. It is true that multivariate autoregressive models can provide a more complex description of the entire data set, rather than a partition into less complex models (which, in the case of the HMM, are state probability distributions). However, this may in turn make the comparison to traditional sleep staging less straightforward and render results more difficult to interpret. One could also have used sliding window-approaches to dynamic FC, but these are likely to perform worse given their lower temporal resolution and inability to access the fastest scales, as well as statistical issues (see

(Hindriks et al., 2015)). The HMM, on the contrary, provides high temporal resolution (up to the limitations inherent to the BOLD signal).

We have updated the revised ms to include this clarification.

Page 4, paragraph 3:

“Compared to other methods for extracting dynamic FC (Preti et al., 2016), the HMM framework explicitly models the transition probabilities between its inferred states. We show that this information can be used to discover new whole-brain aspects of sleep, complementing the traditional segmentation of brain activity offered by PSG.”

Page 26, paragraph 2:

“It should be noted that the HMM framework was chosen over other methods for extracting dynamic states from multivariate neuroimaging datasets, such as sliding-window clustering (Allen et al., 2014, Hindriks et al., 2015, Haimovici et al., 2017), point-process analysis (Tagliazucchi et al., 2012a), co-activation pattern analysis (Liu and Duyn, 2013, Karahanoğlu and Ville, 2015) (for reviews, see (Calhoun et al., 2014, Preti et al., 2016)). The HMM was particularly suitable for our purpose by virtue of its explicit modelling of temporal dynamics, resulting in states that repeat in a predictable way. Although the HMM is not a mechanistic model of brain activity (a limitation shared with the approaches mentioned above) we have shown how the explicitly modelled HMM transition matrix was fundamental to suggest new partitions of dynamic whole-brain states, which future mechanistic frameworks of NREM sleep and wakefulness should take into account.”

With regards to the choice of the specific settings of the HMM, we chose to use the Gaussian distribution because 1) it has been validated in previous work on fMRI (Vidaurre et al., 2017b), 2) it provides a description of both the mean activation and functional connectivity which were both the target of our analysis, and 3) it links more straightforwardly than other state distributions to standard fMRI analysis of BOLD activation and functional connectivity.

The link to the reported non-Markovianity in resting-state EEG by von Wegner et al. 2017 and the potential link to a reduction of long-range temporal dependence in sleep in various fMRI networks across the NREM sleep stages found in Tagliazucchi et al. 2013 are very relevant. We have updated the revised ms to include a discussion.

Page 26, paragraph 3:

“There is growing evidence that neuroimaging timecourses contain long-range temporal dependencies (Maxim et al., 2005, He, 2011, Ciuciu et al., 2012), i.e. they are non-Markovian (von Wegner et al., 2017). The HMM used here follows the Markovian assumption in the sense that the probability of a state transition at a given time point depends only on the state that is active at the preceding time point, and hence it does not model long-range temporal dependencies *parametrically*. Importantly, however, it does not preclude them either. That means that the HMM state time courses can exhibit non-Markovian dynamics and long-term dependencies; see e.g. (Vidaurre et al., 2017b). Notably, our finding of HMM states grouping into modules of transitions represents an analysis that goes beyond Markovianity, and demonstrates non-Markovian dynamics (i.e. long-term dependencies) at the system level of the HMM states. In light of this, our finding that N3 sleep was modelled almost exclusively by a single HMM state, while several states grouped into

modules during wakefulness, is in line with the study by Tagliazucchi and colleagues, showing that long-range temporal dependencies in fMRI signals decreases from wakefulness to N3 sleep (Tagliazucchi et al., 2013b).”.

Results

2 - The number of states was set to 19, which was not due to some formal criterion like minimal free energy. Yet the authors' line of reasoning can be followed, although it doesn't fully tackle the relative arbitrariness of the chosen solution. If we go with this setting, some of the states turn out to have very similar anatomical maps. For instance, the states 4 and 15 that were more N1-related seem to be the inverted map of each other, with some minor differences. If the analysis would have been set to a lower number, would these states still be recognized as different states or would they morph into the same state? Or in other words, are these states different enough to be labelled different or is that a consequence of the setting? This could also be relevant for some of the other findings, e.g. the exclusivity of transitioning from state 8 to 15 at sleep onset – is this still visible with a higher numbers of states for the analyses?

We thank the reviewer for these comments. The lack of a formal criterion for determining the dimensionality of our analysis is a limitation shared with all known decompositions of continuous neuroimaging data. Whereas the free energy is a reasonable criterion, its biological validity is unclear in so far as the HMM is not a biophysical model. However we agree with the reviewer that it is fundamental to demonstrate the robustness of our findings across different numbers of states. We are now showing the full results of using 15, 17, 21 and 23 states in 8 new figures in the supplementary material of the revised ms. Irrespective of the number of states chosen the results are very similar in terms of the main messages of the paper. This includes HMM states expressing sensitivity and specificity to the stages in the sleep cycle (Figures S7A-B to S10A-B). Similarly, the differences of HMM switching and range dynamics (Figures S7D-E to S10D-E) across sleep stages remain highly significant.

We point the reader to these analyses in the Discussion:

Page 26, paragraph 1:

“Acknowledging the potential limitation of manually choosing the number of states, we have reproduced the main result figures for different number of states ($K = [15, 17, 21, 23]$) in the Supplementary Materials (see *Choice of number of HMM states* and Figure S8 to S15). These figures demonstrate the robustness of our main conclusions.”

And we provide a summary of these supplementary findings in the Methods:

Page 33, paragraph 2:

“In appreciation of the potential limitations related to choosing the number of HMM states with no strict, formal criterion, we include the results of using different numbers of HMM states. In Figures S7 to S10 we have reproduced Figure 3 of the main text with HMM results using 15, 17, 21, and 23 states respectively. Demonstrating the robustness of our HMM findings, the conclusions of the main text using 19 states are also found in Figures S7 to S10. Specifically Figures S7A-B to S10A-B show how select HMM states expressed high sensitivity and specificity for different PSG stages. In line with the results for $K = 19$ states, the HMM with lower and higher K identified states with high

sensitivity and specificity for wakefulness, N2, and N3 sleep, but not for N1 sleep. Figures S7D-E to S10D-E quantify the dynamics of HMM states within PSG stages. The relative differences between PSG stages are conserved and highly stable across numbers of HMM states. Interestingly, the absolute values of switching between and range of HMM states within PSG stages were in fact also quite preserved across numbers of HMM states. This is likely caused by the fact that the main effect of changing the number of HMM states is an addition of non-recurring, ‘sporadic’, states that modelled very (participant-) specific periods of the fMRI data (see above).”

The question regarding the potential inversion of the mainly N1-related states 4 and 15 is interesting. However, from a methods point of view, opposite polarities are represented by completely different Gaussian distributions (i.e. having significantly different mean activities), and hence will not morph at lower numbers of states.

Further motivated by the reviewer’s comment, we have made an effort to find a principled way of exploring the transition dynamics of the HMM. This was needed to compare the results related to the HMM transition map when using different numbers of states. In the revised ms we now consider the transition matrix of the HMM as a directed graph, which we submit to a modularity analysis, allowing us to extract sub-groups of HMM states with strong transitions within them compared to transitions to other sub-groups. As shown in the revised Figures 4-7, this new illustration method yields the same overall structure shown in the original ms, yet reveals more of the intricate dynamics and in particular provides a principled way to group HMM states into modules (transition modules in Figure 4C and D). These modules lend important support to our interpretations of the HMM states’ relationship to PSG stages. Furthermore we can use this principled grouping of HMM states to compare transition maps across HMM solutions with different numbers of states as shown for $K = 15, 17, 21, 23$ in the new Figure S11 to S14 in the Supplementary Material of the revised ms. Importantly, we are able to show that the principles of the HMM transition map when using 19 states, as presented in the main text, translate to HMM solution with different numbers of states. The identified transition modules and the discussion they facilitate, are therefore not contingent on the chosen number of states for the HMM.

We explain this method in the Results section of the revised ms:

Page 12, paragraph 2:

“The whole-brain network states organised into a transition map. This is presented in Figure 4, where the 19×19 transition matrix, as returned by the HMM (see Figure 4A), has been submitted to a modularity analysis (see Methods). By considering the most frequent transitions between the HMM states that were consistent across participants (see Figure 4B), the thresholded transition matrix organised into four partitions or *transition modules* (see Figure 4C, and Methods). When these most consistent transitions are presented as a transition map, and each whole-brain network state is represented by a circle plot indicating its specificity for each of the four PSG stages, it is clear that these modules represent a meaningful organisation of the HMM states (see Figure 4D).”

And in the Methods:

Page 34, paragraph 3:

“The matrix of transition probabilities, which were explicitly modelled by the HMM, contained a clear organisation, in which sub-networks of HMM states expressed more frequent transitions within each other than to states outside. In other words, the transition matrix represented a directed graph with modular organisation. We demonstrated this by submitting the transition matrix (shown in Figure 4A) to a modularity analysis, using Matlab functions from the Brain Connectivity Toolbox (<https://sites.google.com/site/bctnet/Home>)(Rubinov and Sporns, 2010), based on Newman’s spectral community detection (Leicht and Newman, 2008). Prior to running the modularity algorithm, we excluded the transitions of the HMM states that did not occur consistently across participants, i.e. sporadic states (see Methods, Choice of number of states, and Figure 4B), and thresholded the remaining transition matrix to include the strongest elements. The choice of this latter threshold was done for visualisation purposes (for the results shown in the main text using 19 states we included the 21% strongest transitions), however different thresholds resulted in highly similar module partitions. The modular organisation is presented in a reordered matrix (Figure 4C) and as a map (Figure 4D).”

And we summarise these additional analyses in the Methods, explaining how HMM across different number of states group into consistent modules:

Page 33, paragraph 3:

“Another main result of this study is presented in the transition map of the HMM states (see Figure 4 of the main text). Again we have re-produced equivalent figures for $K = 15, 17, 21,$ and 23 HMM states (Figures S11 to S14). Our modularity analysis (see below) of the resulting transition matrices illustrates how the four modules from the HMM with 19 states can be identified in the solutions with different numbers of states. This was true for $K = 17, 21,$ and 23 . For $K = 15$, the white and blue modules appear to have merged together. The overall structure of the transition map was therefore robust across the chosen numbers of HMM states. A separate transition module for wakefulness after sleep onset (WASO) was found consistently across all of these values of K , while the intercalated module between wakefulness and consolidated sleep (N2/N3) were found for all but one value of K ($K = 15$).

Whereas these overall configurations of the HMM transitions were found robust to the chosen number of states, the more fine-grained details of the transition map appeared more variable. The gateway-like quality of a DMN-like configuration of brain activity was thus particularly clear for the originally chosen 19 states.”

While these additional analyses demonstrate the robustness of the majority of our findings, it is also clear, as described in the excerpt above, that our claim of exclusive transitioning from state 8 to 15 at sleep onset has to be tempered when using different number HMM states. Our new method still clearly shows the importance of transitioning from a DMN-like configuration in wakefulness (state 8) to a sleep transition state (state 15) for 19 HMM states, and across numbers of states there are clearly gateway states that govern the sleep onset process, but the exact transitions that this is operated through are less obvious.

3 - The triangulation pattern observed among two N2 states and the one N3 state is due to states 3 and 6 in N2, with state 6 showing some overlap with sleep spindle related activity and connectivity as in previous work, which is also noted by the authors in the discussion. This implies that the N3

state can be preceded from N2 sleep with and without spindles, which can be seen in visually scored polysomnographic recordings. This now leads to the possibility that the HMM findings still represent the neural correlates of classical EEG states and events throughout sleep. Since the authors claim that their analysis goes beyond PSG, they have to show that the states do not merely reflect specific EEG events and states. The authors already state that "future work should address more specifically the relationship between data-driven, HMM-identified networks and EEG-defined spindles as well as other graphoelements such as K-complexes", but they can and should do this themselves since it is central to their claim that their analysis expands the classical understanding of sleep. this also relates to other PSG N1 phenomena (reduction in posterior alpha) or EEG-based vigilance fluctuations within wakefulness (e.g., Olbrich et al., 2009).

The reviewer's point hinges on the relationship between our *data-drivenly* identified fMRI networks and more traditional *partly subjective and manually scored* PSG and its features, such as K-complexes and sleep spindles. In the revised version of the ms, we have stressed that the results are not taking sides for or against PSG. Rather, we show that our HMM results are complementing those of PSG. When the HMM results match those obtained via PSG/AASM-based scoring, this supports the reliability of either approach while the HMM adds additional spatial information. In other cases, where the HMM appears to agree less with PSG (such as further characterising N1), these findings offer a novel perspective and potential for improving sleep categorisation in the future. We have made this more clear in the revised ms:

Page 4, paragraph 2:

“Rather than constraining analyses by traditional descriptions of sleep stages, we propose to use novel data-driven analysis methods to elucidate whole-brain networks that can complement and potentially expand the classical understanding of sleep.”

Page 11, paragraph 5:

“Up until this point, we have used the traditional PSG stages to organise and evaluate the temporally resolved whole-brain network states, in terms of the MANOVA results, PSG-sensitivity, PSG-specificity, and dynamics within PSG stages. Yet, the data-driven nature of the HMM also allowed us to reverse the inference, and consider the results, in particular the temporal progression of HMM states, in their own right. This way we were able to ask if the high-resolution, fMRI-based HMM, was able to reveal new aspects of the wake-NREM sleep cycle, hidden from the EEG-based PSG. For this purpose it was informative to examine the transition probabilities of the HMM states. Specifically, we could use these to extract modules of HMM states that transitioned more often between each other than to other states. While agreeing to a certain extent with the PSG staging, the organisation of transition modules and their involved whole-brain network states also demonstrate how this data-driven approach can be used to more fully describe the spatiotemporal complexity of large-scale brain activity across the NREM sleep cycle.”

Page 21, paragraph 2:

“Below we begin by linking our HMM results to existing neuroimaging evidence of PSG stages. This is used to characterise the ways in which the HMM agreed with PSG and establish that the HMM generally inferred meaningful features from the fMRI data. It is on the basis of this general agreement with PSG that we may then move on to a close examination of the new insights with relation to sleep that can be gained from the data-driven HMM.”

Page 23, paragraph 2:

“In the aspects where PSG and the HMM decomposition conformed in time, the HMM lends important support to PSG, while still offering new perspectives in terms of transitions and dynamics. In other cases, the HMM suggests new segmentations of large-scale brain activity that cannot be resolved when following traditional PSG staging.”

We have however refrained from correlating our HMM results to specific graphoelements because the relationship between the fMRI-based HMM states and EEG graphoelements is not at all straightforward to assess, and we feel it falls out of the scope of the present work. A proper investigation would for instance have to deal with the mounting evidence distinguishing between local and global occurrences of sleep spindles and K-complexes (Dehghani et al., 2010a, Dehghani et al., 2010b, Andrillon et al., 2011, Nir et al., 2011, Bonjean et al., 2012, Johnson et al., 2012, Mak-McCully et al., 2015, Piantoni et al., 2016a, b). Methodologically, the commonly observed temporal proximity of sleep spindles and K-complexes cannot be teased apart with the limited temporal resolution of the BOLD signal. We are very interested in embracing these challenges in future studies. For properly addressing the relationship between EEG graphoelements and whole-brain HMM states, source-reconstructed MEG will be a much more appropriate basis for an HMM analysis for this particular purpose (Baker et al., 2014, Vidaurre et al., 2016, Vidaurre et al., 2017a). We have made these points in the revised ms as follows:

Page 25, paragraph 4:

“Finally, there is scope for an even more detailed examination of sleep within the HMM framework, given that BOLD data is not the most temporally sensitive modality available. Recently developed methods combining source-reconstructed MEG data with the HMM framework could prove capable of providing an even more fine-grained picture of sleep’s evolution in whole-brain networks, and allow for an examination of microstructural EEG elements of sleep, such as spindles and K-complexes (Baker et al., 2014, Vidaurre et al., 2016, Vidaurre et al., 2017a), as well as EEG-markers of vigilance fluctuations during wakefulness (Olbrich et al., 2009).”

Also, the authors need to discuss whether the move beyond PSG staging cannot be done by EEG itself, since it allows a much more fine-grained temporal dissection even though the PSG stage labelling itself is rather analogue.

We agree that the future of sleep staging is of course not limited to fMRI, and must take into account the relative strengths of EEG and fMRI, as well as MEG, and would again like to refer the reviewer to:

Page 25, paragraph 4:

“Finally, there is scope for an even more detailed examination of sleep within the HMM framework, given that BOLD data is not the most temporally sensitive modality available. Recently developed methods combining source-reconstructed MEG data with the HMM framework could prove capable of providing an even more fine-grained picture of sleep’s evolution in whole-brain networks, and allow for an examination of microstructural EEG elements of sleep, such as spindles and K-complexes (Baker et al., 2014, Vidaurre et al., 2016, Vidaurre et al., 2017a), as well as EEG-markers of vigilance fluctuations during wakefulness (Olbrich et al., 2009).”

Discussion/Conclusions

4 - *In the title the authors and throughout the paper the authors claim the "discovery of key whole-brain transitions and dynamics underlying the human sleep cycle". But without REM sleep, they are studying the NREM part of the sleep cycle.*

We agree and have reworded throughout.

5 - *One main conclusion is that N1 sleep did not correspond to any single state or a group of states identified by the HMM, which is in line with the notion that N1 is not a consolidated sleep stage but a transition stage between wakefulness and sleep. Note that this does not go against the current consensus in sleep research. For instance in the textbook for sleep, the Principles and Practice of Sleep Medicine (5th ed), Carskadon & Dement write that sleep stage 1 is a "wake-to-sleep transition" stage, and that it "occurs as a transitional stage throughout the night" (Carskadon & Dement, 2011). Furthermore, there is a whole line of research on ERPs in stage 1 sleep, which is in line with the notion that sleep stage 1 is a transition stage and not a straightforward sleep stage. The authors should incorporate such previous work and common notions on N1.*

We agree and have incorporated these suggestions in the revised ms:

Page 24, paragraph 2:

“The two main incongruities between the temporal segmentation suggested by the HMM and the PSG scoring concerned N1 sleep and WASO. N1 sleep did not correspond to any single state or any group of states identified by the HMM. This is likely related to the current consensus that PSG-defined N1 does not represent a clear-cut sleep stage (Carskadon and Dement, 2011), but rather an ill-understood mix of wakefulness and sleep. This is supported by several lines of evidence. Compared to N2 and N3 sleep with their well-defined EEG spectral properties, such as K-complexes, spindles, and slow waves, N1 remains the most vaguely defined sleep stage within PSG. A recent report by the American Academy of Sleep Medicine (AASM) shows that staging of N1 is associated with the highest inter-rater scoring uncertainty of all PSG stages (Rosenberg and Van Hout, 2013). Furthermore, N1 sleep has proven the most difficult PSG stage to classify from fMRI FC information in machine-learning studies (Tagliazucchi and Laufs, 2014, Altmann et al., 2016). Addressing the microstructure of N1, a line of evoked response potential-studies have demonstrated a high degree of variability in the cortical processing of external stimuli during early NREM sleep (for reviews, see (Ogilvie, 2001, Colrain and Campbell, 2007)). Phenomenologically, the sleep onset period is known to be rather complex, with varying mental content and responsiveness to sensory stimuli (Ogilvie, 2001, Goupil and Bekinschtein, 2012), and authors have long advocated against the assumed homogeneity found in PSG definitions of N1 sleep, an opposition exemplified by Hori’s proposed nine stages of early sleep (Hori et al., 2001). If PSG-defined N1 does in fact represent a mix of wakefulness and sleep, this would explain why we found the highest range of whole-brain states during this PSG stage. While this primarily serves to underline the common notion that N1 is unlikely to be a reliable demarcation between wakefulness and sleep, our data-driven identification of a transition module of whole-brain states occurring between wakefulness and consolidated sleep (N2 and N3) suggests that an improved and principled categorisation of

early sleep can be achieved, when the high spatiotemporal resolution of neuroimaging is explored in full.”

6 - *Do the authors have any data on mentations the subjects reported at awakening? If not, the authors may want to attenuate their speculations on mental content and the dynamic state regime in N1.*

This is an interesting suggestion, but we do not have the appropriate data. As seen from the excerpt immediately above, we have attenuated our speculations on the relationship between our findings and the mental content of N1. Instead we have updated the *Perspectives* section of the discussion:

Page 25, paragraph 2:

“The presented findings point ahead to a research agenda making hypothesis-driven assessments of how the alternative, data-driven, temporal segmentations and dynamics of whole-brain networks across the NREM sleep cycle relate to sleep behaviour and cognition, when measured independently of PSG. Features identified by the HMM could prove to essential supplements to PSG and other conventional methods when trying to understand phenomena like the subjective perception of sleep (Bonnet and Moore, 1982, Ogilvie and Wilkinson, 1984), mental content during sleep (Nobili et al., 2012, Siclari et al., 2017), such as the hypnagogic or even hallucinogenic character of sleep onset (Hori et al., 1994, Goupil and Bekinschtein, 2012), sleep inertia of the awakening process (Tassi and Muzet, 2000), sleep-dependent processes related to memory and learning (Diekelmann and Born, 2010), and disordered sleep, like insomnia (Wei et al., 2017). Such studies should explore the theoretical potential of applying the current HMM, parameterised on the present sleep data, to identify the presence of the same dynamical whole-brain network states and transition modules in data from different cohorts, potentially even at the individual level. This new data could then be linked to behaviour and cognition through sophisticated measures of arousal, such as eyelid-closure (Chang et al., 2016), sleep mentation (Siclari et al., 2017), post-sleep memory- and learning performance (Stickgold, 2005), and careful clinical examination of sleep disorders (Edinger et al., 2013).”

7 - *Another main conclusion focuses on the observed differences between pre- to post-sleep wakefulness, about which the authors write that PSG does not differentiate between both and the "neural underpinnings remain scarcely explored". Here it would be helpful to cite the early EEG/PET work on pre- to post-sleep wakefulness differences (e.g., Braun et al., 1997); a careful comparison with this work would be informative.*

We have added this reference, and thank the reviewer for reminding us.

Methods

8 - *The images were smoothed early in the process. It is not exactly clear from which images the regional data were extracted, this needs to be clarified. I presume for now they used the error images from the analyses in which noise components and motion parameters were "regressed out of the signals", and that these error images were then temporally filtered. This would mean that the regional data were extracted from images that were smoothed with an 8 mm FWHM kernel, which*

creates artificial dependencies between the regional time-courses. If that is the case, please re-run the analysis on the images that were not smoothed.

We thank the reviewer for this comment. As stated in the Methods, we chose to use the exact same preprocessing of the EPI data as in the previous studies on this dataset by Tagliazucchi and Laufs (e.g. (Tagliazucchi and Laufs, 2014)) for ease of comparison. While the settings used here for smoothing, i.e. FWHM 8mm, has some support in the literature (see e.g. (Mikl et al., 2008)), we agree with the reviewer that there is a risk of creating artificial dependencies between voxel timecourses. However we do not expect these effects to influence our level of analysis for the following reasons: (i) our goal was grand, whole-brain analysis, with limited focus on fine spatial details, (ii) the HMM is in itself a form of dimensionality reduction, and therefore, when run at the whole-brain level, it is less sensitive to local variability (iii) smoothing will primarily create stationary dependencies (i.e. stationary over time), whereas the HMM specialised in finding relative changes within the time series, (iv) the AAL parcellation consists of ROIs that are large compared to the smoothing parameter, and (v) spatial smoothing was performed with no consideration of the PSG structure of the data. Hence, our findings of relative differences between the various PSG stages should not be much affected by smoothing. For these reasons, and also because of the already very large number of figures in the present ms, we feel that the comparison between different levels of smoothing (or no smoothing at all) should be subject of future work, where we will analyse specific local features driving sleep transition.

9 - The stability of the analyses could be strengthened by using different atlases, for instance based on functional connectivity, and applying the same pipeline. It would be useful to know if this again leads to 19 states, if the same transition matrix can be observed and if they are anatomically similar to the current states.

In their recent review of the literature on the topographic organisation of the brain, Eickhoff and colleagues conclude that it is currently unclear which is the best spatial parcellation of the human brain (Eickhoff et al., 2018). We chose to use the AAL parcellation given that it is based on anatomical information and is the most used parcellation in the literature on functional connectivity in sleep (Spoormaker et al., 2010, Spoormaker et al., 2012a, Tagliazucchi et al., 2013a, Tagliazucchi and Laufs, 2014, Altmann et al., 2016, Haimovici et al., 2017) (and wakefulness). As we discuss in the revised ms, using a parcellation based on functional connectivity could be rather problematic given the established variation of FC between wakefulness and non-REM sleep stages (Tagliazucchi and Laufs, 2014, Altmann et al., 2016). In any case, given that our results were robust to different levels of temporal granularity (expressed through different numbers of states), we expect that these would be robust also to different levels of spatial granularity (expressed through different parcellations). A rigorous assessment of this point will be subject of future investigations, as it also relates to important research questions (i.e. the extent to which local activity is informative to sleep compared to grand, whole-brain activity).

Page 30, paragraph 6:

“We chose the AAL over other possible parcellations because it is the most frequently used in previous fMRI studies of FC during NREM sleep (Spoormaker et al., 2010, Spoormaker et al., 2012b, Tagliazucchi et al., 2013a, Tagliazucchi and Laufs, 2014, Altmann et al., 2016, Haimovici et

al., 2017). Alternative parcellation, such as those derived from FC configurations in the data, could be problematic, since FC has been shown to robustly vary across the sleep cycle (Tagliazucchi and Laufs, 2014, Altmann et al., 2016). Being anatomically defined the AAL is essentially agnostic to potentially changing FC configurations within the data.”

10 - In the fMRI paragraph, the 2nd sentence "MRI and pulse artefact correction were applied on the average artefact subtraction method ..." describes a method used to clean the EEG.

We thank the reviewer, and have corrected this in the revised ms.

References

- Braun, A. R., Balkin, T. J., Wesenten, N. J., et al. (1997). Regional cerebral blood flow throughout the sleep-wake cycle. An h2(15)o pet study. *Brain*, 120 1173-1197.
- Carskadon, M. A., & Dement, W. C. (2011). Chapter 2 - normal human sleep: An overview. In *Principles and practice of sleep medicine (fifth edition)* (pp. 16-26). Philadelphia: W.B. Saunders.
- Olbrich, S., Mulert, C., Karch, S., et al. (2009). Eeg-vigilance and bold effect during simultaneous eeg/fmri measurement. *Neuroimage*, 45(2), 319-332.
- Tagliazucchi, E., von Wegner, F., Morzelewski, A., et al. (2013). Breakdown of long-range temporal dependence in default mode and attention networks during deep sleep. *Proc Natl Acad Sci U S A*, 110(38), 15419-15424.
- von Wegner, F., Tagliazucchi, E., & Laufs, H. (2017). Information-theoretical analysis of resting state eeg microstate sequences - non-markovianity, non-stationarity and periodicities. *Neuroimage*, 158, 99-111.

Reviewer #2 (Remarks to the Author):

The authors describe the use of a data driven method to study the spatiotemporal structure of previously acquired fMRI data with the goal of characterizing changes in across the sleep-wake cycle. This is an interesting topic as it is a relatively unexplored area, in part due to the sleep adverse conditions of the MRI scanner. The approach is based on a Markov model of the fMRI signal and allows classification of brain activity in a set of prototypical patterns whose occurrence and intensity may vary with arousal state. This potentially allows a better characterization of sleep than previous fMRI studies based on polysomnography (PSG) sleep staging or correlations with EEG events such as K-complexes and spindles. The authors, limiting their analysis to non-REM sleep, identify a number of fMRI activity patterns (“states”) that preferentially occur during wake and each of three EEG-defined sleep stages, and quantify transition probabilities between these states. Based on this, they conclude that the results provide a novel and more complete characterization of sleep than conventional PSG-based sleep staging, that the default-mode network serves as a “gate” for the transition into non-REM sleep, and that there is a decrease in “whole brain dynamics during consolidated stages of non-REM sleep”.

Overall I believe this is a valuable study presenting a more rigorous way for the analysis of fMRI sleep data than previously presented. The identified spatiotemporal patterns are interesting and may provide more insight into the nature of sleep.

We thank the reviewer for the positive comments.

I do have a number of reservations though about both the methodology, and the presentation of the results:

1 - Throughout the manuscript, PSG based sleep staging is being used as straw man to contrast with the authors’ method. This is not appropriate, as the purpose of PSG is not to comprehensively characterize sleep and its activity patterns, rather than discriminating between various depths of sleep (arousal thresholds).

We agree and have updated ms accordingly:

Page 3, paragraph 3:

“PSG has been vital in the development of modern sleep research, and remains undoubtedly the quickest and easiest way to establish arousal levels in individuals. Indeed the PSG-defined sleep stages were originally devised from EEG as surrogate markers of arousal thresholds, yet over time many have come to see them as a more or less exhaustive set of intrinsic canonical states that cover the full repertoire of brain activity during sleep. However, from a neurobiological perspective, the use of fixed scoring windows of 30 seconds and only a few EEG electrodes means that PSG involves considerable averaging of brain activity in both time and space (Himanen and Hasan, 2000). This is necessarily an incomplete representation of brain activity, which for instance does not accommodate intracortical evidence showing that slow oscillations, sleep spindles, and K-complexes are not inevitably global phenomena, but can occur locally throughout the cortex (Nir et al., 2011, Vyazovskiy et al., 2011, Piantoni et al., 2017).”

Page 11, paragraph 5:

“Up until this point, we have used the traditional PSG stages to organise and evaluate the temporally resolved whole-brain network states, in terms of the MANOVA results, PSG-sensitivity, PSG-specificity, and dynamics within PSG stages. Yet, the data-driven nature of the HMM also allowed us to reverse the inference, and consider the results, in particular the temporal progression of HMM states, in their own right. This way we were able to ask if the high-resolution, fMRI-based HMM, was able to reveal new aspects of the wake-NREM sleep cycle, hidden from the EEG-based PSG. For this purpose it was informative to examine the transition probabilities of the HMM states. Specifically, we could use these to extract modules of HMM states that transitioned more often between each other than to other states. While agreeing to a certain extent with the PSG staging, the organisation of transition modules and their involved whole-brain network states also demonstrate how this data-driven approach can be used to more fully describe the spatiotemporal complexity of large-scale brain activity across the NREM sleep cycle.”

Page 20, paragraph 1:

“Comparing the temporal evolution of the HMM-derived whole-brain network states with the independently obtained EEG-based PSG scoring, we have discovered a rich repertoire of brain dynamics during the traditional PSG stages. Importantly, the temporal resolution of the HMM solution identified state lifetimes on the order of seconds, providing a temporally fine-grained description of the traditional PSG stages. Meanwhile, a close examination of the HMM transition map offers a credible account of the aspects where PSG does not fully capture the complexity of large-scale network changes, and lead to propositions of new and potentially improved categorisations of brain activity.”

Page 21, paragraph 2:

“Below we begin by linking our HMM results to existing neuroimaging evidence of PSG stages. This is used to characterise the ways in which the HMM agreed with PSG and establish that the HMM generally inferred meaningful features from the fMRI data. It is on the basis of this general agreement with PSG that we may then move on to a close examination of the new insights with relation to sleep that can be gained from the data-driven HMM.”

Page 23, paragraph 2:

“In the aspects where PSG and the HMM decomposition conformed in time, the HMM lends important support to PSG, while still offering new perspectives in terms of transitions and dynamics. In other cases, the HMM suggests new segmentations of large-scale brain activity that cannot be resolved when following traditional PSG staging.”

At the same time, numerous other methods have been used to study sleep, including EEG, MEG, ECoG, and even PET and fMRI (the former 3 distinguishing microstates, the latter 2 exploring e.g. the PET or fMRI correlate of K-complexes and spindles). This literature should be discussed. In addition, it should be mentioned that PSG can be used on single subjects, rather than requiring the large dataset analyzed here.

We have added these important points to the Introduction and Discussion.

On previously used methods to study sleep:

Page 4, paragraph 2:

“Large-scale brain activity has been extensively investigated during *wakefulness* (Biswal et al., 1995, Shulman et al., 1997, Raichle et al., 2001, Beckmann et al., 2005, Fox et al., 2005, Bressler and Menon, 2010, Brookes et al., 2011, Hipp et al., 2012, Baker et al., 2014, Gonzalez-Castillo et al., 2015, Smith et al., 2015, Cabral et al., 2017, Vidaurre et al., 2017b). When applied to *sleep* recordings, however, whole-brain analyses have commonly used PSG staging as a ‘gold standard’, regressing PSG stages onto functional brain data. This approach has yielded whole-brain correlates of PSG stages and sleep graphoelements, in terms of activation maps (Maquet, 2000, Dang-Vu et al., 2010), FC patterns (Horovitz et al., 2009, Larson-Prior et al., 2009, Spormaker et al., 2010, Sämann et al., 2011, Tagliazucchi and Laufs, 2014, Altmann et al., 2016), graph-theoretical measures (Boly et al., 2012, Tagliazucchi et al., 2013a), and EEG-microstates (Brodbeck et al., 2012). However, this top-down constraint by the low-resolution and subjective manual PSG scoring comes at the cost of exploring only a small fraction of the information available in the high-resolution neuroimaging data.”

Page 21, paragraph 3:

“The HMM was sensitive to changes in both mean BOLD activity and FC, and the overall reflection of PSG stages in the whole-brain network states suggest that these two features of brain activity changed reliably with PSG stages. This is in line with previous PET (Braun et al., 1997, Hofle et al., 1997, Maquet, 2000, Kjaer et al., 2002) and fMRI (Laufs et al., 2006, Dang-Vu et al., 2008, Horovitz et al., 2008) studies that have used PSG as gold standard to show differences between individual PSG stages in metabolic and BOLD activity, an EEG study showing PSG-dependent changes in EEG microstates (Brodbeck et al., 2012), as well as demonstrations that machine-learning can be used to classify fMRI recordings into PSG stages based only on FC patterns (Tagliazucchi and Laufs, 2014, Altmann et al., 2016). Further to this, the recent study by Haimovici and colleagues demonstrated that PSG stages could be identified as individual states of dynamic FC (Haimovici et al., 2017).”

Page 22, paragraph 4:

“Two HMM states were clearly specific to periods of N2 sleep as two whole-brain network states. The corresponding mean activation maps showed either increases or decreases in areas consistently identified in a number of studies as fMRI-correlates of sleep spindles (Schabus et al., 2007, Andrade et al., 2011, Caporro et al., 2012). It is likely that the HMM was sensitive to the signal changes caused by spindles, and future work should address more specifically the relationship between data-driven, HMM-identified networks and EEG-defined spindles as well as other graphoelements such as K-complexes.”

Page 23, paragraph 1:

“These frontal decreases are consistent with previous PET findings of decreased metabolism in these areas during N3 sleep, which in turn are believed to reflect the high, localised concentration of slow-wave activity (Werth et al., 1997). Slow waves may also underlie the relative increases in frontal FC that we observed for this whole-brain network state (Murphy et al., 2009).”

On the use of PSG and potentially HMM on single subjects:

Page 3, paragraph 3:

“PSG has been vital in the development of modern sleep research, and remains undoubtedly the quickest and easiest way to establish arousal levels in individuals.”

Page 25, paragraph 2:

“The presented findings point ahead to a research agenda making hypothesis-driven assessments of how the alternative, data-driven, temporal segmentations and dynamics of whole-brain networks across the NREM sleep cycle relate to sleep behaviour and cognition, when measured independently of PSG. Features identified by the HMM could prove to essential supplements to PSG and other conventional methods when trying to understand phenomena like the subjective perception of sleep (Bonnet and Moore, 1982, Ogilvie and Wilkinson, 1984), mental content during sleep (Nobili et al., 2012, Siclari et al., 2017), such as the hypnagogic or even hallucinogenic character of sleep onset (Hori et al., 1994, Goupil and Bekinschtein, 2012), sleep inertia of the awakening process (Tassi and Muzet, 2000), sleep-dependent processes related to memory and learning (Diekelmann and Born, 2010), and disordered sleep, like insomnia (Wei et al., 2017). Such studies should explore the theoretical potential of applying the current HMM, parameterised on the present sleep data, to identify the presence of the same dynamical whole-brain network states and transition modules in data from different cohorts, potentially even at the individual level. This new data could then be linked to behaviour and cognition through sophisticated measures of arousal, such as eyelid-closure (Chang et al., 2016), sleep mentation (Siclari et al., 2017), post-sleep memory- and learning performance (Stickgold, 2005), and careful clinical examination of sleep disorders (Edinger et al., 2013).”

2 - The hidden Markov model (HMM) approach to perform data-driven analysis is interesting but by itself does not guarantee the identification of neuro-scientifically relevant patterns of brain activity. Like a number of previously presented approaches (co-activation pattern based analysis, temporal functional modes, point-process analysis etc. etc), it allows extraction of statistically different patterns of fMRI activity but provides little understanding of what these patterns relate to or mean.

It is correct that HMM does not provide mechanistic information on the mechanisms underlying sleep, yet this is true of all analysis methods that do not have a clear biophysical basis. We have clarified this important limitation shared with all existing methods for extracting dynamic patterns of brain activity. Notwithstanding, the HMM does offer novel spatiotemporal information on NREM sleep cycle, which may be useful for future whole-brain computational modelling studies. We have updated the revised ms to include this important perspective:

Page 26, paragraph 3:

“It should be noted that the HMM framework was chosen over other methods for extracting dynamic states from multivariate neuroimaging datasets, such as sliding-window clustering (Allen et al., 2014, Hindriks et al., 2015, Haimovici et al., 2017), point-process analysis (Tagliazucchi et al., 2012a), co-activation pattern analysis (Liu and Duyn, 2013, Karahanoğlu and Ville, 2015) (for reviews, see (Calhoun et al., 2014, Preti et al., 2016)). The HMM was particularly suitable for our purpose by virtue of its explicit modelling of temporal dynamics, resulting in states that repeat in a

predictable way. Although the HMM is not a mechanistic model of brain activity (a limitation shared with the approaches mentioned above) we have shown how the explicitly modelled HMM transition matrix was fundamental to suggest new partitions of dynamic whole-brain states, which future mechanistic frameworks of NREM sleep and wakefulness should take into account.”

Specific problems I see with the implementation here is the quite arbitrary choice of the number of states (see minor comments as well),

This is a very important point. Please see our response to Reviewer 1 above, where we explore the robustness of the results with regards to the number of HMM states.

and the use of an unequal distribution of data over the different PSG stages.

This could affect the fMRI patterns of the individual states, and the conclusion about what these patterns may mean. Regarding the conclusion that fewer states were found during the deepest PSG sleep stage, this could be simply due to the fact that there was only 10% of deep sleep in the 57 subjects studies.

To this point, HMM state 16 that was the only state found for N3 sleep was only specific to N3 in 4 (out of 57!) subjects. Similarly on page 21, the authors say they “provide direct evidence of a higher state repertoire” during wakefulness. Again, much more data was collected during wake, potentially biasing the number of wake vs sleep states. In addition, more states found across subject does not mean more states WITHIN subjects, while the latter would be the more interesting finding. I would not be surprised if different people were thinking about different things during wake! (see also below).

The reviewer raises another important point. Note that throughout the ms we only refer to a given HMM state as modelling any given PSG stage, if it showed both sensitivity and specificity for this PSG stage, and if it did so consistently across participants. While the HMM is estimated on the full data from 57 participants, where there are indeed a larger amount of wakefulness (47.82% as shown in Table S1), all of the statistics linked to the statement about state repertoire have been carried out in the 18 participants with all four PSG stages. In these 18 participants, wakefulness only takes up 23.72% of the data, while N3 accounts for 31.78%, ie more of the data (see Table S1). In addition, the relevant measures for a higher state repertoire (switching and range of states) were both normalised by the time spent in a state *within* a participant (**Figure 3D** and **3E**). In general, all statistical comparisons were made using paired t-tests and the non-parametric permutations of the PSG scoring were made within participants, hence taking into the account the participant level. Overall, this gives us confidence that the results in fact as interesting as the reviewer is suggesting. Also, please note that we are not making specific claims about the cognition during wakefulness.

Of note, further balancing the distributions of PSG stages is not straightforward since this would imply severe cutting of data segments, effectively destroying the natural progression between stages. As may be seen in Figure 3 the HMM was highly sensitive to even rapid switching between PSG stages, and hence we see this natural order of the data as crucial proof-of-concept for our method.

With regards to the reviewer's specific comments on HMM state 16, we are not quite sure where this information has been gathered. All figures in the main text are based on the data from 18 participants who include all four PSG arousal levels. The results shown in **Figure 2** show the state timecourse of each HMM state and the association to PSG for the 18 participants with number 17 and 52 shown in extra detail. The timecourse of HMM state 16 is shown as the fourth line from the top. As can be seen, this state is present in all 18 participants and quite selectively for PSG defined N3 (highlighted as green periods). The specificity of HMM state 16 to N3 sleep is quantified in **Figure 3B** with a mean across participants of around 80% and a small error bar (across participants).

We have updated the revised ms to emphasise that our statistics take into account within-participant differences, and that the measures for the HMM state dynamics were normalised by time.

Page 5, paragraph 4:

“When comparing the HMM output to the PSG scoring we considered the subset of the HMM output that corresponded to the data from the 18 participants that reached all four PSG stages (wakefulness, N1, N2, and N3, see Table S1). This was necessary to allow for within-participant testing (see Methods), when comparing the sensitivity and specificity of individual HMM states to (as well as the dynamics of the HMM states within) the various PSG stages.”

Page 11, paragraph 3:

“The changes in mean lifetimes across PSG stages were further reflected in the dynamics of the whole-brain network states during the individual PSG stages, for which we present two summary measures: i) the amount of *switching* defined as the average number of transitions between HMM states during a given PSG stage divided by the total time a participant spent in this PSG stage (shown in Figure 3D), and ii) the *range* of HMM states defined as the number of unique states visited during the given PSG stage divided by the total time a participant spent in this PSG stage (shown in Figure 3E). Both measures were estimated for each PSG stage, within each of the 18 participants that included all four PSG stages, and normalised by time.”

Page 34, paragraph 1:

“Please note that the use of paired t-tests ensured that the identified differences were consistent within participants, and not merely as a group effect.”

3 - The conclusion that the default-mode network serves as a “gate” for the transition into non-REM sleep appears based on the higher transition probability between state 8 (with high DMN activity) and state 15. This seems unwarranted, as there are a significant number of transitions between state 2 (low DMN activity) and state 15 as well.

This is a fair point. As stated in our reply to reviewer 1, we have devised a new method for visualising and characterising transition maps. This new method allowed us to extract modules of strong transitions between HMM states. Importantly, these transition modules helped us compare the transition map from different HMM solution with varying numbers of states, which was crucial to show the robustness of our main findings. As can be seen from the new Figure 4C, state 8 is not the only pathway from wakefulness to white module related to N1 sleep, yet still remains the most important, since it represented the strongest transition from the red module to HMM state 15.

However, we have tempered our claims of the exclusivity of the state 8 as a gateway in the revised ms.

Page 13, paragraph 2:

“Although the transition map suggests multiple pathways from wakefulness (red module) to the white module of NREM sleep, it is interesting to note that *HMM state 8* has direct access to *HMM state 15*, which in turn guards the transition to the blue module of N2/N3 sleep.”

4 - The authors comment on the variety of states occurring during each EEG sleep stage but this is a bit misleading as this was a group-based determination. It is possible that some each subject exhibits a smaller repertoire of states. Similarly, talking about a “higher state repertoire” during wake may be valid across subjects, but authors have not demonstrated this to be the case in individual subjects.

Again, we have clarified our description of Figure 2, which shows HMM timecourses for the individual PSG staging for the 18 participants with all four sleep stages. Please see above in our reply to comment 2 for further explanation.

5 - As physiological changes are known to be prominent across sleep (in fact, they are increasingly being used to perform sleep staging), the authors should comments on how much they believe RETROICOR removes systemic physiological effects from the fMRI data, as well as the potential remaining confounds.

This is an important point. We now emphasise the findings of Tagliazucchi and colleagues (Tagliazucchi et al., 2012b), comparing sleep staging via SVM with and without prior RETROICOR-based noise regression to conclude: "Sleep staging performance was improved in the datasets without noise removal." Hence, we do believe that noise removal does reduce systemic physiological effects, which, however, indeed seem to vary systematically between sleep stages:

Page 29, paragraph 5:

“As shown by Tagliazucchi and colleagues, the use of RETROICOR effectively removes sleep-dependent physiological interference from the fMRI signals (Tagliazucchi et al., 2012b).”

6 - The abstract is vague and somewhat speculative. Terms like “top-down defined sleep stages” and “comprehensive image of brain states” are uninformative. “going beyond PSG, HMM allowed ...” is misleading. PSG is not meant to characterize the spatiotemporal patterns of brain activity, merely provide a surrogate marker of arousal threshold. Also, likely other (non HMM) analysis methods will allow this characterization, so I feel the important new thing here is the data driven aspect, not the specific analysis method used.

We have updated the abstract accordingly, aiming to make it more specific.

“The modern era of sleep research relies on measurements of brain activity, yet the mapping of whole-brain activity during human sleep leaves much to be desired and progress has been slow. Here we aimed to move significantly beyond the current state-of-the-art description of sleep, and in particular characterise the spatiotemporal complexity of whole-brain networks and state transitions

during sleep. In order to obtain the most unbiased estimate of how whole-brain network states evolve through the human sleep cycle, we used a Markovian data-driven analysis of continuous neuroimaging data from 57 healthy participants falling asleep during simultaneous functional magnetic resonance imaging and electroencephalography (EEG). This Hidden Markov Model (HMM) facilitated discovery of the dynamic choreography between different whole-brain networks across the wake-non-REM sleep cycle. Notably, our results reveal key trajectories to switch within and between EEG-based sleep stages, while highlighting the heterogeneity of N1 sleep and wakefulness before and after sleep.”

Lastly, the authors should list here the specific neuro-scientific findings of their work.

We have added more on the neuroscientific importance in the abstract. Please see immediately above.

Minor comments:

1) Several times the authors mention the study subjects “Across the sleep wake cycle”. This is not correct as REM sleep was not analyzed.

We completely agree, and have updated the revised ms accordingly.

2) Page 1, Line 1: Is "underlying" the best word for the title? Perhaps not, but there may be a better way to signal to potential readers that the analysis was performed across the conventional sleep stages. Maybe "Discovery of whole-brain transitions and dynamics across the conventional human sleep stages."

We agree with the reviewer that the wording of the title is crucial for appropriately conveying to potential readers the nature of the analyses and results. Please note that we have updated the title of the revised ms when addressing a comment from Reviewer 1 (see above). The updated title is “Discovery of key whole-brain transitions and dynamics during the human non-REM sleep cycle”. We think that this new and improved title signals that our study concerns conventional sleep stages

3) Page 1, Line -14: The first phrase of the abstract is true but unnecessary, distantly related to the specific sleep neuroscience topic under study

We have completely updated the abstract in the revised ms.

4) Page 2, Line -2: Was polysomnography ever indicated in the clinical evaluation of insomnia?

As concisely reviewed by Edinger et al. 2013, polysomnographic parameters such as sleep onset latency, wake time during sleep, and total sleep time have been commonly used for subject selection for clinical trials on sleep medication. Edinger et al. 2013 moreover demonstrate that many people suffering from insomnia will be missed by such procedure.

5) It is stated at the bottom of page 2 that PSG is poor for studying insomnia. Are the authors proposing that fMRI is better?

We may have been imprecise in our formulation. We meant to convey that the standard polysomnographic parameters have not been to helpful in discriminating cases with insomnia from controls, nor in understanding underlying mechanisms. Whereas it is unlikely that fMRI will develop to be a method that is feasible enough to support a diagnosis of insomnia, we consider it likely that probabilities of states and their transitions could provide new insights in underlying mechanisms (see e.g. Wei et al 2017).

Page 25, paragraph 2:

“The presented findings point ahead to a research agenda making hypothesis-driven assessments of how the alternative, data-driven, temporal segmentations and dynamics of whole-brain networks across the NREM sleep cycle relate to sleep behaviour and cognition, when measured independently of PSG. Features identified by the HMM could prove to essential supplements to PSG and other conventional methods when trying to understand phenomena like the subjective perception of sleep (Bonnet and Moore, 1982, Ogilvie and Wilkinson, 1984), mental content during sleep (Nobili et al., 2012, Siclari et al., 2017), such as the hypnagogic or even hallucinogenic character of sleep onset (Hori et al., 1994, Goupil and Bekinschtein, 2012), sleep inertia of the awakening process (Tassi and Muzet, 2000), sleep-dependent processes related to memory and learning (Diekelmann and Born, 2010), and disordered sleep, like insomnia (Wei et al., 2017). Such studies should explore the theoretical potential of applying the current HMM, parameterised on the present sleep data, to identify the presence of the same dynamical whole-brain network states and transition modules in data from different cohorts, potentially even at the individual level. This new data could then be linked to behaviour and cognition through sophisticated measures of arousal, such as eyelid-closure (Chang et al., 2016), sleep mentation (Siclari et al., 2017), post-sleep memory- and learning performance (Stickgold, 2005), and careful clinical examination of sleep disorders (Edinger et al., 2013).”

6) *Top page 3; are predictions by the listed theories captured by states found by authors?*

While acknowledging that our analysis was fundamentally exploratory, and not devised to test any specific predictions from any particular theoretical framework, we would like to draw the reviewer’s attention to the following excerpt from the Discussion:

Page 23, paragraph 3:

“The classical, PSG-defined wakefulness, N2, and N3 sleep each corresponded well to specific collections of whole-brain network states. However, through the HMM we gained access to the large-scale brain dynamics, showing that the state repertoire, when estimated as amount of switching and range of states visited, is higher in wakefulness than in both N2 and N3 sleep. From theoretical frameworks, such as the Integrated Information Theory (Tononi, 2008) and the dynamic core hypothesis (Tononi and Edelman, 1998), it follows that a higher and more complex state repertoire is important for the brain to support wakeful consciousness (Marshall et al., 2016). This has received empirical support from a series of combined TMS and EEG studies (Massimini et al., 2005, Casali et al., 2013). From a large-scale network perspective fMRI has been used to show how an enhanced state repertoire is associated with an “expanded” consciousness during the psychedelic experience (Carhart-Harris et al., 2014, Tagliazucchi et al., 2014). In the context of sleep, however, the large-scale network evidence is mainly represented by static FC studies suggesting decreased

information integration during N2 and N3 sleep using graph theory (Boly et al., 2012, Tagliazucchi et al., 2013a), as well as a higher exploration of the structural connectome during wakefulness (Tagliazucchi et al., 2016). With the present work we provide more *direct* evidence of a higher state repertoire in whole-brain dynamics during wakefulness.”

7) Page 3, Line 12: *Are the text and references on brain activity during wakefulness necessary? It may be prudent to delete the text between "wakefulness" and "however"?*

According to the reviewer’s suggestion, we have updated the manuscript to make the text more focussed.

8) Page 3, Line 19: *The authors exhibit intellectual honesty by including their publications in the list of publications that they, in the next sentence, carefully suggest has a key limitation.*

We thank the reviewer for acknowledging this.

9) Page 3, Line -7: *I do not understand why Reference 54 is unique. Does it not have the same limitation of a top-down constraint of the conventional sleep stages as the above references? It seems out of place here. Should it be moved and grouped with those publications at the beginning of the preceding paragraph? The same questions apply to the text associated with it at the initial two points in the Discussion section where it is specifically mentioned.*

We thank the reviewer for the opportunity to clarify why this reference is unique in the context of the current study. The study by Haimovici and colleagues (Haimovici et al., 2017) hold a special position in that the analysis does not involve any prior assumptions about the sleep structure of the fMRI data. The sliding-window-based clustering is used directly on the full data set, and the resulting states of dynamic functional connectivity are then compared to the sleep scoring. In this sense the approach to the PSG-scored fMRI data in Haimovici et al. 2017 is closely related to the work in the present study. The important difference lies in the motivation and aim of the respective analyses. Whereas Haimovici and colleagues were aiming to find one dynamic FC state for each traditional sleep stage, we have increased the number of states with the HMM, and leveraged the transition matrix output by the HMM to investigate the finer temporal details of the traditional sleep stages. Still, to our knowledge, Haimovici et al. (2017) is the first study to approach fMRI data of sleep without a strict constraint on the analysis from PSG, and that is why we feel it deserves special mention in our ms. We have updated the revised manuscript to make this more clear in the text.

Page 4, paragraph 3:

“A recent study showed, for the first time, that individual PSG stages can be extracted from fMRI recordings in a data-driven way as states of dynamic FC in a sliding-window analysis (Haimovici et al., 2017).”

10) Page 4, Line 1: *The different sample sizes used throughout the manuscript is confusing. It would be worthwhile to add some unambiguous text here and in the Participants subsection in the Methods section.*

We thank the reviewer for pointing this out. We have emphasised the use of a subset of the HMM solution throughout the revised ms:

Page 5, paragraph 4:

“When comparing the HMM output to the PSG scoring we considered the subset of the HMM output that corresponded to the data from the 18 participants that reached all four PSG stages (wakefulness, N1, N2, and N3, see Table S1). This was necessary to allow for within-participant testing (see Methods), when comparing the sensitivity and specificity of individual HMM states to (as well as the dynamics of the HMM states within) the various PSG stages.”

Page 7, paragraph 2:

“The temporal association is apparent when comparing the HMM state timecourses and the PSG scoring, as illustrated in Figure 2A, where data are plotted for the 18 participants that included all four PSG stages.”

Page 9, paragraph 2:

“For each of the 18 participants that included all four PSG stages, we defined the sensitivity of each HMM state as the proportion of total time spent in a PSG stage, in which this HMM state was active. Within the same 18 participants, the specificity was defined for each HMM state as the likelihood of finding this HMM state active during a given PSG stage..”

Page 11, paragraph 1:

“Having the whole-brain network states defined in time allowed us to investigate the large-scale brain dynamics of the traditionally defined PSG stages in the 18 participants that reached all PSG stages during their recordings.”

Page 11, paragraph 3:

“Both measures were estimated for each PSG stage, within each of the 18 participants that included all four PSG stages, and normalised by time.”

Page 30, paragraph 5:

“Following the HMM decomposition, two different subsets of the solution were used for post-hoc evaluation of the HMM. The first corresponded to the 18 participants that reached all four stages of PSG, and the second corresponded to the 31 participants that woke up after having reached consolidated sleep.”

11) Page 4, Line 7: The subsample of the 33 wakefulness-after-sleep-onset subjects should be added to Table S1.

We have added a table with PSG stage distributions for the 31 participants that woke up after having reached N2 sleep. Please also note that ‘33’ was a typo of the first ms, and we have updated this to the correct number of 31 participants, throughout the revised ms.

12) Why are some states not represented in Fig.4 (e.g. state14 which seems to have substantial fractional occupancy)

As we have clarified in our response to Reviewer 1 (please see above), a small number of the HMM states appeared to model participant-specific periods of the fMRI data, and were consequently non-recurring. HMM state 14 represents one of these states, which we refer to as ‘sporadic’. This lack of recurrence across participants is quantified in **Figure S7 B** of the original ms, from which it is clear that HMM state 14 did *not* occur in 80 % of the participants. While the fractional occupancy of HMM state 14 plotted in **Figure 3A** of the original ms may appear non-negligible, it is also evident that the error bars (indicating the standard error across participants) are very large on this estimation. This is again a sign that the fractional occupancy was high only for a small sub-set of participants. In order to address one of Reviewer 1’s comments (please see above), we have updated and improved our method for visualising the transition map of the HMM in **Figure 4** of the revised ms. As a step in this new method we exclude HMM states which occur in less than 75 % of the 18 participants with all four PSG stages.

13) Neuro-electrical activity such as K-complexes and spindles are known to be prevalent during sleep stages N1 and N2. Do the authors expect these activities to be represented in specific HMM states? Please Comment.

This is an interesting question. In our discussion of the mean activity maps identified by the HMM during N2 sleep, we note the visible overlap with areas identified as correlated with EEG-defined spindles in previous fMRI work. Please also see our reply to Reviewer 1 above.

Page 16, paragraph 2:

“N2 sleep was dominated by *HMM states 3 and 6*, and the mean activation maps of these whole-brain network states are shown in Figure 6B. The supplementary motor area was involved in both of these states; in *HMM state 3* as increases in concert with the bilateral precuneus and primary motor cortices; and in *HMM state 6* as decreases together with the bilateral thalamus, middle cingulate, supramarginal cortex, and the rolandic operculum. Interestingly, these configurations overlap considerably with those previously reported in studies mapping fMRI-correlates of sleep spindles (Laufs et al., 2007, Schabus et al., 2007, Andrade et al., 2011, Caporro et al., 2012), which represent a defining EEG-feature of N2 sleep.”

Page 22, paragraph 4:

“Two HMM states were clearly specific to periods of N2 sleep as two whole-brain network states. The corresponding mean activation maps showed either increases or decreases in areas consistently identified in a number of studies as fMRI-correlates of sleep spindles (Schabus et al., 2007, Andrade et al., 2011, Caporro et al., 2012). It is likely that the HMM was sensitive to the signal changes caused by spindles, and future work should address more specifically the relationship between data-driven, HMM-identified networks and EEG-defined spindles as well as other graphoelements such as K-complexes.”

14) The color bar in Fig. 5 needs to be labelled

We thank the reviewer for pointing this out, and we have corrected this in the revised ms.

15) Page 18. “Looking through glass, darkly” is awkward language.

We rather like the use of the biblical phrase from 1 Corinthians 13:12 as an image of the message of the study.

16) Page 18: “other studies have taken PSG as ground truth”. To some extent, this is what the authors do as well! (e.g. in grouping of states in sleep stages in figs 4, 5,7 and the conclusions that flow from that.

This is a fair question. The HMM is completely data-driven and thus independent of PSG or any other partitioning of sleep, yet in order to make sense of the data, we compare the HMM findings to PSG. Thus the argumentation of the ms includes a silver lining in terms of the use of PSG as ground truth. As it is true for most exploratory studies, there is a need for some referential concepts in order to disclose new features in data. For the most part we use the PSG scoring to make sense of the completely data-driven HMM segmentation of the data. The result may be interpreted as a probabilistic representation of PSG stages in the space of the HMM states. As we point out throughout the ms, there are certain aspects where the HMM states align well with PSG, in the sense that either groups of HMM states (in wakefulness and N2 sleep) or a single HMM state (in N3 sleep) closely models PSG stages. In other cases, such as N1 sleep, the HMM appears to suggest alternative groupings of data. Here the comparison between PSG and the HMM solution leads to a novel synthesis, pointing to a new and more robust way of defining transitory brain states of light sleep and the sleep onset process. Similarly, had we used PSG as the ground truth in a strict sense, and e.g. informed the HMM of the PSG scoring, we would have missed the independent module of transitions that appear to model wakefulness after sleep onset (WASO). However there are also parts of the results that consider PSG stages in their own right, such as when we compare the dynamics of the HMM states within the various PSG stages. We have updated the revised ms to make this more clear:

Page 11, paragraph 5:

“Up until this point, we have used the traditional PSG stages to organise and evaluate the temporally resolved whole-brain network states, in terms of the MANOVA results, PSG-sensitivity, PSG-specificity, and dynamics within PSG stages. Yet, the data-driven nature of the HMM also allowed us to reverse the inference, and consider the results, in particular the temporal progression of HMM states, in their own right. This way we were able to ask if the high-resolution, fMRI-based HMM, was able to reveal new aspects of the wake-NREM sleep cycle, hidden from the EEG-based PSG. For this purpose it was informative to examine the transition probabilities of the HMM states. Specifically, we could use these to extract modules of HMM states that transitioned more often between each other than to other states. While agreeing to a certain extent with the PSG staging, the organisation of transition modules and their involved whole-brain network states also demonstrate how this data-driven approach can be used to more fully describe the spatiotemporal complexity of large-scale brain activity across the NREM sleep cycle.”

Page 20, paragraph 1:

“Importantly, the temporal resolution of the HMM solution identified state lifetimes on the order of seconds, providing a temporally fine-grained description of the traditional PSG stages. Meanwhile, a close examination of the HMM transition map offers a credible account of the aspects where PSG

does not fully capture the complexity of large-scale network changes, and lead to propositions of new and potentially improved categorisations of brain activity.”

Page 21, paragraph 2:

“Below we begin by linking our HMM results to existing neuroimaging evidence of PSG stages. This is used to characterise the ways in which the HMM agreed with PSG and establish that the HMM generally inferred meaningful features from the fMRI data. It is on the basis of this general agreement with PSG that we may then move on to a close examination of the new insights with relation to sleep that can be gained from the data-driven HMM.”

Page 23, paragraph 2:

“In the aspects where PSG and the HMM decomposition conformed in time, the HMM lends important support to PSG, while still offering new perspectives in terms of transitions and dynamics. In other cases, the HMM suggests new segmentations of large-scale brain activity that cannot be resolved when following traditional PSG staging.”

17) Page 6, Line -12: Incorporating Reference 54 in this justification is awkward and unnecessary. This justification stands on its own quite well without the introductory phrase.

We thank the reviewer for this suggestion, and agree that the text reads better without the reference.

18) Page 11, Line 5: Perhaps I missed it, but how was the threshold determined?

The thresholding of the transition matrix was performed heuristically for visualisation purposes. As you will see in our various replies, we have developed a novel method for visualising the HMM transition map. This new method still includes thresholding of the transitions, also chosen solely for visualisation purposes. This is included in the revised ms.

Page 34, paragraph 3:

“The matrix of transition probabilities, which were explicitly modelled by the HMM, contained a clear organisation, in which sub-networks of HMM states expressed more frequent transitions within each other than to states outside. In other words, the transition matrix represented a directed graph with modular organisation. We demonstrated this by submitting the transition matrix (shown in Figure 4A) to a modularity analysis, using Matlab functions from the Brain Connectivity Toolbox (<https://sites.google.com/site/bctnet/Home>)(Rubinov and Sporns, 2010), based on Newman’s spectral community detection (Leicht and Newman, 2008). Prior to running the modularity algorithm, we excluded the transitions of the HMM states that did not occur consistently across participants, i.e. sporadic states (see Methods, Choice of number of states, and Figure 4B), and thresholded the remaining transition matrix to include the strongest elements. The choice of this latter threshold was done for visualisation purposes (for the results shown in the main text using 19 states we included the 21% strongest transitions), however different thresholds resulted in highly similar module partitions. The modular organisation is presented in a reordered matrix (Figure 4C) and as a map (Figure 4D).”

19) Page 11, Line -12: *The attempt to extend states across the conventional sleep stages is a very novel and important part of the manuscript. The above validation steps are nice, but this is where we will actually learn something new about sleep neuroscience. Indeed, that is exactly what happened when the authors separated wakefulness and wakefulness after sleep onset.*

We fully agree with the reviewer that this is one of the most novel aspects of the current study and have tried to emphasise this in the revised ms. Our updated modularity-based analysis of the transition map (shown in **Figure 4** of the revised ms) provides an even more rigid procedure for evaluating data-driven organisation of brain activity, resulting in the present case in a discovery of wakefulness after sleep onset and alternative identifications of light sleep.

Page 11, paragraph 5:

“Up until this point, we have used the traditional PSG stages to organise and evaluate the temporally resolved whole-brain network states, in terms of the MANOVA results, PSG-sensitivity, PSG-specificity, and dynamics within PSG stages. Yet, the data-driven nature of the HMM also allowed us to reverse the inference, and consider the results, in particular the temporal progression of HMM states, in their own right. This way we were able to ask if the high-resolution, fMRI-based HMM, was able to reveal new aspects of the wake-NREM sleep cycle, hidden from the EEG-based PSG. For this purpose it was informative to examine the transition probabilities of the HMM states. Specifically, we could use these to extract modules of HMM states that transitioned more often between each other than to other states. While agreeing to a certain extent with the PSG staging, the organisation of transition modules and their involved whole-brain network states also demonstrate how this data-driven approach can be used to more fully describe the spatiotemporal complexity of large-scale brain activity across the NREM sleep cycle.”

Page 20, paragraph 1:

“Meanwhile, a close examination of the HMM transition map offers a credible account of the aspects where PSG does not fully capture the complexity of large-scale network changes, and lead to propositions of new and potentially improved categorisations of brain activity.”

Page 21, paragraph 1:

“Agreements as well as disagreements between the PSG scoring and the independent HMM decomposition were made particularly clear in this transition map, visualising the most likely transitions between the whole-brain network states.”

Page 24, paragraph 2:

“If PSG-defined N1 does in fact represent a mix of wakefulness and sleep, this would explain why we found the highest range of whole-brain states during this PSG stage. While this primarily serves to underline the common notion that N1 is unlikely to be a reliable demarcation between wakefulness and sleep, our data-driven identification of a transition module of whole-brain states occurring between wakefulness and consolidated sleep (N2 and N3) suggests that an improved and principled categorisation of early sleep can be achieved, when the high spatiotemporal resolution of neuroimaging is explored in full.”

Page 24, paragraph 3:

“Like the N1-related findings discussed above, this too serves as a prime example of how information-rich neuroimaging data, when treated in a data-driven way, can be carefully evaluated in light of established knowledge (PSG in this case) to make new discoveries from, and categorisations of, brain activity.”

20) Page 11, Line -12: *Perhaps Figure S3 should be moved to the main text. The statistically significant sensitivity differences in opposite directions for states 2, 8, and 10 (wakefulness is higher) versus states 5, 17, and 18 (wakefulness after sleep onset is higher) is very compelling.*

We would be happy to do so but, given the number of existing figures, we would like guidance from the editor.

21) Page 19, Line 1: *The authors may want to begin this paragraph by briefly discussing how others have attempted to subdivide the conventional sleep stages with more-sophisticated analyses of EEG data (e.g., Reference 7 and Borbély, Baumann, Brandeis, Strauch, & Lehmann, 1981).*

We thank the reviewer for this excellent suggestion, and have used it to update the revised ms.

Page 20, paragraph 3:

“The description of brain activity offered by PSG has for long been acknowledged as incomplete for scientific purposes, and attempts have been made to harvest more information from scalp EEG in a search for features relevant for sleep, overlooked by PSG (Borbély et al., 1981, Hori et al., 2001, Terzano et al., 2001, Olbrich and Achermann, 2005, Abeyesuriya and Robinson, 2016). Our work shares this ambition, but includes an important incorporation of whole-brain spatial detail from fMRI data.”

22) Page 19, Line 4: *Does Figure 7 add information that is not available in the text?*

We feel this figure is useful for clarification purposes, but again we are happy to be guided by the editor.

23) Page 20, Line 9: *The authors may want to review a PET study on post-sleep wakefulness (Balkin et al., 2002) and determine whether it is worth including here.*

We thank the reviewer for reminding us of this pioneering study, which nicely corroborates our WASO-specific findings of mean activity increases in frontal cortices, and we have used it as a reference in this regard in the revised ms

24) Page 20, Line 17: *Reference 86 did not examine N1.*

The reviewer is correct. We have updated the revised ms.

25) Page 21, Line 14: *How dependent is this interpretation on the number of states initially chosen for the analysis?*

As we have shown in the result figures added to the supplementary material of the revised ms (Figure S7 to S10), the dynamics estimated within PSG stages were very robust across the numbers of states chosen for the HMM. Please see also our response to Reviewer 1.

Page 33, paragraph 2:

“In appreciation of the potential limitations related to choosing the number of HMM states with no strict, formal criterion, we include the results of using different numbers of HMM states. In Figures S7 to S10 we have reproduced Figure 3 of the main text with HMM results using 15, 17, 21, and 23 states respectively. Demonstrating the robustness of our HMM findings, the conclusions of the main text using 19 states are also found in Figures S7 to S10. Specifically Figures S7A-B to S10A-B show how select HMM states expressed high sensitivity and specificity for different PSG stages. In line with the results for $K = 19$ states, the HMM with lower and higher K identified states with high sensitivity and specificity for wakefulness, N2, and N3 sleep, but not for N1 sleep. Figures S7D-E to S10D-E quantify the dynamics of HMM states within PSG stages. The relative differences between PSG stages are conserved and highly stable across numbers of HMM states. Interestingly, the absolute values of switching between and range of HMM states within PSG stages were in fact also quite preserved across numbers of HMM states. This is likely caused by the fact that the main effect of changing the number of HMM states is an addition of non-recurring, ‘sporadic’, states that modelled very (participant-) specific periods of the fMRI data (see above).”

26) Page 22, Line -12: The differences between wakefulness and wakefulness after sleep onset should receive greater attention. They are the most novel, interesting, and exciting part of the manuscript. They should be cited as an example of how the approach and other similar approaches can be used to make truly new discoveries in sleep neuroscience.

We agree with the reviewer, and have highlighted this in the revised ms.

Page 24, paragraph 3:

“Like the N1-related findings discussed above, this too serves as a prime example of how information-rich neuroimaging data, when treated in a data-driven way, can be carefully evaluated in light of established knowledge (PSG in this case) to make new discoveries from, and categorisations of, brain activity.”

Page 25, paragraph 2:

“The presented findings point ahead to a research agenda making hypothesis-driven assessments of how the alternative, data-driven, temporal segmentations and dynamics of whole-brain networks across the NREM sleep cycle relate to sleep behaviour and cognition, when measured independently of PSG. Features identified by the HMM could prove to essential supplements to PSG and other conventional methods when trying to understand phenomena like the subjective perception of sleep (Bonnet and Moore, 1982, Ogilvie and Wilkinson, 1984), mental content during sleep (Nobili et al., 2012, Siclari et al., 2017), such as the hypnagogic or even hallucinogenic character of sleep onset (Hori et al., 1994, Goupil and Bekinschtein, 2012), sleep inertia of the awakening process (Tassi and Muzet, 2000), sleep-dependent processes related to memory and learning (Diekelmann and Born, 2010), and disordered sleep, like insomnia (Wei et al., 2017). Such studies should explore the theoretical potential of applying the current HMM, parameterised on the present sleep data, to identify the presence of the same dynamical whole-brain network states and

transition modules in data from different cohorts, potentially even at the individual level. This new data could then be linked to behaviour and cognition through sophisticated measures of arousal, such as eyelid-closure (Chang et al., 2016), sleep mentation (Siclari et al., 2017), post-sleep memory- and learning performance (Stickgold, 2005), and careful clinical examination of sleep disorders (Edinger et al., 2013).”

27) Page 22, Line -12: *Related to the previous comment, the authors may want to discuss other attempts to subdivide the conventional sleep stages with fMRI (Picchioni et al., 2008; Watanabe et al., 2014).*

We thank the reviewer for the suggestion, which has been followed in the revised ms.

Page 20, paragraph 3:

“The description of brain activity offered by PSG has for long been acknowledged as incomplete for scientific purposes, and attempts have been made to harvest more information from scalp EEG in a search for features relevant for sleep, overlooked by PSG (Borbély et al., 1981, Hori et al., 2001, Terzano et al., 2001, Olbrich and Achermann, 2005, Abeysuriya and Robinson, 2016). Our work shares this ambition, but includes an important incorporation of whole-brain spatial detail from fMRI data. Previous studies have indicated that fMRI can be used to identify dynamic re-configurations of large-scale brain activity during conventional, EEG-based sleep stages, either in form of voxel-wise changes in activity (Picchioni et al., 2008), changes in connection strengths in resting-state networks (Watanabe et al., 2014), or through long-range temporal dependencies in the BOLD signal (Tagliazucchi et al., 2013b). To our knowledge, however, there are no existing reports on how this information in fMRI data can be explored in a principled and data-driven way to suggest new categorisations of brain activity during human wake-NREM sleep cycle.”

28) Page 22, Line -9: *I appreciate how the authors make a call for the inclusion of other important brain processes. One that they might consider mentioning are behavioral measures of sleep such as eyelid closure, which has been applied to fMRI data during fluctuations of arousal (Chang et al., 2016), and arousal threshold, which forms the most important component of the original behavioral definition of sleep.*

29) Page 22, Line -9: *Related to the previous comment, when going beyond the conventional sleep stages, it may be worthwhile to mention that pre- versus postsleep adaptive brain processes related to the function of sleep (e.g., memory consolidation) should also be included.*

We thank the reviewer for the important suggestions of the two comments above and now include in the Discussion section of the revised ms:

Page 25, paragraph 2:

“The presented findings point ahead to a research agenda making hypothesis-driven assessments of how the alternative, data-driven, temporal segmentations and dynamics of whole-brain networks across the NREM sleep cycle relate to sleep behaviour and cognition, when measured independently of PSG. Features identified by the HMM could prove to essential supplements to PSG and other conventional methods when trying to understand phenomena like the subjective perception of sleep (Bonnet and Moore, 1982, Ogilvie and Wilkinson, 1984), mental content during sleep (Nobili et al., 2012, Siclari et al., 2017), such as the hypnagogic or even hallucinogenic

character of sleep onset (Hori et al., 1994, Goupil and Bekinschtein, 2012), sleep inertia of the awakening process (Tassi and Muzet, 2000), sleep-dependent processes related to memory and learning (Diekelmann and Born, 2010), and disordered sleep, like insomnia (Wei et al., 2017). Such studies should explore the theoretical potential of applying the current HMM, parameterised on the present sleep data, to identify the presence of the same dynamical whole-brain network states and transition modules in data from different cohorts, potentially even at the individual level. This new data could then be linked to behaviour and cognition through sophisticated measures of arousal, such as eyelid-closure (Chang et al., 2016), sleep mentation (Siclari et al., 2017), post-sleep memory- and learning performance (Stickgold, 2005), and careful clinical examination of sleep disorders (Edinger et al., 2013).

The wake-NREM sleep cycle merely represents a sub-part of a continuum of activities that the brain supports. Other important brain processes should be sought integrated with the presented transition map, most obviously including REM sleep, but also other altered states of consciousness, such as anaesthesia (Ni Mhuircheartaigh et al., 2013, Uhrig et al., 2014, Barttfeld et al., 2015), the psychedelic experience (Carhart-Harris et al., 2014), and even different contents of consciousness during wakefulness (Dehaene et al., 2014).”

30) Page 22, Line -3: Is discussing the application to other altered states of consciousness too loosely linked to the current study? Perhaps this sentence should be deleted.

Following the reviewer’s suggestion, we have tempered the sentence.

Page 25, paragraph 3:

“The wake-NREM sleep cycle merely represents a sub-part of a continuum of activities that the brain supports. Other important brain processes should be sought integrated with the presented transition map, most obviously including REM sleep, but also other altered states of consciousness, such as anaesthesia (Ni Mhuircheartaigh et al., 2013, Uhrig et al., 2014, Barttfeld et al., 2015), the psychedelic experience (Carhart-Harris et al., 2014), and even different contents of consciousness during wakefulness (Dehaene et al., 2014).”

31) Page 26, Line -5: The terms "more robust" and "potential noisy" are vague. It is not clear to the reader why principal component analysis was performed. Why was it performed here and not in the authors' prior publication (Reference 34)?

The HMM is a decomposition that intrinsically implies a dimensionality reduction. Because of its robustness and its capacity to filter out high-frequency noise, PCA is often used prior to other dimensionality reductions such as ICA. In (Vidaurre et al., 2017b), the authors ran the HMM on ICA components. In that case, given the huge spatial dimensionality of the data, PCA necessarily preceded ICA, and thus the HMM. In this case, we bypassed ICA by using an anatomical parcellation. Although not strictly necessary here, PCA is however convenient as it increases the signal-to-noise ratio, reduces the number of parameters estimated by the HMM and improves the robustness of the result. In this case, robustness refers to the statistical sense, i.e. less prone to estimation variability.

The revised ms has been updated:

Page 31, paragraph 1:

“The HMM represents a tool for decomposing multivariate data into fewer dimensions. Given the high spatial dimensionality of fMRI, it is common to use principal component analysis (PCA) to reduce the number of parameters to be estimated in the decomposition, increasing the signal-to-noise ratio of the data and improving the robustness of the results (Baker et al., 2014, Vidaurre et al., 2017b). Accordingly, we submitted the demeaned, standardised, and concatenated BOLD timecourses to PCA prior to the HMM inference.”

32) Page 26, Line -2: *Figure S1 is not necessary in general and is included as a panel in Figure 1.*

We agree with the reviewer and have removed the figure from the revised ms.

33) Page 27, Line 11: *More details on what exactly minimum free energy is measuring may be warranted.*

Page 31, paragraph 3:

“The free energy is the statistical measure that is minimised during the (variational inference) Bayesian optimisation process. Technically speaking, it is an approximation of the model evidence, and includes two terms: how well the model fits the data, and the complexity of the model (measured as how much it departs from the prior distribution).”

34) Page 27, Line 16: *Why did minimum free energy fail as the method for choosing the number of states? What is unique about these data that triggered this deviation from standard practice?*

As we note in our reply to Reviewer 1 above:

Whereas the free energy is a reasonable criterion, its biological validity is unclear as far as the HMM is not a biophysical model. As we already note in the Discussion section of the original ms, it remains “difficult to determine the ‘correct’ number of states, when decomposing continuous recordings of brain activity”. Ultimately, the question boils down to find an appropriate trade-off between richness and complexity of the model, and the appropriate number depends on the research question at hand. Similar to our case, furthermore, Baker et al. 2014 found minimum free energy to be non-informative for the purpose of choosing the number of states.

Page 32, paragraph 1:

“Whereas the free energy is a reasonable criterion for choosing the ideal number of states for the HMM, its biological validity remains unclear in so far as the HMM does not represent a biophysical model. As apparent from the plot in Figure S1A, the minimum free energy was monotonically decreasing over the large range of tested numbers of states, showing no negative peaks. Hence, like in previous applications of the HMM (Baker et al., 2014, Vidaurre et al., 2017b), the free energy was not informative for choosing the number of states in our case.”

35) Page 27, Line -8: *The authors used how well the states related to the conventional nonrapid eye movement sleep stages here to help them determine how many states to use. Does this not defeat the purpose of analyzing the fMRI data independently of the conventional sleep stages?*

Please see our response to minor comment 16.

36) Page 29, Line 1: *The AASM manual defines wakefulness after sleep onset as wakefulness after having reached any stage. Why did the authors choose to deviate from this standard?*

We agree with the reviewer that should be clarified. In line with inter-rater studies, cited in our ms, our results demonstrate PSG-defined N1 as the least reliable sleep stage to estimate. In order to stay consistent with our own findings we chose to examine wakefulness after consolidated sleep (N2/N3), which is also in line with previous studies investigating post-sleep wakefulness (Balkin et al., 2002).

Page 12, paragraph 4:

“Given the poor correspondence between the HMM states and the general uncertainty associated with the staging of PSG-defined N1 sleep (see Discussion), we chose to define WASO as PSG-staged wakefulness, which followed after visits to N2 sleep (Balkin et al., 2002).”

37) *Supplementary Material Page 5, Line 1: The authors should consider simplifying this figure further by also excluding the bars representing N1 and N2. The key question is whether W and WASO differ.*

We agree with the reviewer and have updated the figure in the revised ms.

38) *Supplementary Material Page 5, Line 1: "Including" may not be the best word. Is "separating" better?*

We thank the reviewer for this suggestion and have updated the revised ms accordingly.

39) *The justification for choosing the number of HMM states is not sound. E.g. in caption of Fig. S2 it says: that the “median fractional occupancy stagnates around $K=19$ ” to make the point that that higher K did not split states. These issues are not directly related though.*

The reviewer raises an important point, which is related to the minor comments 25 and 34, as well as a comment from Reviewer 1 (please see above). Appreciating that no formal criterion can be used to determine the ideal number of states for the HMM, we opted for a solution that could contribute with as high temporal detail as possible. The median fractional occupancy is related to this, since the rationale for increasing the number of states would be to model the data with more states. This should distribute the data across more states, and theoretically lead to lower median fractional occupancies of the resulting HMM states. However this is not what we observed when increasing the number of states above 19. When using $K > 19$ states, we found that the extra states modelled very specific periods of the data, often “specialising” in only one, or a few, participants. We illustrated this in **Figure S7** of the original manuscript, where we plot the proportion of participants including all HMM states against the number of states inferred by the HMM. The very same phenomenon is responsible for the stagnation of the median fractional occupancy shown in in the original **Figure S2**.

40) I found Figure 7 quite speculative, with several statements that are questionable in the light of my major comments (unequal amount of data in various sleep stages, states present in some subject but not in others, multiple pathways for the transition of wake to sleep etc.

Following the reviewer's comments, we have revised the ms, clarifying why the statements of Figure 7 are sound.

References:

- Balkin, T. J., Braun, A. R., Wesensten, N. J., Jeffries, K., Varga, M., Baldwin, P., . . . Herscovitch, P. (2002). The process of awakening: a PET study of regional brain activity patterns mediating the re-establishment of alertness and consciousness. *Brain*, 125, 2308-2319.
- Borbely, A. A., Baumann, F., Brandeis, D., Strauch, I., & Lehmann, D. (1981). Sleep deprivation: effect on sleep stages and EEG power density in man. *Electroencephalography and Clinical Neurophysiology*, 51, 483-495.
- Chang, C., Leopold, D. A., Scholvinck, M. L., Mandelkow, H., Ir, D., Liu, X., . . . Duyn, J. H. (2016). Tracking brain arousal fluctuations with fMRI. *Proceedings of the National Academy of Sciences of the United States of America*, 113, 4518-4523. doi: 10.1073/pnas.1520613113
- Noirhomme, Q., Soddu, A., Lehembre, R., Vanhaudenhuyse, A., Boveroux, P., Boly, M., & Laureys, S. (2010). Brain connectivity in pathological and pharmacological coma. *Frontiers in Systems Neuroscience*, 4, 160. doi: 10.3389/fnsys.2010.00160
- Picchioni, D., Fukunaga, M., Carr, W. S., Braun, A. R., Balkin, T. J., Duyn, J. H., & Horowitz, S. G. (2008). fMRI differences between early and late stage-1 sleep. *Neuroscience Letters*, 441, 81-85. doi: 10.1016/j.neulet.2008.06.010
- Watanabe, T., Kan, S., Koike, T., Misaki, M., Konishi, S., Miyauchi, S., . . . Masuda, N. (2014). Network-dependent modulation of brain activity during sleep. *Neuroimage*, 98, 1-10. doi: 10.1016/j.neuroimage.2014.04.079

Reviewer #3 (Remarks to the Author):

I - My commits will mainly focus on the HMM part. When applying Gaussian HMM on fMRI analysis, some papers assume the mean vector is zero and only study the covariance matrix as a functional connectivity matrix. Some papers assume the covariance matrix to be identity matrix and only focus on the mean vector as a mean activation pattern. In this paper, there seem to be no constraints on mean vector or covariance matrix; therefore, the state is represented by a mean activation pattern and a covariance matrix.

The reviewer is correct in pointing this out. Our choice of a full Gaussian distribution was based on an agnostic view of the data. We preferred to be as data-driven as possible in determining whether it is the functional connectivity or the signed amplitude (the mean of the distribution), which drive the segmentation. A more hypothesis-driven analysis can be done by suppressing either of these two elements, and is an interesting perspective for future investigations.

In this case, subtracting/averaging two/multiple covariance matrices from different brain states may be problematic (Figure S5 and S6), because this ignores the fact that these brain states has a different baseline activation pattern.

This is an excellent point. We interpret the covariance as representative of how the ROI timecourses co-fluctuate after the mean vector has accounted for their respective mean activities, which are different between the states. Note, however, that comparing (subtracting or averaging) functional connectivity matrices (either in the form of covariance or correlation) between different data sets – with different baselines – is relatively common in practice; see Danaher, Wang and Witten’s *The joint graphical lasso for inverse covariance estimation across multiple classes* for a principled example (Danaher et al., 2013).

To address the reviewer’s concern, we sought to verify that our results were robust by using the cosine similarity instead of Pearson’s correlation as a measure of functional connectivity. The difference between these two metrics is that the cosine similarity does not demean the time series, and, therefore, the differences in baselines are accounted for. As observed, the results are very similar between these approaches.

Page 35, paragraph 3:

“To make sure that the differential FC maps were not biased by the effect of the HMM states having different baseline mean activation patterns (as modelled by the mean vector of the Gaussian distribution), we produced equivalent maps using the cosine similarity instead of the covariance matrices outputted by the HMM. Unlike Pearson’s correlation, the cosine similarity does not demean the time series, and, therefore, the differences in baselines are accounted for. These maps are shown in Figures S17 and S18. This analysis yielded 19 matrices of 90×90 cosine similarity values. By taking each cosine similarity matrix and subtracting from it the average of the remaining 18 cosine similarity matrices, we obtained maps equivalent to the differential FC maps, which were based on the covariance information modelled directly by the HMM. As may be seen, when comparing Figure S4 to Figure S17, and Figure S5 to Figure S18, the maps are highly similar.”

2 - *If we look at the 19 covariance matrices in Figure S4, most part of them is positive. Even for the negative part, the values are relatively low compared with the positive part (-0.2 vs. +1.2). I am wondering whether this is due to the lack of global signal regression or it is because each of the covariance matrices is associated with a mean vector and that makes it different from the conventional FC matrix.*

We thank the reviewer for pointing this out. As explained above, please note that the FC matrices shown in Figure S3 were/are based on correlations. We have added a histogram to the revised Figure S3, showing the distribution of super-diagonal elements of the static FC matrices computed as the time-averaged Pearson's correlation between the original ROI timecourses within the four PSG stages across participants. These values are independent of the HMM and the histogram demonstrates that the positively skewed FC values were a feature of the fMRI data, and not the HMM.

As the reviewer points out, this is most likely a result of omitting global signal regression. The use of global signal regression as a preprocessing step for fMRI connectivity analysis is controversial (Murphy and Fox, 2016). In the context of vigilance fluctuations, it has been shown that the global signal may in fact carry physiological information, by varying across PSG stages (Wong et al., 2013). For these reasons we chose not to perform global signal regression.

3 - *I suspect the temporal features (e.g. mean life time (around 10-20s), switching frequency) will be highly influenced by the temporal filter (0.01-0.1Hz). It would be nice to prove that at least their relative relationship will not change by using a different temporal filter.*

The reviewer is correct. Omitting the use of the temporal filter does lead to increases in mean life times of the HMM states. We have added Figure S16, which demonstrates that the relative differences between PSG stages are effectively unchanged.

Figure text of Figure S16:

“All plots were computed in the same way as for Figure 4, using an HMM with 19 states. The only difference was that the BOLD data was not temporally filtered. Note in A and B how, similarly to the results of the main text, select HMM states were sensitive and specific for certain PSG stages (although not for N1 sleep). In C the overall mean life time of the HMM states is decreased compared to the results using a low-pass temporal filter (compare with Figure 4). As a consequence, D shows increased switching frequencies within all of the four PSG stages. Importantly, the relative differences between the PSG stages were effectively unchanged (even if the significant difference between ‘Wake’ and ‘N2’ was no longer evident). Interestingly, in E, the number of unique HMM states visited per unit time were numerically quite stable with or without the use of temporal filter.”

4 - *My biggest concern about these results is their reproducibility. As far as I know, methods like HMM are sensitive to its initialization. Training the model twice with different initialization may give you different results. Some states may appear slightly differently and some states may disappear. The temporal features of the model, including life time and transition matrix, may also change. Therefore, which part of the results is actually reproducible (insensitive to initialization)?*

This is a good point and we have carried out a thorough analysis of the reproducibility.

Page 26, paragraph 2:

“Another potential caveat of our analyses pertains to the initialisation of the HMM, which is not deterministic. In the Supplementary Materials, we provide a summary analysis showing that the HMM infers consistent states across independent initialisations and splits of data (see Figure S16).”

Page 36, paragraph 1:

“The initialisation of the HMM includes a stochastic element. To make sure that the states inferred by the HMM were not contingent on the initialisation, we ran the HMM with 19 states an additional four times on the full dataset ($N = 57$), and five times on each of the two half-splits of the data ($N = 29$ and $N = 28$). The 19 resulting states of each HMM repetition were matched to the states of the original HMM. Each state of a repetition was thus paired to an original HMM state, based on the similarity between their respective Gaussian distributions. The similarity was estimated using the Bhattacharyya distance (Bhattacharyya, 1943), and the matching of states across repetitions were carried out using the Munkres algorithm (Munkres, 1957).

Following the pairing of states, all resulting states were compared in an all-to-all manner, again using the Bhattacharyya distance as a measure of similarity. The resulting matrix $[(N_{\text{dataset}} + N_{\text{states}} + N_{\text{repetitions}}) \times (N_{\text{dataset}} + N_{\text{states}} + N_{\text{repetitions}})]$ is shown in Figure S15A. The common pattern in the dataset-specific sub-matrices indicates that consistent HMM-state distributions were inferred across initialisation repetitions and data-splits.

Following the matching of the HMM Gaussian distributions from independent initialisations, we tested the temporal correspondence between the original HMM states and their counterparts from the repetition runs. This was done by comparing the corresponding state timecourses. For a pair of HMM states (one original and one from a repetition run) the temporal correspondence was quantified as the ratio between time points of overlap (simultaneous activity or inactivity) and time points of misses (off-sets of activity or inactivity). In Figure S15B are plotted the mean values and standard deviations within data-splits, and it is clear that temporal overlaps outweighed misses for all runs of the HMM. This is an important indication that the evaluations of the HMM dynamics presented in the main text would be highly similar for other initialisations.”

5 - In page 5, please use multiplication symbol instead of letter "x" in the dimension of the matrix.

We have changed throughout the revised ms.

References for our replies to the reviewers

- Abeyesuriya RG, Robinson PA (2016) Real-time automated EEG tracking of brain states using neural field theory. *Journal of Neuroscience Methods* 258:28-45.
- Allen EA, Damaraju E, Plis SM, Erhardt EB, Eichele T, Calhoun VD (2014) Tracking whole-brain connectivity dynamics in the resting state. *Cerebral cortex* (New York, NY : 1991) 24:663-676.
- Altmann A, Schroter MS, Spormaker VI, Kiem SA, Jordan D, Ilg R, Bullmore ET, Greicius MD, Czisch M, Samann PG (2016) Validation of non-REM sleep stage decoding from resting state fMRI using linear support vector machines. *NeuroImage* 125:544-555.
- Andrade KC, Spormaker VI, Dresler M, Wehrle R, Holsboer F, Samann PG, Czisch M (2011) Sleep spindles and hippocampal functional connectivity in human NREM sleep. *The Journal of neuroscience : the official journal of the Society for Neuroscience* 31:10331-10339.

- Andrillon T, Nir Y, Staba RJ, Ferrarelli F, Cirelli C, Tononi G, Fried I (2011) Sleep spindles in humans: insights from intracranial EEG and unit recordings. *The Journal of neuroscience : the official journal of the Society for Neuroscience* 31:17821-17834.
- Baker AP, Brookes MJ, Rezek IA, Smith SM, Behrens T, Probert Smith PJ, Woolrich M (2014) Fast transient networks in spontaneous human brain activity. *eLife* 3:e01867.
- Balkin TJ, Braun AR, Wesensten NJ, Jeffries K, Varga M, Baldwin P, Belenky G, Herscovitch P (2002) The process of awakening: a PET study of regional brain activity patterns mediating the re - establishment of alertness and consciousness. *Brain : a journal of neurology* 125:2308-2319.
- Barttfeld P, Uhrig L, Sitt JD, Sigman M, Jarraya B, Dehaene S (2015) Signature of consciousness in the dynamics of resting-state brain activity. *Proceedings of the National Academy of Sciences of the United States of America* 112:887-892.
- Beckmann CF, DeLuca M, Devlin JT, Smith SM (2005) Investigations into resting-state connectivity using independent component analysis. *Philosophical transactions of the Royal Society of London Series B, Biological sciences* 360:1001-1013.
- Bhattacharyya A (1943) On a measure of divergence between two statistical populations defined by their probability distributions. *Bulletin of the Calcutta Mathematical Society* 35:99-109.
- Biswal B, Yetkin FZ, Haughton VM, Hyde JS (1995) Functional connectivity in the motor cortex of resting human brain using echo-planar MRI. *Magnetic resonance in medicine* 34:537-541.
- Boly M, Perlberg V, Marrelec G, Schabus M, Laureys S, Doyon J, Pelegriani-Issac M, Maquet P, Benali H (2012) Hierarchical clustering of brain activity during human nonrapid eye movement sleep. *Proceedings of the National Academy of Sciences of the United States of America* 109:5856-5861.
- Bonjean M, Baker T, Bazhenov M, Cash S, Halgren E, Sejnowski T (2012) Interactions between core and matrix thalamocortical projections in human sleep spindle synchronization. *The Journal of neuroscience : the official journal of the Society for Neuroscience* 32:5250-5263.
- Bonnet MH, Moore SE (1982) The threshold of sleep: perception of sleep as a function of time asleep and auditory threshold. *Sleep* 5:267-276.
- Borbély AA, Baumann F, Brandeis D, Strauch I, Lehmann D (1981) Sleep deprivation: Effect on sleep stages and EEG power density in man. *Electroencephalogr Clin Neurophysiol* 51:483-493.
- Braun AR, Balkin TJ, Wesenten NJ, Carson RE, Varga M, Baldwin P, Selbie S, Belenky G, Herscovitch P (1997) Regional cerebral blood flow throughout the sleep-wake cycle. An H2(15)O PET study. *Brain : a journal of neurology* 120 (Pt 7):1173-1197.
- Bressler SL, Menon V (2010) Large-scale brain networks in cognition: emerging methods and principles. *Trends in cognitive sciences* 14:277-290.
- Brodbeck V, Kuhn A, Wegner Fv, Morzelewski A, Tagliazucchi E, Borisov S, Michel CM, Laufs H (2012) EEG microstates of wakefulness and NREM sleep. *NeuroImage* 62:2129-2139.
- Brookes MJ, Woolrich M, Luckhoo H, Price D, Hale JR, Stephenson MC, Barnes GR, Smith SM, Morris PG (2011) Investigating the electrophysiological basis of resting state networks using magnetoencephalography. *Proceedings of the National Academy of Sciences of the United States of America* 108:16783-16788.
- Cabral J, Vidaurre D, Marques P, Magalhaes R, Silva Moreira P, Miguel Soares J, Deco G, Sousa N, Kringelbach ML (2017) Cognitive performance in healthy older adults relates to spontaneous switching between states of functional connectivity during rest. *Sci Rep* 7:5135.
- Calhoun Vince D, Miller R, Pearlson G, Adalı T (2014) The Chronnectome: Time-Varying Connectivity Networks as the Next Frontier in fMRI Data Discovery. *Neuron* 84:262-274.
- Caporro M, Haneef Z, Yeh HJ, Lenartowicz A, Buttinelli C, Parvizi J, Stern JM (2012) Functional MRI of sleep spindles and K-complexes. *Clinical neurophysiology : official journal of the International Federation of Clinical Neurophysiology* 123:303-309.

- Carhart-Harris RL, Leech R, Hellyer PJ, Shanahan M, Feilding A, Tagliazucchi E, Chialvo DR, Nutt D (2014) The entropic brain: a theory of conscious states informed by neuroimaging research with psychedelic drugs. *Frontiers in human neuroscience* 8:20.
- Carskadon MA, Dement WC (2011) Chapter 2 - Normal Human Sleep: An Overview. In: *Principles and Practice of Sleep Medicine (Fifth Edition)* (Kryger, M. H. et al., eds), pp 16-26 Philadelphia: W.B. Saunders.
- Casali AG, Gosseries O, Rosanova M, Boly M, Sarasso S, Casali KR, Casarotto S, Bruno MA, Laureys S, Tononi G, Massimini M (2013) A theoretically based index of consciousness independent of sensory processing and behavior. *Sci Transl Med* 5:198ra105.
- Chang C, Leopold DA, Schölvinck ML, Mandelkow H, Picchioni D, Liu X, Ye FQ, Turchi JN, Duyn JH (2016) Tracking brain arousal fluctuations with fMRI. *Proceedings of the National Academy of Sciences* 113:4518-4523.
- Ciuciu P, Varoquaux G, Abry P, Sadaghiani S, Kleinschmidt A (2012) Scale-Free and Multifractal Time Dynamics of fMRI Signals during Rest and Task. *Front Physiol* 3:186.
- Colrain IM, Campbell KB (2007) The use of evoked potentials in sleep research. *Sleep medicine reviews* 11:277-293.
- Danaher P, Wang P, Witten Daniela M (2013) The joint graphical lasso for inverse covariance estimation across multiple classes. *Journal of the Royal Statistical Society: Series B (Statistical Methodology)* 76:373-397.
- Dang-Vu TT, Schabus M, Desseilles M, Albouy G, Boly M, Darsaud A, Gais S, Rauchs G, Sterpenich V, Vandewalle G, Carrier J, Moonen G, Balteau E, Degueldre C, Luxen A, Phillips C, Maquet P (2008) Spontaneous neural activity during human slow wave sleep. *Proceedings of the National Academy of Sciences of the United States of America* 105:15160-15165.
- Dang-Vu TT, Schabus M, Desseilles M, Sterpenich V, Bonjean M, Maquet P (2010) Functional neuroimaging insights into the physiology of human sleep. *Sleep* 33:1589-1603.
- Dehaene S, Charles L, King JR, Marti S (2014) Toward a computational theory of conscious processing. *Current opinion in neurobiology* 25:76-84.
- Dehghani N, Cash SS, Chen CC, Hagler Jr DJ, Huang M, Dale AM, Halgren E (2010a) Divergent cortical generators of MEG and EEG during human sleep spindles suggested by distributed source modeling. *PloS one* 5:e11454.
- Dehghani N, Cash SS, Rossetti AO, Chen CC, Halgren E (2010b) Magnetoencephalography demonstrates multiple asynchronous generators during human sleep spindles. *Journal of neurophysiology* 104:179-188.
- Diekelmann S, Born J (2010) The memory function of sleep. *Nature reviews Neuroscience* 11:114-126.
- Edinger JD, Ulmer CS, Means MK (2013) Sensitivity and Specificity of Polysomnographic Criteria for Defining Insomnia. *Journal of clinical sleep medicine : JCSM : official publication of the American Academy of Sleep Medicine* 9:481-491.
- Eickhoff SB, Constable RT, Yeo BTT (2018) Topographic organization of the cerebral cortex and brain cartography. *NeuroImage* 170:332-347.
- Fox MD, Snyder AZ, Vincent JL, Corbetta M, Van Essen DC, Raichle ME (2005) The human brain is intrinsically organized into dynamic, anticorrelated functional networks. *Proceedings of the National Academy of Sciences of the United States of America* 102:9673-9678.
- Gonzalez-Castillo J, Hoy CW, Handwerker DA, Robinson ME, Buchanan LC, Saad ZS, Bandettini PA (2015) Tracking ongoing cognition in individuals using brief, whole-brain functional connectivity patterns. *Proceedings of the National Academy of Sciences of the United States of America* 112:8762-8767.
- Goupil L, Bekinschtein TA (2012) Cognitive processing during the transition to sleep. *Archives italiennes de biologie* 150:140-154.
- Haimovici A, Tagliazucchi E, Balenzuela P, Laufs H (2017) On wakefulness fluctuations as a source of BOLD functional connectivity dynamics. *Sci Rep* 7:5908.

- He BJ (2011) Scale-free properties of the functional magnetic resonance imaging signal during rest and task. *Journal of Neuroscience* 31:13786-13795.
- Himanen SL, Hasan J (2000) Limitations of Rechtschaffen and Kales. *Sleep medicine reviews* 4:149-167.
- Hindriks R, Adhikari MH, Murayama Y, Ganzetti M, Mantini D, Logothetis NK, Deco G (2015) Can sliding-window correlations reveal dynamic functional connectivity in resting-state fMRI? *NeuroImage* 127:242-256.
- Hipp JF, Hawellek DJ, Corbetta M, Siegel M, Engel AK (2012) Large-scale cortical correlation structure of spontaneous oscillatory activity. *Nature neuroscience* 15:884-890.
- Hofle N, Paus T, Reutens D, Fiset P, Gotman J, Evans AC, Jones BE (1997) Regional Cerebral Blood Flow Changes as a Function of Delta and Spindle Activity during Slow Wave Sleep in Humans. *The Journal of Neuroscience* 17:4800-4808.
- Hori T, Hayashi M, Morikawa T (1994) Topographical EEG changes and the hypnagogic experience.
- Hori T, Sugita Y, Koga E, Shirakawa S, Inoue K, Uchida S, Kuwahara H, Kousaka M, Kobayashi T, Tsuji Y (2001) Proposed supplements and amendments to 'a manual of standardized terminology, techniques and scoring system for sleep stages of human subjects', the Rechtschaffen & Kales (1968) standard. *Psychiatry and clinical neurosciences* 55:305-310.
- Horowitz SG, Braun AR, Carr WS, Picchioni D, Balkin TJ, Fukunaga M, Duyn JH (2009) Decoupling of the brain's default mode network during deep sleep. *Proceedings of the National Academy of Sciences of the United States of America* 106:11376-11381.
- Horowitz SG, Fukunaga M, de Zwart JA, van Gelderen P, Fulton SC, Balkin TJ, Duyn JH (2008) Low frequency BOLD fluctuations during resting wakefulness and light sleep: a simultaneous EEG-fMRI study. *Human brain mapping* 29:671-682.
- Johnson LA, Blakely T, Hermes D, Hakimian S, Ramsey NF, Ojemann JG (2012) Sleep spindles are locally modulated by training on a brain-computer interface. *Proceedings of the National Academy of Sciences* 109:18583-18588.
- Karahanoglu FI, Ville DVD (2015) Transient brain activity disentangles fMRI resting-state dynamics in terms of spatially and temporally overlapping networks. *Nature communications* 6:7751.
- Kjaer TW, Law I, Wiltschiøtz G, Paulson OB, Madsen PL (2002) Regional cerebral blood flow during light sleep—a H215O - PET study. *J Sleep Res* 11:201-207.
- Larson-Prior LJ, Zempel JM, Nolan TS, Prior FW, Snyder AZ, Raichle ME (2009) Cortical network functional connectivity in the descent to sleep. *Proceedings of the National Academy of Sciences of the United States of America* 106:4489-4494.
- Laufs H, Holt JL, Elfont R, Krams M, Paul JS, Krakow K, Kleinschmidt A (2006) Where the BOLD signal goes when alpha EEG leaves. *NeuroImage* 31:1408-1418.
- Laufs H, Walker MC, Lund TE (2007) 'Brain activation and hypothalamic functional connectivity during human non-rapid eye movement sleep: an EEG/fMRI study'--its limitations and an alternative approach. *Brain : a journal of neurology* 130:e75; author reply e76.
- Leicht EA, Newman MEJ (2008) Community Structure in Directed Networks. *Phys Rev Lett* 100:118703.
- Liu X, Duyn JH (2013) Time-varying functional network information extracted from brief instances of spontaneous brain activity. *Proceedings of the National Academy of Sciences* 110:4392-4397.
- Mak-McCully RA, Rosen BQ, Rolland M, Regis J, Bartolomei F, Rey M, Chauvel P, Cash SS, Halgren E (2015) Distribution, Amplitude, Incidence, Co-Occurrence, and Propagation of Human K-Complexes in Focal Transcortical Recordings. *eNeuro* 2:ENEURO.0028-0015.2015.
- Maquet P (2000) Functional neuroimaging of normal human sleep by positron emission tomography. *Journal of sleep research* 9:207-231.

- Marshall W, Gomez-Ramirez J, Tononi G (2016) Integrated Information and State Differentiation. *Frontiers in psychology* 7:926.
- Massimini M, Ferrarelli F, Huber R, Esser SK, Singh H, Tononi G (2005) Breakdown of cortical effective connectivity during sleep. *Science* 309:2228-2232.
- Maxim V, Şendur L, Fadili J, Suckling J, Gould R, Howard R, Bullmore E (2005) Fractional Gaussian noise, functional MRI and Alzheimer's disease. *NeuroImage* 25:141-158.
- Mikl M, Mareček R, Hlušík P, Pavlicová M, Drastich A, Chlebus P, Brázdil M, Krupa P (2008) Effects of spatial smoothing on fMRI group inferences. *Magn Reson Imaging* 26:490-503.
- Munkres J (1957) Algorithms for the Assignment and Transportation Problems. *Journal of the Society for Industrial and Applied Mathematics* 5:32-38.
- Murphy K, Fox MD (2016) Towards a consensus regarding global signal regression for resting state functional connectivity MRI. *NeuroImage*.
- Murphy M, Riedner BA, Huber R, Massimini M, Ferrarelli F, Tononi G (2009) Source modeling sleep slow waves. *Proceedings of the National Academy of Sciences* 106:1608-1613.
- Ni Mhuircheartaigh R, Warnaby C, Rogers R, Jbabdi S, Tracey I (2013) Slow-wave activity saturation and thalamocortical isolation during propofol anesthesia in humans. *Sci Transl Med* 5:208ra148.
- Nir Y, Staba RJ, Andrillon T, Vyazovskiy VV, Cirelli C, Fried I, Tononi G (2011) Regional slow waves and spindles in human sleep. *Neuron* 70:153-169.
- Nobili L, De Gennaro L, Proserpio P, Moroni F, Sarasso S, Pigorini A, De Carli F, Ferrara M (2012) Chapter 13 - Local aspects of sleep: Observations from intracerebral recordings in humans. In: *Prog Brain Res*, vol. Volume 199 (Andries Kalsbeek, M. M. T. R. and Russell, G. F., eds), pp 219-232: Elsevier.
- Ogilvie RD (2001) The process of falling asleep. *Sleep medicine reviews* 5:247-270.
- Ogilvie RD, Wilkinson RT (1984) The detection of sleep onset: behavioral and physiological convergence. *Psychophysiology* 21:510-520.
- Olbrich E, Achermann P (2005) Analysis of oscillatory patterns in the human sleep EEG using a novel detection algorithm. *J Sleep Res* 14:337-346.
- Olbrich S, Mulert C, Karch S, Trenner M, Leicht G, Pogarell O, Hegerl U (2009) EEG-vigilance and BOLD effect during simultaneous EEG/fMRI measurement. *NeuroImage* 45:319-332.
- Piantoni G, Halgren E, Cash SS (2016a) The contribution of thalamocortical core and matrix pathways to sleep spindles. *Neural plasticity* 2016:3024342.
- Piantoni G, Halgren E, Cash SS (2016b) Spatiotemporal characteristics of sleep spindles depend on cortical location. *NeuroImage* 146:236-245.
- Piantoni G, Halgren E, Cash SS (2017) Spatiotemporal characteristics of sleep spindles depend on cortical location. *NeuroImage* 146:236-245.
- Picchioni D, Fukunaga M, Carr WS, Braun AR, Balkin TJ, Duyn JH, Horowitz SG (2008) fMRI differences between early and late stage-1 sleep. *Neuroscience letters* 441:81-85.
- Preti MG, Bolton TAW, Ville DVD (2016) The dynamic functional connectome: State-of-the-art and perspectives. *NeuroImage*.
- Raichle ME, MacLeod AM, Snyder AZ, Powers WJ, Gusnard DA, Shulman GL (2001) A default mode of brain function. *Proceedings of the National Academy of Sciences of the United States of America* 98:676-682.
- Rosenberg RS, Van Hout S (2013) The American Academy of Sleep Medicine inter-scoring reliability program: sleep stage scoring. *Journal of clinical sleep medicine : JCSM : official publication of the American Academy of Sleep Medicine* 9:81-87.
- Rubinov M, Sporns O (2010) Complex network measures of brain connectivity: uses and interpretations. *NeuroImage* 52:1059-1069.
- Sāmān PG, Wehrle R, Hoehn D, Spoormaker VI, Peters H, Tully C, Holsboer F, Czisch M (2011) Development of the brain's default mode network from wakefulness to slow wave sleep. *Cerebral cortex* 21:2082-2093.

- Schabus M, Dang-Vu TT, Albouy G, Baiteau E, Boly M, Carrier J, Darsaud A, Degueldre C, Desseilles M, Gais S, Phillips C, Rauchs G, Schnakers C, Sterpenich V, Vandewalle G, Luxen A, Maquet P (2007) Hemodynamic cerebral correlates of sleep spindles during human non-rapid eye movement sleep. *Proceedings of the National Academy of Sciences of the United States of America* 104:13164-13169.
- Shulman GL, Fiez JA, Corbetta M, Buckner RL, Miezin FM, Raichle ME, Petersen SE (1997) Common Blood Flow Changes across Visual Tasks: II. Decreases in Cerebral Cortex. *Journal of cognitive neuroscience* 9:648-663.
- Siclari F, Baird B, Perogamvros L, Bernardi G, LaRocque JJ, Riedner B, Boly M, Postle BR, Tononi G (2017) The neural correlates of dreaming. *Nature neuroscience* 20:872-878.
- Smith SM, Nichols TE, Vidaurre D, Winkler AM, Behrens TE, Glasser MF, Ugurbil K, Barch DM, Van Essen DC, Miller KL (2015) A positive-negative mode of population covariation links brain connectivity, demographics and behavior. *Nature neuroscience* 18:1565-1567.
- Spoormaker VI, Gleiser PM, Czisch M (2012a) Frontoparietal connectivity and hierarchical structure of the brain's functional network during sleep. *Frontiers in neurology* 3:80.
- Spoormaker VI, Schroter MS, Andrade KC, Dresler M, Kiem SA, Goya-Maldonado R, Wetter TC, Holsboer F, Samann PG, Czisch M (2012b) Effects of rapid eye movement sleep deprivation on fear extinction recall and prediction error signaling. *Human brain mapping* 33:2362-2376.
- Spoormaker VI, Schroter MS, Gleiser PM, Andrade KC, Dresler M, Wehrle R, Samann PG, Czisch M (2010) Development of a large-scale functional brain network during human non-rapid eye movement sleep. *The Journal of neuroscience : the official journal of the Society for Neuroscience* 30:11379-11387.
- Stickgold R (2005) Sleep-dependent memory consolidation. *Nature* 437:1272-1278.
- Tagliazucchi E, Balenzuela P, Fraiman D, Chialvo DR (2012a) Criticality in large-scale brain FMRI dynamics unveiled by a novel point process analysis. *Frontiers in physiology* 3:15.
- Tagliazucchi E, Carhart-Harris R, Leech R, Nutt D, Chialvo DR (2014) Enhanced repertoire of brain dynamical states during the psychedelic experience. *Human brain mapping* 35:5442-5456.
- Tagliazucchi E, Crossley N, Bullmore ET, Laufs H (2016) Deep sleep divides the cortex into opposite modes of anatomical-functional coupling. *Brain Struct Funct* 221:4221-4234.
- Tagliazucchi E, Laufs H (2014) Decoding wakefulness levels from typical fMRI resting-state data reveals reliable drifts between wakefulness and sleep. *Neuron* 82:695-708.
- Tagliazucchi E, von Wegner F, Morzelewski A, Borisov S, Jahnke K, Laufs H (2012b) Automatic sleep staging using fMRI functional connectivity data. *NeuroImage* 63:63-72.
- Tagliazucchi E, von Wegner F, Morzelewski A, Brodbeck V, Borisov S, Jahnke K, Laufs H (2013a) Large-scale brain functional modularity is reflected in slow electroencephalographic rhythms across the human non-rapid eye movement sleep cycle. *NeuroImage* 70:327-339.
- Tagliazucchi E, von Wegner F, Morzelewski A, Brodbeck V, Jahnke K, Laufs H (2013b) Breakdown of long-range temporal dependence in default mode and attention networks during deep sleep. *Proceedings of the National Academy of Sciences of the United States of America* 110:15419-15424.
- Tassi P, Muzet A (2000) Sleep inertia. *Sleep medicine reviews* 4:341-353.
- Terzano MG, Parrino L, Sherieri A, Chervin R, Chokroverty S, Guilleminault C, Hirshkowitz M, Mahowald M, Moldofsky H, Rosa A, Thomas R, Walters A (2001) Atlas, rules, and recording techniques for the scoring of cyclic alternating pattern (CAP) in human sleep. *Sleep medicine* 2:537-553.
- Tononi G (2008) Consciousness as integrated information: a provisional manifesto. *Biol Bull* 215:216-242.
- Tononi G, Edelman GM (1998) Consciousness and complexity. *Science* 282:1846-1851.
- Uhrig L, Dehaene S, Jarraya B (2014) Cerebral mechanisms of general anesthesia. *Annales francaises d'anesthesie et de reanimation* 33:72-82.

- Vidaurre D, Abeysuriya R, Becker R, Quinn AJ, Alfaro-Almagro F, Smith SM, Woolrich MW (2017a) Discovering dynamic brain networks from big data in rest and task. *NeuroImage*.
- Vidaurre D, Quinn AJ, Baker AP, Dupret D, Tejero-Cantero A, Woolrich MW (2016) Spectrally resolved fast transient brain states in electrophysiological data. *NeuroImage* 126:81-95.
- Vidaurre D, Smith SM, Woolrich MW (2017b) Brain network dynamics are hierarchically organized in time. *Proceedings of the National Academy of Sciences of the United States of America* 114:12827-12832.
- von Wegner F, Tagliazucchi E, Laufs H (2017) Information-theoretical analysis of resting state EEG microstate sequences - non-Markovianity, non-stationarity and periodicities. *NeuroImage* 158:99-111.
- Vyazovskiy VV, Olcese U, Hanlon EC, Nir Y, Cirelli C, Tononi G (2011) Local sleep in awake rats. *Nature* 472:443-447.
- Watanabe T, Kan S, Koike T, Misaki M, Konishi S, Miyauchi S, Miyahsita Y, Masuda N (2014) Network-dependent modulation of brain activity during sleep. *NeuroImage* 98:1-10.
- Wei Y, Colombo MA, Ramautar JR, Blanken TF, van der Werf YD, Spiegelhalder K, Feige B, Riemann D, Van Someren EJW (2017) Sleep Stage Transition Dynamics Reveal Specific Stage 2 Vulnerability in Insomnia. *Sleep* 40:zsx117-zsx117.
- Werth E, Achermann P, Borbely AA (1997) Fronto-occipital EEG power gradients in human sleep. *Journal of sleep research* 6:102-112.
- Wong CW, Olafsson V, Tal O, Liu TT (2013) The amplitude of the resting-state fMRI global signal is related to EEG vigilance measures. *NeuroImage* 83:983-990.

Reviewers' comments:

Reviewer #1 (Remarks to the Author):

The authors provided a very detailed response and addressed most of my concerns. I appreciate the effort and fairness in reporting (e.g. about the default mode network gateway that turned out to be threshold-specific). A few issues remain:

The new title is somewhat confusing as there is no NREM sleep cycle, although sleep cycles contain a NREM part.

Re comment #3

My question tried to address the conceptual advancement of the paper rather than making a case for or against PSG. It's interesting that there is a good overlap between manually scored PSG and a data-driven fMRI approach, but that's a methodological feat not necessarily a conceptual advancement. The question is to what extent does the paper go beyond that and in my reading, there seem to be three main issues: the multiple states of N1, the transition matrix and the pre-/post-sleep wakefulness differences.

As the authors now also clearly describe in the revised version, the higher state repertoire in N1 is fully in line with the current consensus on N1 being a mixed stage with features of wakefulness and sleep. That post-sleep wakefulness is different from pre-sleep wakefulness is also to be expected given the earlier EEG/PET work that is now included in the revised version - and various studies on sleep inertia, for instance employing EEG or cognitive tests. If you ask sleep researchers whether they expect that brain activity in the first 10-15 min after awakening is the same as pre-sleep brain activity, not many would say yes. But that there are multiple ways in which sleep progresses and brain states transition, that is very novel and a clear advancement of our concept of sleep. A good illustration of this is the 'strong triangular transition structure within the blue module between the N2-specific whole-brain network states (HMM states 3 and 6) and the N3-modelling HMM state 16' (page 13, small typo in the sentence). However, when it turns out that such a transition matrix is just reflecting some basic EEG patterns, this would reduce the conceptual advancement. After all, N3 could follow N2 sleep with a relative high incidence of specific N2-events versus N2 sleep with no (or different) N2-events.

Examining this does not require a temporal dissection of such events at the millisecond level and examining global-local differences of certain events is not needed initially. The authors just have to show that these HMM states do not merely reflect such basic sleep EEG events, which is easiest and most conclusive to do for the N2 states initially. If the EEG recordings of these states do not yield any differences (expressed in effect size due to the relatively low power) in for instance spindle and K-complex incidence, or fast versus slow spindles, and/or there is no temporal correlation between HMM states and EEG events, that will strengthen the impact of the findings considerably. Since the authors have both the HMM time-courses and the EEG recordings, computing such differences and/or correlations (or some better measures the authors might propose) should be straightforward.

Re comment #8

I agree with the authors that the effect of smoothing (i.e. creating artificial dependencies between regions) should be relatively minor and I can live with the authors not analyzing this given the ratio between amount of effort and expected benefits. Yet a discussion of this limitation as done in their reply to this comment would improve the paper, I thought particularly the authors' second argument was persuasive. The other arguments were not as convincing:

(i) one could say that creating artificial dependencies could still result in systemic biases at the whole brain level;

(iii) do the authors assume that all voxel signals are stationary over time, say for instance the BOLD signal from the CSF (affected by physiological noise), that may get smoothed into the thalamus time-course?

(iv) the AAL also contains smaller regions that may play a disproportionate role;

(v) artificial dependencies could increase or decrease or the overlap with PSG.

Re comment #9

The authors compellingly argue that a functional atlas derived from the data might not be that appropriate for their data, although I had a functional hotspot atlas in mind that has already been published (not derived from the data itself). But going with this line of reasoning, I also expect as the authors state that their results 'would be robust to different levels of spatial granularity (expressed through different parcellations)'. Moreover, in their answer to comment #8 the authors argued that 'the HMM is in itself a form of dimensionality reduction, and therefore, when run at the whole-brain level, it is less sensitive to local variability'. Showing instead of expecting this would be helpful at this stage, otherwise it remains unclear if the results are atlas-specific (and for instance the consequence of atlas-specific artefacts).

Reviewer #2 (Remarks to the Author):

The authors have done a commendable job addressing my critique/questions and the manuscript appears improved. Two of my critique points however were unsatisfactorily answered in my view.

- My previous point 2: The number of HMM states and their brain distribution were determined on 57 subjects, where there was only 10% deep sleep. Given the limit on the total number of states, this biases the analysis toward finding more states during stages that are more prevalently represented in the data. This affects the states found in the final analysis on 18 subjects, and the conclusion on how "diverse" each sleep stage is in terms of number of HMM states. This should be acknowledged by the authors and the description in the paper should reflect this.

- My previous point 5: Contrary to what the authors claim, Tagliazucchi 2012b does not show that RETROICOR effectively removes heart beat and respiratory effects. Rather, it shows that sleep characterization with fMRI suffers from removal of these physiological signals. To be clear, RETROICOR by no means allows comprehensive removal of these signals, and in fact was not designed to do that: for example it does not remove cardiac rate effects. Thus, it is obvious that physiological effects remain in the data and that these, at least in part, affect the fMRI-based sleep staging and sleep -specific "networks". In this regard, it is important to realize that currently a number of wrist-worn devices exists that allow sleep staging based on physiological signals alone and this has in fact a long history. So I think it quite naive to think that the fMRI data analyzed in this work is free of physiological effects and these would not in part affect the HMM states. At a minimum, the potential (an likelihood) of these signal affecting the data and results should be made clear in the discussion.

Reviewer #3 (Remarks to the Author):

Thank you for showing the reproducibility and the robustness of these results. That makes the foundation of your claims even more solid.

Reviewers' comments:

Reviewer #1 (Remarks to the Author):

The authors provided a very detailed response and addressed most of my concerns. I appreciate the effort and fairness in reporting (e.g. about the default mode network gateway that turned out to be threshold-specific). A few issues remain:

We thank the Reviewer. We have put a lot of effort into making sure that our reporting of our results is fair and balanced.

The new title is somewhat confusing as there is no NREM sleep cycle, although sleep cycles contain a NREM part.

We thank the Reviewer for pointing this out and have updated the title of the revised ms: "Discovery of key whole-brain transitions and dynamics during human wakefulness and non-REM sleep".

Re comment #3

My question tried to address the conceptual advancement of the paper rather than making a case for or against PSG. It's interesting that there is a good overlap between manually scored PSG and a data-driven fMRI approach, but that's a methodological feat not necessarily a conceptual advancement. The question is to what extent does the paper go beyond that and in my reading, there seem to be three main issues: the multiple states of N1, the transition matrix and the pre-/post-sleep wakefulness differences.

As the authors now also clearly describe in the revised version, the higher state repertoire in N1 is fully in line with the current consensus on N1 being a mixed stage with features of wakefulness and sleep. That post-sleep wakefulness is different from pre-sleep wakefulness is also to be expected given the earlier EEG/PET work that is now included in the revised version - and various studies on sleep inertia, for instance employing EEG or cognitive tests. If you ask sleep researchers whether they expect that brain activity in the first 10-15 min after awakening is the same as pre-sleep brain activity, not many would say yes. But that there are multiple ways in which sleep progresses and brain states transition, that is very novel and a clear advancement of our concept of sleep. A good illustration of this is the 'strong triangular transition structure within the blue module between the N2-specific whole-brain network states (HMM states 3 and 6) and the N3-modelling HMM state 16' (page 13, small typo in the sentence). However, when it turns out that such a transition matrix is just reflecting some basic EEG patterns, this would reduce the conceptual advancement. After all, N3 could follow N2 sleep with a relative high incidence of specific N2-events versus N2 sleep with no (or different) N2-events.

Examining this does not require a temporal dissection of such events at the millisecond level and examining global-local differences of certain events is not needed initially. The authors just have to show that these HMM states do not merely reflect such basic sleep EEG events, which is easiest and most conclusive to do for the N2 states initially. If the EEG recordings of these states do not yield any differences (expressed in effect size due to the relatively low power) in for instance spindle and K-complex incidence, or fast versus slow spindles, and/or there is no temporal correlation between HMM states and EEG events, that will strengthen the impact of the findings considerably. Since the authors have both the HMM time-courses and the EEG recordings, computing such differences and/or correlations (or some better measures the authors might propose) should be straightforward.

We thank the Reviewer for these helpful and important insights (and have corrected the typo on page 13). In order to further ascertain the validity of our findings, we have carried out the analysis comparing HMM states with EEG events. As may be seen from the new Figures S19, S20 and S21, the results show no significant differences between either sleep spindles or K-complexes and the N2-specific HMM states. This finding provides valuable information to the understanding of the inferred HMM states, as we note in the description of the added analysis:

p. 22, paragraph 4:

“We tested the temporal relationships between the data-driven HMM states and the occurrence of sleep spindles, as identified in the EEG (see Figures S19–S20 and Methods), and indeed we found HMM states 3 and 6 to account for the majority of time where sleep spindles were present. However, the same was true for K-complexes. When identifying K-complexes from the EEG and comparing their occurrences with the HMM states, we once again found HMM states 3 and 6 accounting for the majority (see Figures S19 and S21). While HMM states 3 and 6 correlated more with, and expressed higher specificity to, both sleep spindles and K-complexes than the other HMM states, the numeric values of both correlation and specificity were low (see Figures S20–S21). This may seem surprising, since both types of graphoelements have previously been shown to be reliably represented as distinct event-related patterns in the BOLD signal^{63, 64, 65, 95}. Here it is important to note that unlike event-related analyses, our HMM analysis did not include any convolution of the hemodynamic response according to the sleep graphoelements. We followed the AASM criteria to identify the sleep graphoelements, thus employing a homogenous categorisation of both spindles and K-complexes. There is, however, growing evidence that these electrophysiological events may in fact be more heterogeneous than previously thought; in the spatial sense as illustrated by descriptions of local and more widespread spindles^{6, 8} and K-complexes⁹⁶, and in the temporal sense in the distinction between slow and fast spindles^{63, 97}. Furthermore, the low correlation and specificity values of the data-driven HMM states could be explained by a potential presence of spindles and K-complexes that were not detected due to sources too deep or at “blind” angles to the EEG electrodes. Finally, it is worth appreciating the possibility that the HMM is in fact suggesting an alternative state-description for N2 sleep that do not depend crucially on the presence of sleep graphoelements.”

p. 37, paragraph 4:

“Relationship between HMM states and sleep graphoelements

In order to determine the effect of micro-structural features in the sleep EEG (sleep graphoelements) on the HMM states we used information on the occurrence of sleep spindles and K-complexes during the fMRI recordings. The procedure for obtaining this information from the EEG for the present data has previously been described in Jahnke et al.¹⁴⁰. Briefly, sleep graphoelements were manually identified according to the criteria set out in the AASM guidelines³. This included the use of an EEG montage with frontal, central, and occipital electrodes referenced to the contra-lateral mastoid electrodes (TP9, TP10). The resulting temporal markings of sleep spindles and K-complexes were re-sampled to the sampling frequency of the fMRI acquisition (TR = 2.08 seconds) and collected in the variables SS-timecourse and KC-timecourse (for illustration of the SS- and KC-timecourse in an example participant, see Figure S19). Specifically, SS-timecourse and KC-timecourse were binary and of the same length as the fMRI data, with ones representing the fMRI samples during which the respective graphoelement occurred. We evaluated in turn the temporal association of each HMM activity timecourse to both sleep spindles and K-complexes. Three summary measures of association were used: i) Pearson’s correlation was computed between each of the HMM state timecourses and the SS- and KC-timecourses within the set of participants that included the given graphoelement (see Table S3 for an overview of the occurrence of sleep spindles and K-complexes). ii) Sensitivity to sleep spindles/K-complexes was quantified for each HMM state as the proportion of sleep spindles/K-complexes occurring during that given HMM state. iii) Specificity for sleep spindles/K-complexes was defined for each HMM

state as the likelihood of finding that given HMM state active during an instance of the given graphoelement. The distributions across participants of these three summary measures are plotted in Figures S20 (for sleep spindles) and S21 (for K-complexes).

To test if any HMM states expressed higher association with the sleep graphoelements than others, we used t-tests comparing each combination of the 19 HMM states ($n_{\text{comparisons}} = [19 \times 19 - 19] / 2 = 171$). To establish a chance level we compared the original summary measures (correlation, sensitivity, and specificity) to surrogate data created by permuting the HMM state timecourses 1000 times, and re-calculating the summary measures for each permutation. Each permutation consisted in a random switching of the labels of each instance of an HMM state, keeping the number of occurrences of each HMM state and state transition times constant within participants (see Figure S19 for an illustration of the permutation principle). “

Supplementary materials, p. 1:

“

Sleep graphoelement	Mean count (for participants with count > 0)	Number of participants with count > 0 (% of 57)
Sleep spindles	29.42 (S.D. 29.36)	33 (57.89 %)
K-complexes	32.73 (S.D. 29.19)	37 (64.91 %)

Table S 3 Summary statistics of sleep graphoelements in dataset The EEG acquired simultaneously with the fMRI was used to identify sleep graphoelements. This scoring information was re-sampled to the fMRI, such that volumes during which either a sleep spindle or a K-complex occurred were marked. The first column of the table shows the mean number of occurrences of each sleep graphoelement after this re-sampling. The mean value is calculated within the participants that included at least one of the given graphoelement. The number of participants including at least on sleep spindle or K-complex is shown in the first and second row, respectively, of the second column of the table.

”

Supplementary materials, p. 33:

“

Figure S 19 Presence of sleep graphoelements in example participant. **A)** Plotted together are the 19 HMM state timecourses, the markers of sleep spindles and K-complexes (SS-timecourse and KC-timecourse), and the PSG scoring for an example participant. The SS- and KC-timecourses were based on AASM scoring of sleep graphoelements in the EEG data. Specifically, SS-timecourse and KC-timecourse were binary and of the same length as the fMRI data, with ones representing the fMRI samples during which the respective graphoelement occurred **B)** The same information for the same participant is plotted again, however this time the HMM state timecourses have been randomly permuted. Each permutation consisted in a random switching of the labels of each instance of an HMM state, keeping the number of occurrences of each HMM state and state transition times constant within participants. This was done 1000 times for the purpose of comparing correlation, sensitivity, and specificity of the HMM states to the presence of sleep graphoelements (see Figures S20 and S21).

Figure S 20 Relationship between the 19 HMM states and the presence of sleep spindles. **A)** The violin plot shows the distribution of Pearson's correlation values computed between the timecourse of each HMM state and the timecourse of sleep spindles for each of the 57 participants that included sleep spindles (see Table S3 for summary statistics of sleep spindles). The black crosses denote the means across participants. **B)** The sub-diagonal part of the 19×19 matrix includes the t-statistics resulting from paired t-tests on the correlation values between each pair of the 19 HMM states. A '*' in the center of an entry (x, y) denotes a significant difference between the corresponding pair of HMM state X and Y as evaluated through 1000 random permutations of the HMM state timecourses, at a significance level that has been Bonferroni-corrected for the multiple comparisons between pairs of HMM states. **C)** Distributions of sensitivity of each of the 19 HMM states to sleep spindles across participants. Sensitivity was defined as the proportion of sleep spindles that occurred within a given HMM state. **D)** The 19×19 matrix shows the t-statistics resulting from paired t-tests between sensitivity scores of each pair of HMM states. Significant differences were again evaluated through random permutations of the HMM state timecourses and Bonferroni-correction. **E)** Distributions of the 19 HMM states' specificity for sleep spindles across participants. Specificity was defined as the likelihood of finding a given HMM state active during a spindle, i.e. the ratio of an HMM state's occurrences taking place during spindles. **F)** 19×19 matrix showing the t-statistics resulting from paired t-tests, this time between specificity values of each pair of HMM states. Significant differences were again evaluated through random permutations of the HMM state timecourses and Bonferroni-correction. **G)** For reference are included the circle plots, used throughout the manuscript, indicating the specificity of each HMM state to the sleep stages, calculated for the 18 participants that included all sleep stages. It is clear that sleep spindles correlated higher with the HMM states with high specificity for N2 sleep. HMM states 3 and 6 were thus found to correlate significantly higher with spindles than most of other HMM states, while no significant difference were found between the two. HMM states 3 and 6 also accounted for the majority of spindle occurrences, as quantified through their sensitivity. Given the generally low specificity values, it is also clear that no HMM state occurred exclusively during spindles.

Figure S 21 Relationship between the 19 HMM states and the presence of K-complexes. **A)** The violin plot shows the distribution of Pearson’s correlation values computed between the timecourse of each HMM state and the timecourse of K-complexes for each of the 57 participants that included K-complexes (see Table S3 for summary statistics of K-complexes). The black crosses denote the means across participants. **B)** The sub-diagonal part of the 19×19 matrix includes the t-statistics resulting from paired t-tests on the correlation values between each pair of the 19 HMM states. A ‘*’ in the center of an entry (x, y) denotes a significant difference between the corresponding pair of HMM state X and Y as evaluated through 1000 random permutations of the HMM state timecourses, at a significance level that has been Bonferroni-corrected for the multiple comparisons between pairs of HMM states. **C)** Distributions of sensitivity of each of the 19 HMM states to K-complexes across participants. Sensitivity was defined as the proportion of K-complexes that occurred within a given HMM state. **D)** The 19×19 matrix shows the t-statistics resulting from paired t-tests between sensitivity scores of each pair of HMM states. Significant differences were again evaluated through random permutations of the HMM state timecourses and Bonferroni-correction. **E)** Distributions of the 19 HMM states’ specificity for K-complexes across participants. Specificity was defined as the likelihood of finding a given HMM state active during a K-complex, i.e. the ratio of an HMM state’s occurrences taking place during K-complexes. **F)** 19×19 matrix showing the t-statistics resulting from paired t-tests, this time between specificity values of each pair of HMM states. Significant differences were again evaluated through random permutations of the HMM state timecourses and Bonferroni-correction. **G)** For reference are included the circle plots, used throughout the manuscript, indicating the specificity of each HMM state to the sleep stages, calculated for the 18 participants that included all sleep stages. It is clear that the HMM states relate to K-complexes in a fashion highly similar to that of spindles, presented in Figure 19. K-complexes correlated higher with the HMM states with high specificity for N2 sleep. HMM states 3 and 6 were thus found to correlate significantly higher with K-complexes than most of other HMM states, while no significant difference were found between the two. HMM states 3 and 6 also accounted for the majority of spindle occurrences, as quantified through their sensitivity. Given the generally low specificity values, it is also clear that no HMM state occurred exclusively during K-complexes. “

Re comment #8

I agree with the authors that the effect of smoothing (i.e. creating artificial dependencies between regions) should be relatively minor and I can live with the authors not analyzing this given the ratio between amount of effort and expected benefits. Yet a discussion of this limitation as done in their reply to this comment would improve the paper, I thought particularly the authors' second argument was persuasive. The other arguments were not as convincing:

- (i) one could say that creating artificial dependencies could still result in systemic biases at the whole brain level;*
- (iii) do the authors assume that all voxel signals are stationary over time, say for instance the BOLD signal from the CSF (affected by physiological noise), that may get smoothed into the thalamus time-course?*
- (iv) the AAL also contains smaller regions that may play a disproportionate role;*
- (v) artificial dependencies could increase or decrease or the overlap with PSG.*

We have added the following caveat about the potential limitations of smoothing to the *Discussion* under *Methodological considerations*:

“In addition, it should be noted that there could be potentially confounding effects of spatially smoothing the fMRI data, which can create artificial dependencies between regions of interest. In the context of HMM, which focuses on the aspects of the data that represent more variance, this confound is likely to be minor.”

Re comment #9

The authors compellingly argue that a functional atlas derived from the data might not be that appropriate for their data, although I had a functional hotspot atlas in mind that has already been published (not derived from the data itself). But going with this line of reasoning, I also expect as the authors state that their results 'would be robust to different levels of spatial granularity (expressed through different parcellations)'. Moreover, in their answer to comment #8 the authors argued that 'the HMM is in itself a form of dimensionality reduction, and therefore, when run at the whole-brain level, it is less sensitive to local variability'. Showing instead of expecting this would be helpful at this stage, otherwise it remains unclear if the results are atlas-specific (and for instance the consequence of atlas-specific artefacts).

We agree with the Reviewer that it is important to confirm the robustness of our results across different levels of spatial granularity. We have carried out further analyses to test the HMM in a different parcellation. This is summarised in the text and the Figures S22, S23, S24 and S25. Overall, we now show that the results are highly comparable with an atlas of finer spatial detail (the Brainnetome atlas with 246 regions). As we report in the revised ms, this increase in spatial resolution requires a very minor compensation in the HMM pipeline and particular in the level of PCA preprocessing (keeping 85% as opposed to 90% of the variance). Our interpretation of this is that the increased spatial detail leads to a decrease in signal-to-noise ratio, which can be counterbalanced by keeping less of the variance from the PCA.

p. 31, final paragraph:

“Alternative parcellations, such as those derived from FC configurations in the data, could be problematic, since FC has been shown to robustly vary across the sleep cycle ^{49, 50}, however please see *Robustness across different parcellations* for a demonstration of the robustness of our results using an alternative parcellation.”

p. 38, paragraph 3:

“Robustness across different parcellations

In order to make sure that the use of the HMM generalises to different levels of spatial granularity and that the interpretation following from our results were not specific to the use of the AAL atlas, we re-ran the HMM with a different parcellation. While the field of proposed parcellations for large-scale neuroimaging is rapidly expanding ¹⁴¹, we opted for the Brainnetome atlas originally published by Fan and colleagues ¹⁴². Unlike many of the most popular parcellation schemes, the Brainnetome is not solely derived from fMRI functional connectivity (FC), but also depends on structural connectivity information for its partitioning of the brain volume. As alluded to in the section *HMM general overview*, the use of an FC-derived atlas could bias results, since FC has a well-established dependence on vigilance. Another advantage of the Brainnetome is that it, like the AAL, includes sub-cortical regions, which, as shown in the *Results* and *Discussion*, undergo important changes in activity across NREM sleep. Finally, the 246 regions of the Brainnetome atlas compared to the 90 regions of the AAL provides a good test for the robustness of the HMM across different levels of spatial granularity.

We followed exactly the same steps as explained above, but extracted ROI timecourses from the Brainnetome atlas instead of the AAL. It became clear that the increase in spatial detail, going from the AAL to the Brainnetome, had an impact on the ability of the HMM to track the sleep scoring. As such, when using 90% of the variance from the PCA on the Brainnetome ROI timecourses (step ii in Figure 1), the performance of the HMM, as quantified through MANOVA between the resulting HMM state timecourses and the sleep scoring, was inferior to the original results using the AAL (see Figure S22). However, a slightly stronger regularisation of the ROI timecourses, using only 85% of the variance, made the results from the Brainnetome highly comparable to the original results using the AAL. At 85% of the variance the HMM on the Brainnetome data performed in a very similar fashion to the HMM on the 90% of the AAL data, in terms of MANOVA and the development of median fractional occupancy across number of HMM states (see Figure S22B and C). The difference between using 90% and 85% of the variance was importantly also evident in the number of HMM states that were consistent across participants for a given HMM solution ($K = 19$, see Figure S22D). For 90% of the variance, only 6 HMM states occurred in more than 25% of the participants, whereas this number increased to 12, when 85% of the variance was used. In Figure S23 we have re-constructed Figure 3 of the main text, but with the results using 19 HMM states on 85% of the variance of the Brainnetome data. The results are highly consistent, with individual HMM states showing sensitivity and specificity to different sleep stages, and in terms of the differences in dynamics found between sleep stages. Regarding the spatial configuration of the HMM states resulting from the Brainnetome data, these were also highly consistent with the original HMM states using the AAL. This is illustrated in Figures S24 and S25, where we have matched HMM states from the Brainnetome to the original HMM states, based on their specificity profiles to sleep stages and spatial patterns.

Overall, the above analysis shows that increasing the spatial granularity by introducing a different parcellation comes at the cost of decreasing the signal-to-noise ratio on the HMM estimation. However when this is controlled through PCA, results can be brought to convergence.”

Figure S 22 Performance of HMM using the Brainnetome atlas **A)** Curve showing the cumulative percentage of variance represented by the components of the PCA performed on the ROI timecourses extracted from the Brainnetome atlas (see Figure 1B for an equivalent plot for the AAL data). The blue and the red dashed lines show the two cases analysed; 40 PC's ~ 85% and 70 PC's ~ 90%, respectively. **B)** Plot showing the development across HMM model orders (from 5 to 45 states) of the MANOVA performance of the HMM when compared to the EEG-based sleep scoring for the 18 participants that included all PSG stages (see Figure S 1B for an equivalent plot for the AAL data). Going from 90% of the variance (red) to 85% of the variance (blue) had a significant effect on how well the HMM states related to the PSG scoring. Notice how the red line rarely goes below the zone representing the permuted, random cases, whereas the blue line emulates the original analysis on the AAL data (see curve in Figure S 1B). **C)** Tracking of the median fractional occupancy of the HMM states across model orders, within the 18 participants that included all

four PSG stages. In blue is shown the curve for 40 PC's ~ 85%, while the result using 70 PC's ~ 90% is shown in red (An equivalent plot for the AAL data may be found in Figure S 1C). The fact that the red line is consistently lower than the blue suggests that using the higher percentage of variance implied a high occurrence of 'sporadic' HMM states that accounted for only small portions of the data. This is also evident in **D**) where the HMM solution using 19 states are shown in more detail, for 40 PC's on the left and for 70 PC's on the right. These plots are equivalent to that of Figure S 6B, which pertains to the original HMM on the AAL data, and show the percentage of participants that did not include each of the 19 HMM states. Using 40 PC's ~ 85 % of variance produced a result more similar to the original HMM on the AAL, with 12 HMM states being included in more than 25% of the participants, whereas including 70 PC's ~ 90% meant that only 6 HMM states were represented in more than 25% of the participants.

Figure S 23 Robustness of HMM results when using an alternative parcellation (Brainnetome) (Equivalent to Figure 3 but for the HMM run on 40 PC's ~ 85% of the variance of the Brainnetome ROI timecourses with 19 states) **A)** Select HMM states account for the majority of different PSG stages as quantified through their fractional occupancies. **B)** In the same way as for the original analysis on the AAL data there is an overlap between the HMM states with high sensitivity for a given PSG stage and the HMM states with high specificity for the same PSG stage. **C)** HMM states with high specificity for N3 sleep expressed higher mean life times, as was the case for the original analysis. **D)** The relative as well as the absolute values of switching were very similar to those of the original analysis on the AAL data. **E)** Similarly to the switching dynamics in **E)**, the ranges of unique HMM states visited within each PSG stage were similar to the original analysis. Please note that all plots were calculated from the 18 participants that included all PSG stages, and that significant differences between HMM states or PSG stages were calculated in the same way as for Figure 3.

Figure S 24 Spatial correspondence between HMM states from AAL and Brainnetome in wakefulness-related HMM states. (Brain plots from the Brainnetome data are extracted from the HMM solution with 19 states on the 40 PC's ~ 85% of the variance). To demonstrate the correspondence between the original HMM solution on the AAL data and the HMM solution on the Brainnetome data, the original brain plots of mean activation distributions (from Figure 5) are shown together with brain plots from the HMM on Brainnetome. These have been matched based on visual similarity between the spatial maps and their specificity profiles for PSG stages, represented in pie plots. **A)** The original wake-related HMM states from the AAL together with HMM states from the Brainnetome. Notice the high correspondence not only in spatial distribution but also in the PSG-specificity. The original HMM state 8 appeared to show similarity to

two HMM states from the Brainnetome analysis (HMM states 16 and 3). **B)** The original three WASO-related HMM states appeared to have two equivalents in the Brainnetome solution.

Figure S 25 Spatial correspondence between HMM states from AAL and Brainnetome in N1-, N2-, and N3-related HMM states. Equivalent to Figure S 24, but for N1-, N2-, and N3-sleep. **A)** The original N1-related HMM states found two equivalents from the Brainnetome HMM states. **B)** Each of the three original N2-related HMM states had equivalents from the Brainnetome HMM, which was also the case for the original N3-related HMM state shown in **C).**

Reviewer #2 (Remarks to the Author):

The authors have done a commendable job addressing my critique/questions and the manuscript appears improved.

We thank the Reviewer and agree that the ms has improved.

Two of my critique points however were unsatisfactorily answered in my view.

- My previous point 2: The number of HMM states and their brain distribution were determined on 57 subjects, where there was only 10% deep sleep. Given the limit on the total number of states, this biases the analysis toward finding more states during stages that are more prevalently represented in the data. This affects the states found in the final analysis on 18 subjects, and the conclusion on how “diverse” each sleep stage is in terms of number of HMM states. This should be acknowledged by the authors and the description in the paper should reflect this.

We agree with the Reviewer that the unevenness of PSG stages could lead to PSG-stage-specific increases in signal-to-noise ratios, which in turn could affect the HMM states assigned to ‘similar’ periods of fMRI data (where we in this regard may define ‘similar’ as belonging to the same PSG stage). We prefer to think of it as differences in confidence on the estimation between the PSG stages. Quantifying this difference is difficult however, since the signal-to-noise ratio may differ inherently between each of the PSG stages. We do not believe that the effect of increasing the amount of data from one PSG stage is simply to increase the number of HMM states assigned to this stage. Had this been the case then we would have expected to find the highest diversity of states in wakefulness, since this PSG stage was by far the most frequently represented in the full dataset of 57 participants. Yet in fact it turned out that N1 showed the highest HMM state diversity of the included PSG stages.

We have added the following discussion to our *Methodological considerations*:

p. 26, final paragraph:

“For the HMM analysis we chose to make use of the full dataset of 57 participants, when inferring the states. Subsequently we analysed the part of the HMM solution that corresponded to the 18 participants that included all PSG stages. We chose to include as much data as possible for the initial inference in order to maximise the signal-to-noise ratio and amount of evidence for the HMM parameter estimation. Yet, as it is clear from Table S1, the full dataset included rather uneven distributions of PSG stages. While PSG stages are more evenly distributed in the subset of 18 participants, and we diligently made sure to normalise the relevant summary measures by number of samples of PSG stages within participants, there still exists a possibility that the HMM could be biased by having more data available from certain PSG stages than others, even if the HMM remained uninformed of the PSG staging. For instance, one could imagine that more data from a certain PSG stage would lead to more HMM states per time being assigned to data from that PSG stage. Such an effect is not immediately present in our results, however, since for instance we found both switching and range of HMM states to be higher in N1 sleep, even though the original full dataset included more than twice as much data from wakefulness (see Table S1 and Figure 3D–E).”

- My previous point 5: Contrary to what the authors claim, Tagliazucchi 2012b does not show that RETROICOR effectively removes heart beat and respiratory effects. Rather, it shows that sleep characterization with fMRI suffers from removal of these physiological signals. To be clear, RETROICOR by no means allows comprehensive removal of these signals, and in fact was not designed to do that: for example it does not remove cardiac rate effects. Thus, it is obvious that physiological effects remain in the data and that these, at least in part, affect the fMRI-based sleep

staging and sleep -specific “networks”. In this regard, it is important to realize that currently a number of wrist-worn devices exists that allow sleep staging based on physiological signals alone and this has in fact a long history. So I think it quite naïve to think that the fMRI data analyzed in this work is free of physiological effects and these would not in part affect the HMM states. At a minimum, the potential (an likelihood) of these signal affecting the data and results should be made clear in the discussion.

We thank the Reviewer for this comment, and have now added the following to our *Methodological considerations* (and clarified in the Methods):

“Sleep is of course a process associated with profound physiological changes, not merely those reflected in brain activity. Despite our use of the RETROICOR method to reduce the effects on the fMRI data of cardiac and respiratory signals (see Methods), it is currently not possible to completely isolate the neural effects of sleep in fMRI. Our results hence share the limitation with other neuroimaging studies of sleep of being potentially being influenced by physiological changes not directly linked to brain activity. On the other hand, certain sleep-dependent peripheral changes such as those of the autonomic nervous system will also induce genuine activities in the brain, which in future studies would be important to investigate and with the proper recordings could potentially be evaluated within an HMM framework.”

Reviewer #3 (Remarks to the Author):

Thank you for showing the reproducibility and the robustness of these results. That makes the foundation of your claims even more solid.

We thank the Reviewer for the highly constructive feedback.

Reviewers' comments:

Reviewer #1 (Remarks to the Author):

I'd like to thank the authors for addressing my concerns in such an extensive and professional manner. I have one last question and one last recommendation.

The answer on the overlap between HMM state timecourses and N2 events was highly informative, but one thing seemed to be missing: why were the HMM state timecourses correlated to the actual electrophysiological events and not (additionally) to the HRF-convolved spindle and K-complex timecourses? Even though the goal is to go beyond EEG events and the authors provide convincing arguments in favor of that, it is not that convincing if there is a low correlation between an EEG event and the BOLD signal of the same volume, since there is a delay in the BOLD response to these neurophysiological events. I'm not sure why the authors decided to provide only these direct temporal correlations, but please provide the ones with the HRF-convolved timecourses as well. I don't think it'd be problematic if the correlations (or sensitivity/specificity) turn out to be higher, as long as there are no clear differences between the HMM states most strongly correlated to the N2-events. One bad outcome would be, for instance, that N2 HMM state X is more strongly correlated to one event, N2 HMM state Y to the other, and N2 HMM state Z to N2 sleep without such events. It appears from the data that this is not the case (e.g. the authors report no differences in spindle and K-complex occurrence between two of the N2 states). This should be shown with the HRF-convolved spindle and K-complex timecourses in a comparison among the N2 states. For excluding a difference, i.e. ensuring there are no differences among the states, a multiple-test corrected threshold is unnecessary.

My recommendation is that the authors might not want to lean too heavily on local versus global differences or temporal dissection of events in their interpretation (p22, paragraph 4). Sleep spindles are typically understood as thalamocortical events and their frequency could be subject-specific. It might be helpful to interpret these data more in the light of the notion that there seems to be much more going on than the scalp EEG suggests, without whether this 'deep' brain activity is bound to the same events, albeit at a more local level or with varying temporal characteristics.

Reviewer #2 (Remarks to the Author):

The authors have satisfactorily addressed my critiques.

Reviewers' comments:

Reviewer #1 (Remarks to the Author):

I'd like to thank the authors for addressing my concerns in such an extensive and professional manner. I have one last question and one last recommendation.

We would like to thank the reviewer for many insightful inputs.

The answer on the overlap between HMM state timecourses and N2 events was highly informative, but one thing seemed to be missing: why were the HMM state timecourses correlated to the actual electrophysiological events and not (additionally) to the HRF-convolved spindle and K-complex timecourses? Even though the goal is to go beyond EEG events and the authors provide convincing arguments in favor of that, it is not that convincing if there is a low correlation between an EEG event and the BOLD signal of the same volume, since there is a delay in the BOLD response to these neurophysiological events. I'm not sure why the authors decided to provide only these direct temporal correlations, but please provide the ones with the HRF-convolved timecourses as well. I don't think it'd be problematic if the correlations (or sensitivity/specificity) turn out to be higher, as long as there are no clear differences between the HMM states most strongly correlated to the N2-events. One bad outcome would be, for instance, that N2 HMM state X is more strongly correlated to one event, N2 HMM state Y to the other, and N2 HMM state Z to N2 sleep without such events. It appears from the data that this is not the case (e.g. the authors report no differences in spindle and K-complex occurrence between two of the N2 states). This should be shown with the HRF-convolved spindle and K-complex timecourses in a comparison among the N2 states. For excluding a difference, i.e. ensuring there are no differences among the states, a multiple-test corrected threshold is unnecessary.

My recommendation is that the authors might not want to lean too heavily on local versus global differences or temporal dissection of events in their interpretation (p22, paragraph 4). Sleep spindles are typically understood as thalamocortical events and their frequency could be subject-specific. It might be helpful to interpret these data more in the light of the notion that there seems to be much more going on than the scalp EEG suggests, without whether this 'deep' brain activity is bound to the same events, albeit at a more local level or with varying temporal characteristics.

We thank the reviewer for alluding to this. We agree that the addition of an analysis convolving the sleep spindles and K-complexes with the HRF makes for a more comprehensive investigation of this question. We have now performed this analysis, which as it turns out provides results that are very much in line with the initial analysis using the raw spindle and K-complex timecourses. As the reviewer predicted the overall values of correlation and specificity increased slightly when applying the HRF convolution. Importantly, however, this was true for both HMM state 3 and HMM state 6, which are the two states accounting for the majority of N2 sleep. Hence the differences between HMM states 3 and 6 remain non-significant when correcting for multiple comparisons. The reviewer requests the non-corrected p-values on the tests comparing HMM states 3 and 6; for both sleep spindles and K-complexes HMM state 3 tended to show higher values than state 6 in correlation and specificity at a significance level of $p < 0.05$, non-corrected. Crucially this did not depend on the use of the HRF, but was also true in our initial analysis using the raw spindle and K-complex timecourses. Furthermore, this means that one state tended to be more selective for both graphoelements, and the results does therefore not follow “the bad outcome” posited by the reviewer, where one HMM state would simply reflect sleep spindles and the other K-complexes. On the contrary the values of correlation, sensitivity, and specificity that we now report with and without HRF convolution strongly suggest that both graphoelements are included without real signs of exclusivity in both HMM state 3 and HMM state 6.

We report these findings and address the reviewer's points more specifically in the revised ms:

Page 22, paragraph 4:

“Two HMM states were clearly specific to periods of N2 sleep as two whole-brain network states. The corresponding mean activation maps showed either increases or decreases in areas consistently identified in a number of studies as fMRI-correlates of sleep spindles^{63, 64, 65}. We tested the temporal relationships between the data-driven HMM states and the occurrence of sleep spindles, as identified in the EEG (see Figures S19–S20 and Methods), and indeed we found HMM states 3 and 6 to account for the majority of time where sleep spindles were present. However, the same was true for K-complexes. When identifying K-complexes from the EEG and comparing their occurrences with the HMM states, we once again found HMM states 3 and 6 accounting for the majority (see Figures S19 and S21). In line with previous studies investigating the effects of spindles and K-complexes on the BOLD signal^{63, 65, 95} we also looked at the temporal relationships to the HMM states after convolving the sleep graphoelements with the canonical hemodynamic response function (HRF, see Methods and Figure S19 for illustration). This led to an increase in the correlation and specificity values of the HMM states, but overall did not change the differences on these scores between the states. HMM states 3 and 6 still accounted for the majority of the HRF-convolved sleep spindles and K-complexes. Within HMM states 3 and 6 we did not find marked differences in their relationship to the graphoelements. Both with and without the HRF convolution HMM state 3 tended to show higher correlation and specificity values than HMM state 6 for both spindles and K-complexes, although these differences did not survive correction for multiple comparisons, as shown in Figures S20 and S21. The theoretical scenario that the two HMM states (3 and 6), accounting for the majority of N2 sleep, might represent direct reflections of different sleep graphoelements does thus not seem to be supported. Rather, both of these HMM states included each of their share of both spindles and K-complexes. In light of previous event-related demonstrations of robust effects of sleep graphoelements on the BOLD signal^{63, 64, 65, 95} as well as the similarity between these event-related patterns and the mean activation maps of HMM states 3 and 6 it seems unlikely that spindles and K-complexes did not influence the HMM state description, however this did not result in any HMM state coding exclusively for one or the other graphoelement. There is of course the possibility that HMM states 3 and 6 are suggesting a categorisation of spindles and K-complexes beyond what the scalp EEG is able to resolve, or at least beyond the classical interpretation of these graphoelements. Future mapping of the spectral and spatial properties of these graphoelements at higher resolutions than the AASM criteria, that were used here, could bring more insights in this regard, as could a combination or comparison with intracortical evidence^{6, 8, 96}.”

We have updated the methods to include our analysis using the HRF convolution:

Page 39, line 1:

“To account for the delay in the BOLD response, we also created versions of the SS- and KC-timecourses convoluted with the canonical hemodynamic response function (HRF). We used the HRF included in the SPM12 function *spm_hrf.m* (<http://www.fil.ion.ucl.ac.uk/spm/>). Specifically, SS-timecourse and KC-timecourse were binary and of the same length as the fMRI data, with ones representing the fMRI samples during which the respective graphoelement occurred, while the HRF-convolved versions were scaled between 0 and 1 with the canonical delays and undershoots (an example of the HRF-convolved timecourses is provided in Figure S19C).”

In the supplementary materials we have updated the relevant figures and legends:

Page 32:

Figure S 1 Presence of sleep graphoelements in example participant. **A)** Plotted together are the 19 HMM state timecourses, the markers of sleep spindles and K-complexes (SS-timecourse and KC-timecourse), and the PSG scoring for an example participant. The SS- and KC-timecourses were based on AASM scoring of sleep graphoelements in the EEG data. Specifically, SS-timecourse and KC-timecourse were binary and of the same length as the fMRI data, with ones representing the fMRI samples during which the respective graphoelement occurred **B)** The same information for the same participant is plotted again, however this time the HMM state timecourses have been randomly permuted. Each permutation consisted in a random switching of the labels of each instance of an HMM state, keeping the number of occurrences of each HMM state and state transition times constant within participants. This was done 1000 times for the purpose of comparing correlation, sensitivity, and specificity of the HMM states to the presence of sleep graphoelements (see Figures S20 and S21). **C)** The analyses were also performed after convolution of the SS- and KC-timecourses with the canonical hemodynamic response function (HRF). An illustration of the convolution is shown in an enlarged view of the SS- and KC-timecourses from ~500 to ~2000 seconds. The red timecourses with characteristic delays and undershoots represent the HRF-convoluted SS- and KC-timecourses for this example participant.

”

Figure S 2 Relationship between the 19 HMM states and the presence of sleep spindles. **A)** The grey violin plot shows the distribution of Pearson's correlation values computed between the timecourse of each HMM state and the raw timecourse of sleep spindles for each of the 57 participants that included sleep spindles (see Table S3 for summary statistics of sleep spindles). The black crosses denote the means across participants. In red are shown the outlines of an equivalent violin plot when considering HRF-convolved versions of the spindle timecourses. **B)** The sub-diagonal part of the 19×19 matrix includes the t-statistics resulting from paired t-tests on the correlation values between each pair of the 19 HMM states. The super-diagonal part includes the t-stats considering the HRF-convolved versions of the spindle timecourses. A '*' in the center of an entry (x, y) denotes a significant difference between the corresponding pair of HMM state X and Y as evaluated through 1000 random permutations of the HMM state timecourses, at a significance level that has been Bonferroni-corrected for the multiple comparisons between pairs of HMM states. **C)** Distributions of sensitivity of each of the 19 HMM states to sleep spindles across participants for raw spindle timecourses (grey) and HRF convolved spindles (red). Sensitivity was defined as the proportion of sleep spindles that occurred within a given HMM state. **D)** Equivalent to B but for sensitivity values. **E)** Distributions of the 19 HMM states' specificity for sleep spindles across participants for raw spindle timecourses (grey) and HRF convolved spindles (red). Specificity was defined as the likelihood of finding a given HMM state active during a spindle, i.e. the ratio of an HMM state's occurrences taking place during spindles. **F)** Equivalent to B and D but for specificity values. **G)** For reference are included the circle plots, used throughout the manuscript, indicating the specificity of each HMM state to the sleep stages, calculated for the 18 participants that included all sleep stages. It is clear that sleep spindles correlated higher with the HMM states with high specificity for N2 sleep. HMM states 3 and 6 were thus found to correlate significantly higher with spindles than most of the other HMM states, while no significant difference were found between the two. This was true regardless of HRF convolution of the spindles. HMM states 3 and 6 also accounted for the majority of spindle occurrences, as quantified through their sensitivity. Given the generally low specificity values, it is also clear that no HMM state occurred exclusively during spindles.

“

Figure S 3 Relationship between the 19 HMM states and the presence of K-complexes. **A)** The grey violin plot shows the distribution of Pearson’s correlation values computed between the timecourse of each HMM state and the raw timecourse of K-complexes for each of the 57 participants that included K-complexes (see Table S3 for summary statistics of K-complexes). The black crosses denote the means across participants. In red are shown the outlines of an equivalent violin plot when considering HRF-convolved versions of the KC-timecourses. **B)** The sub-diagonal part of the 19×19 matrix includes the t-statistics resulting from paired t-tests on the correlation values between each pair of the 19 HMM states. The super-diagonal part includes the t-stats considering the HRF-convolved versions of the KC-timecourses. A ‘*’ in the center of an entry (x, y) denotes a significant difference between the corresponding pair of HMM state X and Y as evaluated through 1000 random permutations of the HMM state timecourses, at a significance level that has been Bonferroni-corrected for the multiple comparisons between pairs of HMM states. **C)** Distributions of sensitivity of each of the 19 HMM states to K-complexes across participants for raw KC-timecourses (grey) and HRF convolved K-complexes (red). Sensitivity was defined as the proportion of K-complexes that occurred within a given HMM state. **D)** Equivalent to B but for sensitivity values. **E)** Distributions of the 19 HMM states’ specificity for K-complexes across participants for raw KC-timecourses (grey) and HRF convolved K-complexes (red). Specificity was defined as the likelihood of finding a given HMM state active during a K-complex, i.e. the ratio of an HMM state’s occurrences taking place during K-complexes. **F)** Equivalent to B and D but for specificity values. **G)** For reference are included the circle plots, used throughout the manuscript, indicating the specificity of each HMM state to the sleep stages, calculated for the 18 participants that included all sleep stages. It is clear that the HMM states relate to K-complexes in a fashion highly similar to that of spindles, presented in Figure 19. K-complexes correlated higher with the HMM states with high specificity for N2 sleep. HMM states 3 and 6 were thus found to correlate significantly higher with K-complexes than most of other HMM states, while no significant difference were found between the two. HMM states 3 and 6 also accounted for the majority of K-complex occurrences, as quantified through their sensitivity. Given the generally low specificity values, it is also clear that no HMM state occurred exclusively during K-complexes.

”

Reviewer #2 (Remarks to the Author):

The authors have satisfactorily addressed my critiques.

We thank the reviewer for the helpful comments and questions.